



# Review Article: Scaling, dynamical regimes and stratification: How long does weather last? How big is a cloud?

Shaun Lovejoy[1]

[1] Physics, McGill University, 3600 University st., Montreal, Que. H3A 2T8, Canada

*Correspondence to*: Shaun Lovejoy (lovejoy@physics.mcgill.ca)

**Abstract.** Until the 1980's, scaling notions were restricted to self-similar homogeneous special cases. I review developments over the last decades, especially in multifractals and Generalized Scale Invariance (GSI). The former is necessary for characterizing and modelling strongly intermittent scaling processes while the GSI formalism extends scaling to strongly

anisotropic (especially stratified) systems. Both of these generalizations are necessary for atmospheric applications. The theory and (some) of the now burgeoning empirical evidence in its favour is reviewed.

Scaling can now be understood as a very general symmetry principle. It is needed to clarify and quantify the notion of dynamical regimes. In addition to the weather and climate, there is an intermediate "macroweather regime" and at time scales beyond the climate regime, (up to Milankovitch scales) there is a macroclimate and megaclimate regime. By objectively

distinguishing weather from macroweather it answers the question "how long does weather last?". Dealing with anisotropic scaling systems - notably atmospheric stratification – requires new (non-Euclidean) definitions of the notion of scale itself. These are needed to answer the question "how big is a cloud?". In anisotropic scaling systems morphologies of structures change systematically with scale even though there is no characteristic size. GSI shows that it is unwarranted to infer dynamical processes or mechanisms from morphology.

Two "sticking points" preventing the more widespread acceptance of the scaling paradigm are also discussed. The first is an often implicit phenomenological "scalebounded" thinking that postulates *a priori* the existence of new mechanisms, processes every factor of two or so in scale. The second obstacle is the reluctance to abandon isotropic theories of turbulence and accept that the atmosphere's scaling is anisotropic. Indeed there is currently appears to be no empirical evidence that the turbulence in any atmospheric field is isotropic.

Most atmospheric scientists rely on General Circulation Models, and these are scaling – they inherited the symmetry from the (scaling) primitive equations upon which they are built. Therefore, the real consequence of ignoring wide range scaling is that it blinds us to alternative scaling approaches to macroweather and climate - especially to new models for long range forecasts and to new scaling approaches to climate projections. Such stochastic alternatives are increasingly needed notably to reduce uncertainties in climate projections to the year 2100.





# 1. Introduction

## 1.1 Dynamical ranges, fluctuations and scale

Perhaps the most obvious difficulty in understanding the atmosphere is in dealing with its the enormous range of scales. The single picture, fig. 1 shows clouds with horizontal spatial variability ranging from millimeters to the size of the planet, a factor of ten billion in scale. In the vertical direction the range is more modest but still huge: about ten million. The range of temporal variability is extreme, spanning a range of one hundred billion billion: from milliseconds to the planet's age, fig. 2.

The earliest approach to atmospheric variability was phenomenological: weather as a juxtaposition of various processes with characteristic morphologies, airmasses, fronts and the like. Circumscribed by the poor quality and quantity of the then available data, these were naturally associated with narrow scale range, mechanistic processes.

At first, ice ages, "medieval warming" and other evidence of low frequency processes were only vaguely discerned. Weather processes were thought to occur with respect to a relatively constant (and unimportant) background: climate was conceived as simply long term "average" weather. It wasn't until the 1930's, when the International Meteorological Organisation defined "climate normals" in an attempt to quantify the background "climate state". The duration of the normals - 30 years - was imposed essentially by fiat: it conveniently corresponded to the length of high quality data then available: 1900-1930. This 30 year duration is still with us today with the implicit consequence that - purely by convention - "climate change" occurs at scales longer than 30 years.

Interestingly, there has developed yet another official time scale, for defining "anomalies". Again, for reasons of convenience (and partly – for temperatures - due to the difficulty in making absolute measurements), anomalies are defined with respect to monthly averages. Ironically, a month is not even a well-defined unit of time!

The overall consequence of adopting out of convenience, monthly and thirty year time scales, is a poorly theorized, inadequately justified division of atmospheric processes into three regimes: scales less than a month, a month up to 30 years and a lumping together of all slower processes with time scales longer than 30 years. While the high frequency regime is clearly "weather" and the slow processes – at least up to ice age scales -is "climate", until [*Lovejoy*, 2013] the intermediate regime lacked even a name. Using scaling - and with somewhat different transition scales - the three regimes were finally put on an objective quantitative basis with the middle regime baptized "macroweather". By using scaling to quantitatively define weather, macroweather and climate we can finally objectively answer the question: how long does weather last? A bonus, detailed in section 2, is that scaling analyses showed that beyond macroweather, there are three longer time scale dynamical regimes. Rather than lumping all low frequencies together simply as "climate", we must also distinguish macroclimate and megaclimate. What had hitherto been considered simply "climate" is itself composed of three distinct dynamical regimes.

To review how scaling defines dynamical regimes, let's define scaling using fluctuations - for example of the temperature or of a component of the wind. For the moment, consider only one dimension, i.e. time series or spatial transects. Temporal scaling means that the amplitudes of fluctuations are proportional to their time scale raised to a scale invariant exponent. For appropriately nondimensionalized quantities:



$$\text{Fluctuation} = (\text{scale})^{\zeta} \qquad\qquad (1)$$

Every term in this equation needs appropriate definition, but for now, consider the classical ones. First, the usual turbulence definition of a fluctuation is a difference (of temperature, wind components etc.) taken over an interval in space or in time. This interval ("lag") defines the time ($\Delta t$) or space ($\Delta x$) scale of the corresponding fluctuation. Also, classically, one considers the statistically averaged fluctuation (indicated by "$< >$"). If we decompose $\zeta$ into a random singularity $\gamma$ and a nonrandom "fluctuation exponent" $H,$ then the average fluctuation will also be scaling with:

$<\text{Fluctuation}> = (\text{scale})^{H}$;  $\qquad\qquad < (\text{scale})^{\gamma} > = 1$;  $\qquad\qquad \zeta = H + \gamma \qquad (2)$

Where the symbol "$< >$" indicates statistical (ensemble) averaging.

Later in section 2.5, fluctuations as differences (sometimes called "poor man's wavelets") - are replaced by (nearly as simple) Haar fluctuations based on Haar wavelets (see also appendix B) and in section 3, eq. 1 is interpreted stochastically. Finally, in section 4, we generalize the notion of scale itself by introducing a scale function that replaces the usual (Euclidean) distance

function (metric). These anisotropic scale functions are needed to handle scale in two or higher dimensional spaces, especially with regard to stratification.

In atmospheric regimes where equation 1 holds, average fluctuations over durations $\lambda \Delta t$ are $\lambda^{H}$ times those at duration $\Delta t$ i.e. they differ only in their amplitudes, they are qualitatively of the same nature, they are therefore part of the same dynamical regime. More generally (eq. 1), appropriately rescaled probabilities of random fluctuations also have scale invariant exponents

("codimensions", section 3) so that the entire statistical behaviour is scaling. Scaling therefore allows us to objectively identify the different atmospheric regimes.

Over the Phanerozoic eon (the last 540Myrs), the five scaling regimes are weather, macroweather, climate, macroclimate and megaclimate [*Lovejoy*, 2015]. Starting at around a millisecond (the dissipation time), this covers a total range of $\approx 10^{19}$ in scale (section 2.5). Scaling therefore gives an unequivocal answer to the question posed in the title: "how long does weather

last", the answer is the lifetime of planetary structures, typically around 10 days (section 2.6).

If the key statistical characteristics of the atmosphere at any given scale are determined by processes acting over wide ranges of scale – and not by a plethora of narrow range ones - then we must conclude that the fundamental dynamical processes are in fact dynamical "regimes" – not uninteresting "backgrounds". While there may also be narrow range processes, they can only be properly understood in the context of the dynamical regime in which they operate, and in any event, spectral or other

analysis shows that they generally contribute only marginally to the overall variability. The first task is therefore to define and understand the dynamical regimes and then - when necessary – the narrow range processes occurring within them.



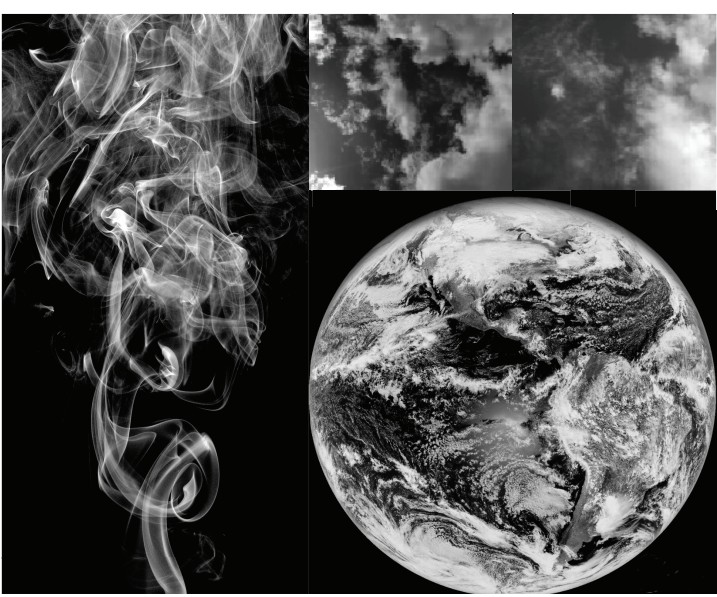

**Fig. 1: Cigarette smoke (left) showing wisps and filaments smaller than a millimeter up to about a meter in overall size. The upper right shows two cloud each several kilometers across with resolutions of a meter or so. The lower right, shows the global scale arrangement of clouds taken from a infra-red satellite image of the earth with a resolution of several kilometers. Taken together, the three images span a range of several billion in spatial scale. Reproduced from [*Lovejoy*, 2019].**

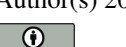
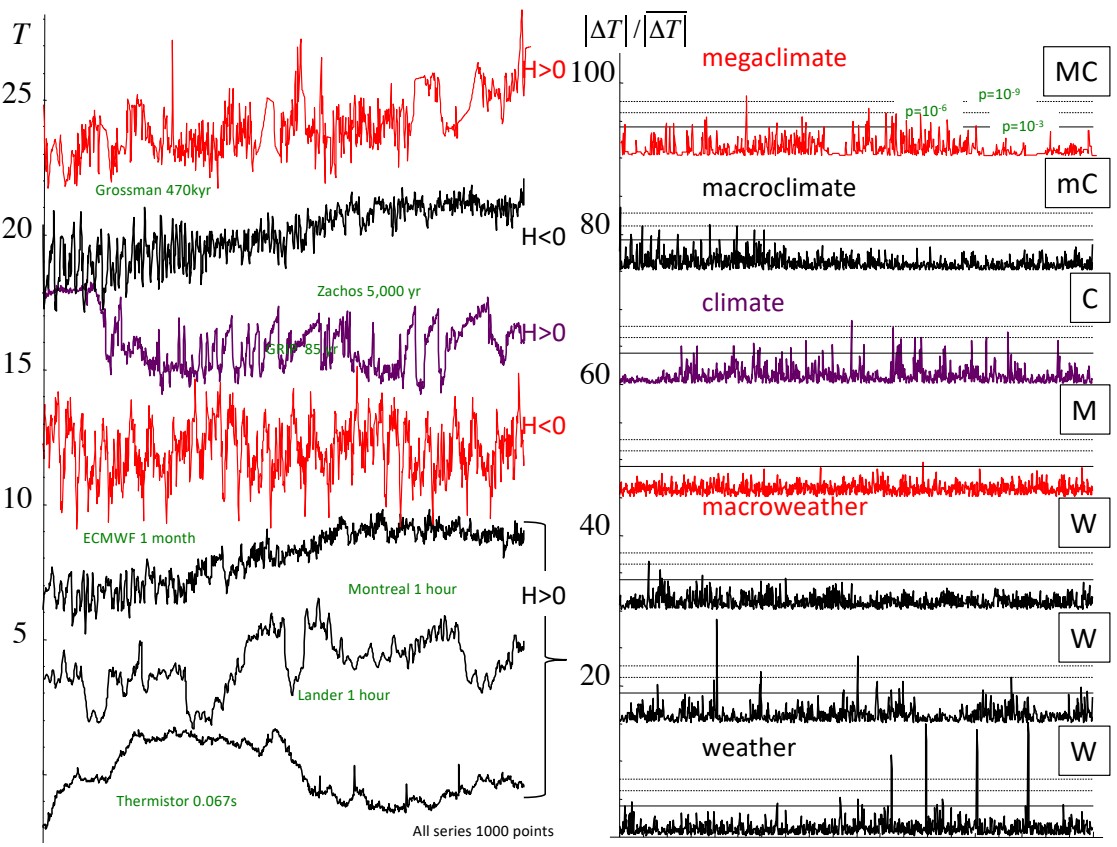

**Fig. 2** Left: 1000 points of various time series collectively spanning the range of scales of 470Myrs to 0.067s = 2.4x10$^{17}$, each series was normalized so as to have the same overall range, and offset in the vertical for clarity. The right hand column shows the absolute first differences normalized by the mean. The solid horizontal line shows the maximum value expected for Gaussian variables ($p$ = 10$^{-3}$), the dashed show the corresponding $p$ = 10$^{-6}$, 10$^{-9}$ probability levels.

Representative series from each of the five scaling regimes taken with the addition of the hourly surface temperatures from Lander Wyoming, (bottom, detrended daily and annually). The Berkeley series was taken from a fairly well estimated period before significant anthropogenic effects and was annually detrended. The top was taken over a particularly data - rich epoch, but there are still traces of the interpolation needed to produce a series at a uniform resolution. The resolutions (indicated) were adjusted so that as much as possible, the smallest scale was at the inner scale of the regime indicated. In the macroclimate regime, the inner scale was a bit too small and the series length a bit too long. The resulting megaclimate regime influence on the low frequencies was therefore removed using a linear trend of 0.25 δ$^{18}$O/Myr. The resolutions and time periods are indicated next to the curves. The black curves have $H$>0, the red, $H$<0, see the parameter estimates in Section A. The figure is from [Lovejoy, 2018] updated only in the top megaclimate series that is at higher resolution than the previous, (from [Grossman and Joachimski, 2022].

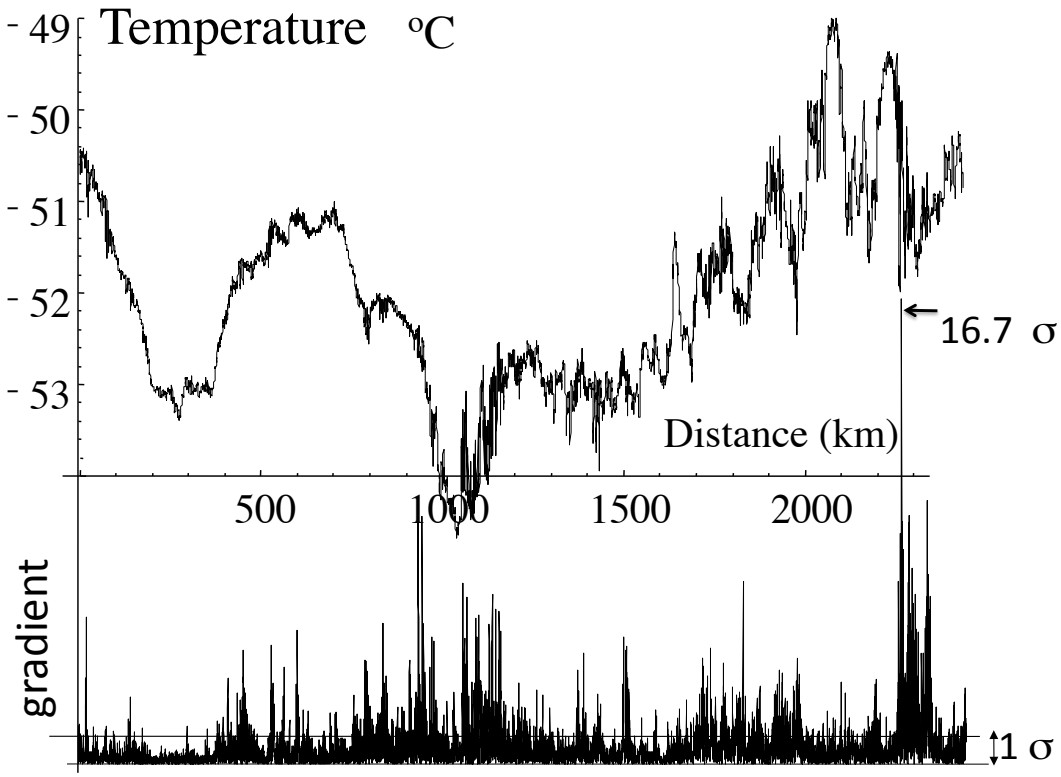

**Fig. 3; The first 8196 points of the temperature series measured by a GulfStream 4 flight over the Pacific Ocean at 196mb and 1 second resolution (corresponding to 280m). Because the aircraft speed is much greater than the wind, this can be considered as a spatial transect. The bottom shows the absolute change in temperature from one measurement to the next normalized by dividing by the typical change (the standard deviation). This differs from the spike plot in the right hand side of fig. 2 only in the**
**normalization: here by the standard deviation, not the absolute difference. Reproduce from [*Lovejoy and Schertzer*, 2013].**

## 1.2 A multiscaling/multifractal complication

In order to answer the quite different scaling question "How big is a cloud?", it is first necessary to discuss a complication: that the scaling is different for every level of activity. It turns out that the wide range over which the variability occurs is only one of its aspects: even at fixed scales, the variability is much more extreme than is commonly believed. Interestingly, the

extremeness of the variability at a fixed scales is a *consequence* of the wide range scaling itself, it allows the variability to build up scale by scale in a multiplicative cascade manner. As a result, mathematically, the scaling of the average fluctuation (eq. 2) gives only a partial view of the variability, we need to consider eq. 1 in its full stochastic sense. In particular, if the exponent $\zeta$ is random, then it is easy to imagine that the variability may be huge.





To graphically see this, it is sufficient to produce a "spike plot" (the right hand columns of fig. 2, time, and the corresponding
spatial plot, fig. 3). These spike plots are simply the absolute first differences in the values normalized by their overall means
(in fig. 3, the normalization is slightly different, by the standard deviaton). In the right hand column of fig. 2 and the bottom
of fig. 3, we see - with a single but significant exception - macroweather in time (fig. 2) – that they all have strong spikes
signalling sharp transitions. In turbulence jargon, the series are highly "intermittent".

How strong are the spikes? Using classical (Gaussian) statistics we may use probability levels quantify it. For example, the
fig. 2 (right) shows solid horizontal lines that indicate the maximum spike that would expected from a Gaussian process with
the given number of spikes. For the 1000 points in each series in fig. 2, this line thus corresponds to a Gaussian probability $p$
$= 10^{-3}$. In addition, horizontal dashed lines show spikes at levels $p = 10^{-6}$, $p = 10^{-9}$. Again with the exception of macroweather,
we see that the $p = 10^{-6}$ level is exceeded in every series and that the megaclimate, climate and weather regimes are particularly
intermittent with spikes exceeding the $p = 10^{-9}$ levels. In section 3 we show how the spikes can be tamed by multifractal
theory and the maxima predicted reasonably accurately (appendix A) by simply characterizing the statistics of the process near
the mean (i.e. using their non extreme) behaviour.

The spikes visually underline the fact that variability is not simply a question of the range of scales that are involved: at any
given scale, variability can be strong or weak. In addition, events can be highly clustered with strong ones embedded inside
weak ones and even stronger ones inside strong ones in a fractal pattern repeating to smaller and smaller scales. And this
fractal sparseness itself can itself become more and more accentuated for the more and more extreme events/ regions: the series
will generally be multifractal.

### 1.3 How big is a cloud?

Scaling is also needed to answer the question "how big is a cloud?" (here "cloud" is taken as a catchall meaning an atmospheric
structure or eddy). Now the problem is what do we mean by "scale"? The series and transects in figs 2, 3, are one dimensional
so that it is sufficient to define the scale of a fluctuation by the duration (time) or length (space) over which it occurs (actually,
time involves causality so that the sign of $\Delta t$ is important, see [*Marsan et al.*, 1996], we ignore this issue here). However the
state of the atmosphere is mathematically represented as fields in three dimensional space evolving in time.

Consider fig. 4 that displays a cloud vertical cross-section from the CloudSat radar. In the figure, the gravitationally induced
stratification is striking, and since each pixel in the figure has a horizontal resolution of 1km but vertical resolution of 250m,
the actual stratification is actually four times stronger than it appears. What is this cloud's scale? If we use the usual Euclidean
distance to determine the scale, should we measure it in the horizontal – or in the vertical direction? In this case, is the cloud
scale its width (200 km) or its height (only ≈10 km)?

If the horizontal/ vertical aspect ratio was the same for all clouds, the two choices would be identical to within a constant
factor, the anisotropy would be "trivial". The trouble is that the aspect ratio itself turns out to be a strong power law function
of (either) horizontal or vertical scale so that for any cloud:





$$\text{Vertical scale} = (\text{Horizontal scale})^{H_z} \tag{3}$$

In section 4, we will see that theoretically, the stratification exponent $H_z = 5/9$, a value that we confirm empirically on various
atmospheric fields.

To further appreciate the issue, consider the simulation in fig. 5 that shows a vertical cross section of a multifractal cloud liquid water density field. The left hand column (top to bottom) shows a series of blow-ups in an isotropic (self-similar) cloud. Moving from top to bottom, blow-ups of the central regions by successive factors of 2.9 are displayed. In order for the cross-sections to maintain a constant 50% "cloud cover", the density threshold distinguishing the cloud (white/grey) from the non-
cloud (black) must be systematically adjusted to account for this change in resolution. This systematic readjustment of the threshold is required due to the multifractality and with this adjustment, we see that the cross-sections are "self-similar" i.e. they look the same at all scales.

The effect of differential (scale dependent) stratification is revealed in the right hand column that shows the analogous zoom through a anisotropic multifractal simulation with a stratification exponent $H_z = 5/9$. The low resolution (top) view of the
simulation is highly stratified in the horizontal. Now, the blow ups reveal progressively more and more roundish structures. Eventually – the bottom cross-section (a blow up of a total factor of $\approx 5000$), we can start to see vertically oriented structures "dangling" below more roundish ones.

In the isotropic simulations (left hand), the only difficulty in defining the size of the cloud is the multifractal problem of deciding, for each resolution, which threshold should be used to distinguish cloud from no cloud. However, in the more
realistic anisotropic simulation on the right, there is an additional difficulty in answering the question of "how big is a cloud?" Should we use the horizontal or the vertical cloud extent? It turns out (in section 4) that to ensure that the answer is well defined, we need a new notion of scale itself: "Generalized Scale Invariance" (GSI).

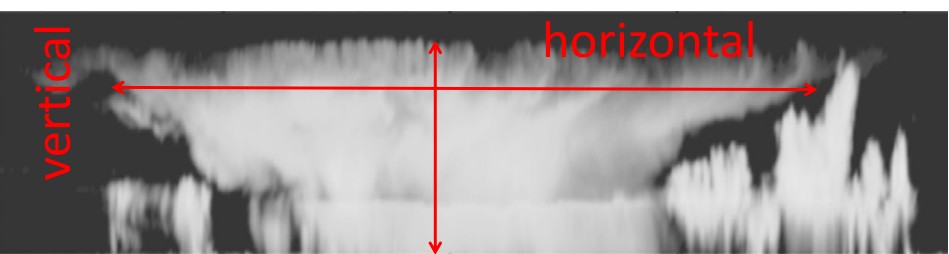

**Fig. 4: A vertical cloud cross section of radar backscatter taken by the radar on the CloudSat satellite with resolution of 250m in the vertical, 1 km in the horizontal. The black areas are those whose radar reflectivities are below the radar's minimum detectable signal. The arrows show rough estimates of the horizontal and vertical extent of the cloud. The two differ by a factor of more than 10. How doe characterize the size of this cloud? Reproduced from. Adapted from [*Lovejoy et al.*, 2009b].**





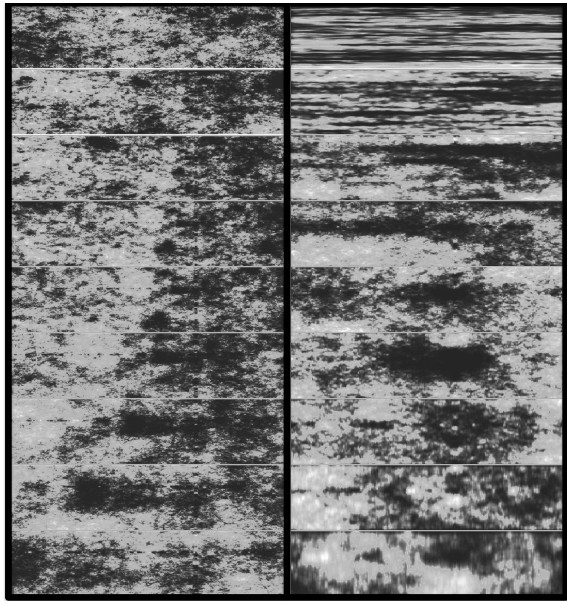

**Fig. 5: Left column: a sequence "zooming" into vertical cross section of an isotropic multifractal cloud (the density of liquid water**
**was simulated and then displayed using false colours with grey sky below a low threshold). From top to bottom, we progressively**
**zoom in by factors of 2.9 (total factor ≈ 1000). We can see that typical cloud structures are self-similar.**

**Right column: A multifractal cloud with the same statistical parameters as at left, but anisotropic, the zoom is still by factors of 2.9**
**in the horizontal, but the structures are progressively "squashed" in the horizontal. Notice that while at large scales, the clouds are**
**strongly horizontally stratified, when viewed close up they show structures in the opposite direction. The sphero-scale is equal to**
**the vertical scale in the right-most simulation on the bottom row The film version of this (and other anisotropic space-time**
**multifractal simulations can be found at: http://www.physics.mcgill.ca/~gang/multifrac/index.htm). Reproduced from [*Blöschl et***
***al.*, 2015].**

## 1.4 Wide range scaling and the scalebound and isotropic turbulence alternatives

### 1.4.1 Comparison with narrow scale range, "scalebound" approaches

The presentation and emphasis of this review reflects experience over the last years that has shown how difficult it is to shake
traditional ways of thinking. In particular, traditional mechanistic meteorological approaches are based on a widely
internalized but largely unexamined "scalebound" view that prevents scaling from being taken as seriously as it must be. As
we'll see (section 2), the scalebound view persists in spite of its increasing divorce from the real world. Such a persistent
divorce is only possible because practising atmospheric scientists rely almost exclusively on Numerical Weather Prediction
(NWP) or Global Circulation Models (GCM) and these inherit the scaling symmetry from the atmosphere's primitive equations
upon which they are built.





The problem with scaleboundedness is not so much that it doesn't fit the facts, but rather that it blinds us to promising alternative scaling approaches. And new approaches are urgently needed. As argued in [*Lovejoy*, 2022b], climate projections based on GCMs are reaching diminishing returns with the latest IPCC AR6 (2021) uncertainty ranges larger than ever before: c.f. the latest climate sensitivity range of $2 - 5.5K$ rise of global temperature following a $CO_2$ doubling. This is currently more than double the range of expert judgement: $(2.5 - 4K)$. New low uncertainty approaches are thus needed and scaling approaches based on direct stochastic scaling macroweather models are promising [*Hébert et al.*, 2021], [*Procyk et al.*, 2022].

### 1.4.2 "Scaling primary" versus "isotropy primary" turbulence approaches

There are also sticking points whose origin is in the other, statistical, turbulence strand of atmospheric science. Historically, turbulence theories have been built around two statistical symmetries: a scale symmetry (scaling) and a direction symmetry (isotropy). While these two are conceptually quite distinct, even today, they are almost invariably considered together in the special case called "self-similarity" that is a basic assumption of theories and models of isotropic two dimensional and isotropic three dimensional turbulence. Formalizing scaling as a (nonclassical) symmetry principle clarifies the distinct nature of scale and direction symmetries. In the atmosphere, due to gravity (not to mention sources of differential rotation) there is no reason to assume that the scale symmetry is an isotropic one: indeed, atmospheric scaling is fundamentally anisotropic. The main unfortunate consequence of assuming isotropy is that it implies an otherwise unmotivated (and unobserved) scale break somewhere near the scale height ($\approx 10km$).

As we show (section 4), scaling accounts for both the stratification that systematically increases with scale as well as its intermittency. Taking into account gravity in the governing equations provides an anisotropic scaling alternative to quasi-geostrophic turbulence ("fractional vorticity equations", see [*Schertzer et al.*, 2012]). The argument in this review is thus that scaling is the primary scale symmetry, it takes precedence over other scale symmetries such as isotropy, indeed, it seems that isotropic turbulence is simply not relevant in the atmosphere.

### 1.5 The scope and structure of this review

This review primarily covers scaling research over the last four decades, especially multifractals, generalized scale invariance and their now extensive empirical validations. This work involved theoretical and technical advances, revolutions in computing power, the development of new data analysis techniques and the systematic exploitation of mushrooming quantities of geodata. The basic work has already been the subject of several reviews ([*Lovejoy and Schertzer*, 2010b], [*Lovejoy and Schertzer*, 2012a], but especially the monograph [*Lovejoy and Schertzer*, 2013]. Although a book covering some of the subsequent developments was published more recently [*Lovejoy*, 2019], it was nontechnical, so that this new review brings its first four chapters up to date and includes some of the theory and mathematics that was deliberately omitted so as to render the material more accessible. The last three chapters of [*Lovejoy*, 2019] focused on developments in the climate (and lower





frequency) scale regimes that will be reviewed elsewhere. This review is thus limited to the (turbulent) weather regime and its transition to macroweather at scales of ≈ 10 days.

In order to maintain focus on the fundamental physical scaling issues and implications, the mathematical formalism is introduced progressively - as needed - so that it will not be an obstacle to accessing the core scientific ideas.

     This review also brings to the fore several advances that have occurred in the last ten years, especially Haar fluctuation analysis (developed in detail in appendix B), and a more comprehensive criticism of scalebound approaches made possible by combining Haar analysis with new high resolution instrumental and paleodata sources [*Lovejoy*, 2015]. On the other hand, it

leaves out an emerging body of work on macroweather modelling based on the Fractional Energy Balance Equation for both prediction and climate projections [*Del Rio Amador and Lovejoy*, 2019; *Del Rio Amador and Lovejoy*, 2021b], [*Del Rio Amador and Lovejoy*, 2021a], [*Procyk et al.*, 2022], as well as their implications for the future of climate modelling [*Lovejoy*, 2022b].

     The presentation is divided into three main sections. Keeping the technical – mathematical aspects to a minimum, section 2

focuses on a foundational atmospheric science issue: what is the appropriate conceptual, theoretical framework for handling the atmosphere's variability over huge ranges of scales? It discusses how the classical scalebound approach is increasingly divorced from real world data and numerical models. Scaling is discussed but with emphasis on its role as a symmetry principle. It introduces fluctuation analysis based on Haar fluctuations that allow for a clear quantitative empirical overview of the variability over seventeen orders of magnitude in time. Scaling is essential to defining the basic dynamical regimes,

underlining the fact that between the weather and the climate sits a new "macroweather" regime.

     Section 3 discusses the general scaling process: multifractals. Multifractals naturally explain and quantify the ubiquitous intermittency of atmospheric processes. The section also discusses an under-appreciated consequence: the divergence of high order statistical moments - equivalently power law probability tails – and relate this to "tipping points" and "black swans". The now large body of evidence for the divergence of moments is discussed and special attention paid to the velocity field

where the divergence of moments was first empirically shown 40 years ago in the atmosphere, then in wind tunnels and most recently in large Direct Numerical Simulations of hydrodynamic turbulence.

     In section 4 a totally different aspect of scaling is covered anisotropic scaling, notably scaling stratification. The section outlines the formalism of Generalized Scale Invariance (GSI) needed to define the notion of scale in anisotropic scaling systems. By considering buoyancy driven turbulence, the 23/9D model is derived, it is a consequence of Kolmogorov scaling

in the horizontal and Bolgiano-Obukhov scaling in the vertical. This model is "in between" flat 2D isotropic turbulence and "voluminous" isotropic 3D turbulence – it is strongly supported by now burgeoning quantities of atmospheric data. It not only allows us to answer the question "how big is a cloud"? but also to understand and model differentially rotating structures needed to quantify cloud morphologies.



## 2 Scaling or scalebound? From van Leuwenhoek to Mandelbrot

**2.1 The scalebound view examined**

In the introduction, the conventional paradigm based on (typically deterministic) narrow range explanations and mechanisms, was contrasted with the alternative scaling paradigm that builds statistical models expressing the collective behaviour of high numbers of degrees of freedom and that provides explanations over huge range of scales.

Let's consider the narrow range paradigm in more detail. It follows in the steps of van Leuwenhoek who - peering through an
early microscope - was famously said to have discovered a "new world in a drop of water" – micro-organisms (circa 1675). Over time, it evolved into a "powers of ten" view ([*Boeke*, 1957]) in which every factor of ten or so of zooming revealed qualitatively different processes, morphologies. [*Mandelbrot*, 1981] termed this view "scalebound" (written as one word), which is a useful short-hand for the idea that every factor of ten or so involves something qualitatively new: a new world, new mechanisms, new morphologies etc.

The first weather maps were at extremely low spatial resolution so that only a rather narrow range of phenomena could be discerned. Unsurprisingly, the corresponding atmospheric explanations and theories were scalebound. Later, in the 1960's and 70's under the impact of new data, especially in the mesoscale, the ambient scalebound paradigm was quantitatively made explicit in space-time "Stommel" diagrams (discussed at length in section 2.6) in which various conventional mechanisms, morphologies, phenomena are representing by the space-time scales over which they operate. For a recent inventory of
scalebound mechanisms from seconds to decades, see [Williams et al., 2017].

While the Stommel diagrams reflected scalebound thinking, their goal was the modest one of categorizing and classifying existing empirical phenomenology and this, in the light of prevailing mechanistic analytic dynamical meteorology approaches and models. It was [*Mitchell*, 1976], writing at the dawn of the paleo-climate revolution who more than anyone, ambitiously elevated the scalebound paradigm into a general framework spanning a range of scales from (at least) an hour to the age of the
planet, (a factor of tens of billion, upper left, fig. 6). Mitchell's data was limited, and he admittedly that his spectrum was only an "educated guess". He imagined that when the data would become available, that their spectra would consist of an essentially uninteresting white noise "background" interspersed with interesting quasi-periodic signals representing the important physical processes. Ironically, Mitchell's scalebound paradigm was proposed at the same time as the first general circulation models (GCMs, [*Manabe and Wetherald*, 1975]). Fortunately, the GCMs are scaling, inheriting the symmetry from the governing
equations ([*Schertzer et al.*, 2012], see ch. 2 of [*Lovejoy and Schertzer*, 2013].

Mitchell's schematic was so successful, that more than four decades later, his original figure is still faithfully reproduced (e.g. [*Dijkstra*, 2013]) or updated by very similar scalebound schematics with only minor updates. Even though the relevant geodata has since mushroomed, the updates notably have *less* quantification and *weaker* empirical support than the original. The forty - five year evolution of the scalebound paradigm is shown in the other panels of fig. 6. Moving to the right in the figure, there
is a twenty - five year update, modestly termed an "artist's rendering" [*Ghil*, 2002]. This figure differs from the original in the excision of the lowest frequencies and by the inclusion of several new multimillennial scale "bumps". In addition, whereas





Mitchell's spectrum was quantitative, the artist's rendering retreated to using "arbitrary units" making it more difficult to empirically verify. Nearly twenty years later, the same authors approvingly reprinted it in a review [*Ghil and Lucarini*, 2020].
As time passed, the retreat from quantitative empirical assessments continued so that the scalebound paradigm has become
more and more abstract. The bottom left in fig. 6, shows an update downloaded from the NOAA paleoclimate data site in 2015, claiming to be a "mental model". Harkening back to [*Boeke*, 1957], the site went on to state that the figure is "intended … to provide a general 'powers of ten' overview of climate variability". Here, the vertical axis is simply "variability" and the uninteresting background - presumably a white noise – is shown as a perfectly flat line.

At about the same time [*Lovejoy*, 2015], pointed out that Mitchell's original figure was in error by an astronomical factor
(section 2.4) so that - in an effort to partially address the criticism – an update in the form of a "conceptual landscape" was proposed (fig. 6 bottom right, [*von der Heydt et al.*, 2021]). Rather than plotting the log of the spectrum $E(\omega)$ as a function of the log frequency $\omega$, the "landscape's" main innovation was the use of the unitless "relative variance" $\omega E(\omega)$ plotted linearly. indicated as a function of log $\omega$ (bottom right of the figure). Such plots have the property that areas under the curves are equal to the total variance contributed over the corresponding frequency range. Before returning to these schematics let's discuss
the scaling alternative.


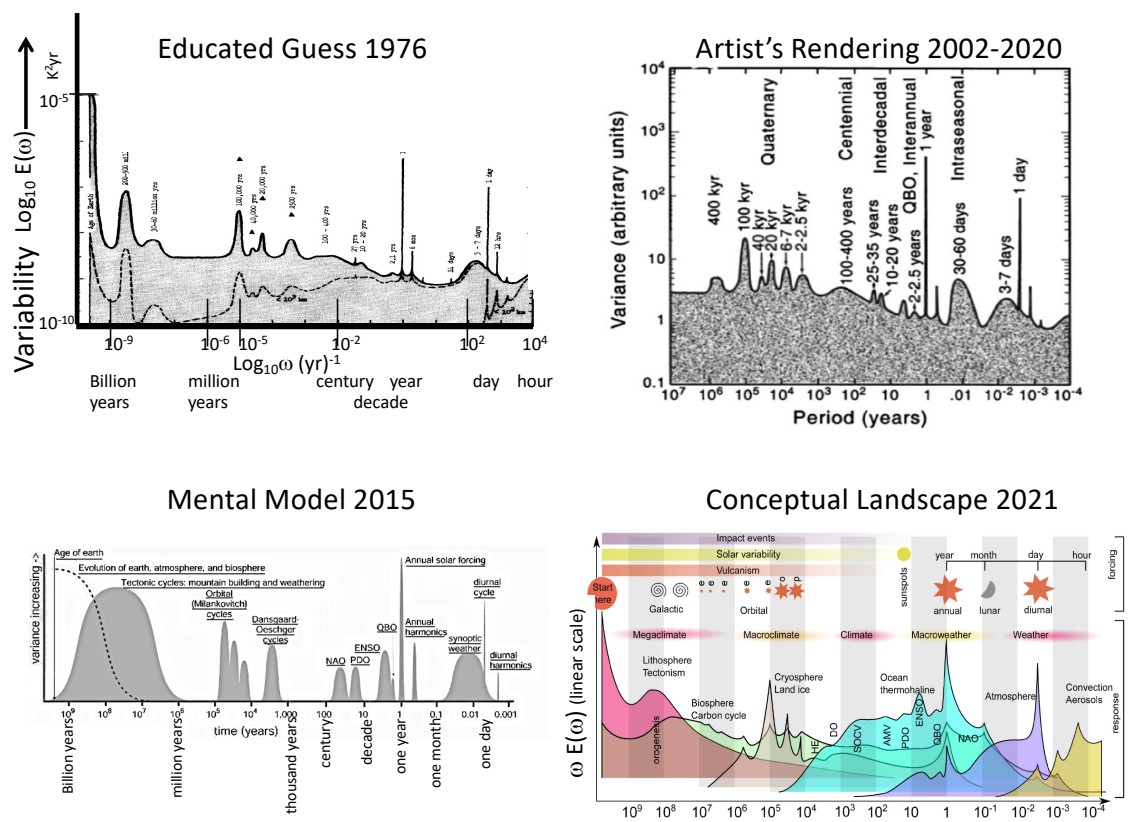

**Fig. 6: The evolution of the scalebound paradigms of atmospheric dynamics, 1976-2021. The upper left "educated guess" is from [*Mitchell*, 1976], the upper right "artist's rendering" from [*Ghil*, 2002], [*Ghil and Lucarini*, 2020]. Lower left shows NOAA's "mental model" (downloaded from the site in 2015), the lower right is the "conceptual model" from [*von der Heydt et al.*, 2021].**

## 2.2 The scaling alternative

### 2.2.1 Scaling as a symmetry

Although scaling in atmospheric science goes back to Richardson in the 1920's, it was the "Fractal Geometry of Nature" [*Mandelbrot*, 1977], [*Mandelbrot*, 1982] that first proposed scaling as a broad alternative to the scalebound paradigm. Alongside deterministic chaos and nonlinear waves, fractals rapidly became part of the nascent nonlinear revolution. Contrary to scaleboundedness, scaling supposes that zooming results in something that is the qualitatively *unchanged*.

Although Mandelbrot emphasized fractal *geometry* i.e. the scaling of geometry sets of points, it soon became clear [*Schertzer and Lovejoy*, 1985a] that the physical basis of scaling (more generally scaling fields, scaling processes) is in fact a scale symmetry principle – effectively a scale conservation law that is respected by many nonlinear dynamical systems, including those governing fluids [*Schertzer and Lovejoy*, 1985c], [*Schertzer and Lovejoy*, 1987; *Schertzer et al.*, 2012].




Scaling is seductive because it is a symmetry. Ever since Noether published her eponymous theorem [*Noether*, 1918] demonstrating the equivalence between symmetries and conservation laws, physics has been based on symmetry principles. Thanks to Noether's theorem, by formulating scaling as a general symmetry principle, the scaling *is* the physics. Symmetry principles represent a kind of maximal simplicity, and since "entities must not be multiplied beyond

necessity" (Occam's razor) physicists always assume that symmetries hold unless there is evidence for *symmetry breaking*.

In the case of fluids, we can verify this symmetry on the equations as implemented for example in GCMs (e.g. [*Stolle et al.*, 2009], [*Stolle et al.*, 2012] and discussion in section 2.2.3)- but only for scales larger than the (millimetric) dissipation scales, where the symmetry is broken and mechanical energy is converted into heat. The scaling is also broken at the large scales by

the finite size of the planet. In between, boundary conditions such as the ocean surface or topography might potentially have broken the scaling but in fact, they turn out to themselves be scaling and so do not introduce a characteristic scale (e.g. [*Gagnon et al.*, 2006]).

In the atmosphere one therefore *expects* scaling. It is expected to hold unless processes can be identified that act preferentially and strongly enough at a specific scales that could break it. This turns the table on scalebound thinking:

if we can explain the atmosphere's structure in a scaling manner, then this is the simplest explanation and should *a priori*, be adopted. The onus must be on the scalebound approach to demonstrate the inadequacy of scaling and the need to replace the hypothesis of a unique wide scaling range regime by (potentially numerous) distinct scalebound mechanisms.

Once a scaling regime is identified – either theoretically or empirically (preferably by a combination of both), it is

associated with a single basic dynamical mechanism that repeats scale after scale over a wide range, hence it provides an objective classification principle.

### 2.2.2 Wide range scaling in atmospheric science

The atmospheric scaling paradigm is almost as old as numerical weather prediction, both being proposed by

Richardson in the 1920's. Indeed, ever since Richardson's scaling 4/3 law of turbulent diffusion ([*Richardson*, 1926] - the precursor of the better known Kolmogorov law [*Kolmogorov*, 1941]) - scaling has been the central turbulence paradigm. From the beginning, Richardson argued for a *wide range scaling*, holding from millimeters to thousands of kilometers (fig. 7). Richardson himself attempted an empirical verification notably using data from pilot balloons and volcanic ash (and later - in the turbulent ocean - with bags of parsnips that he watched diffusing from a bridge [*Richardson and*

*Stommel*, 1948]). However, there remained a dearth of data spanning the key "meso-scale" range ≈ 1 – 100km corresponding to the atmosphere's scale height, so that for several decades following Richardson, progress in atmospheric turbulence was largely theoretical. In particular, in the 1930's the turbulence community made rapid





advances in understanding the simplified isotropic turbulence problem, notably the Karman-Howarth equations [*Karman and Howarth*, 1938]) and the discovery of numerous isotropic scaling laws for passive scalar advection and for mechanically driven

and buoyancy driven turbulence. At first, Kolmogorov and the other the pioneers recognized that atmospheric stratification strongly limited the range of applicability of isotropic laws. Kolmogorov for example, estimated that his famous law of three dimensional isotropic turbulence would only hold at scales below 100m. As discussed in section 4, modern data shows that this was a vast *over*-estimate, that if his isotropic law ever holds anywhere in the atmosphere, that it is below 5m! Yet, at the same time, in the horizontal, the anisotropic generalization of the Kolmogorov law apparently holds up to planetary scales!

In the 1970's motivated by Charney's isotropic 2D geostrophic turbulence [*Charney*, 1971], the ambitious "EOLE" experiment was undertaken specifically to study large scale atmospheric turbulence. EOLE (for the Greek wind God) ambitiously used a satellite to track the diffusion of hundreds of constant density balloons ([*Morel and Larchevêque*, 1974]), but the results turned out to be difficult to interpret. Worse, the initial conclusions – that the mesoscale wind did *not* follow the Kolmogorov law – turned out to be wrong and it was later re-interpreted ([*Lacorta et al.*, 2004], and

then further re-re-interpreted [*Lovejoy and Schertzer*, 2013]) finally vindicating Richardson nearly ninety years later. Therefore, when [*Lovejoy*, 1982], benefitting from modern radar and satellite data, discovered scaling right through the mesoscale, (fig. 7, right), it was the most convincing support to date for Richardson's daring 1926 wide range scaling hypothesis. Although at first, it was mostly cited for its empirical verification that clouds were indeed fractals, today, 40 years later, we increasingly appreciate its vindication of Richardson's scaling from 1 – 1000 km, right through the

mesoscale. It marks the beginning of modern scaling theories of the atmosphere. This has since been confirmed by massive quantities of remotely sensed and in situ data both on earth fig. 8, and more recently on Mars (fig. 9, discussed in detail in section 3.4).

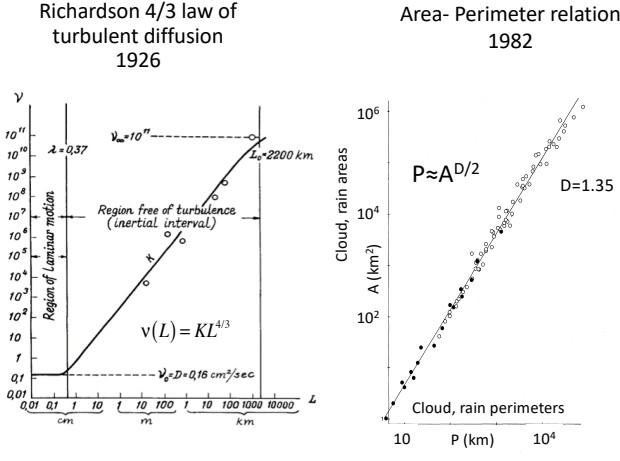




**Fig. 7: Richardson' pioneering scaling model [Richardson, 1926] of turbulent diffusion (left) with an early update [Lovejoy, 1982] (right) using radar rain data (black) and satellite cloud data (open circles).**

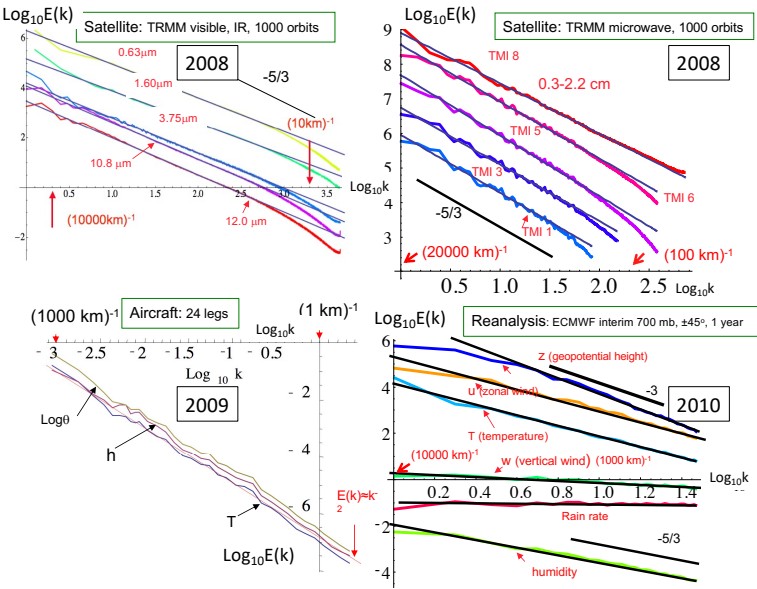

**Fig. 8: Planetary scale power law spectra ($E(k) = k^{-\beta}$) from satellite radiance (top), aircraft bottom left and reanalyses (bottom**
**right).**

**Upper left: Spectra from over 1000 orbits the Tropical Rainfall Measurement Mission (TRMM); of five channels visible through thermal IR wavelengths displaying the very accurate scaling down to scales of the order of the sensor resolution (≈ 10 km). Adapted from [*Lovejoy et al.*, 2008b].**

**Upper left: Spectra from five other (microwave) channels from the same satellite. The data are at lower resolution and the latter**
**depends on the wavelength, again the scaling is accurate up to the resolution. Adapted from [*Lovejoy et al.*, 2008b].**

**Lower left: The spectrum of temperature (*T*), humidity (*h*) and log potential temperature (logθ) averaged over 24 legs of aircraft flight over the Pacific Ocean at 200 mb. Each leg had a resolution of 280m and had 4000 points (1120 km). A reference line corresponding to $k^{-2}$ spectrum is shown in red. The meso-scale (1 – 100 km) is shown between the dashed blue lines. Adapted from [Lovejoy et al., 2010]**

**Lower right: Zonal Spectra of reanalyses from the European Centre for Medium Range Weather Forecasting (ECMWF), once daily for the year 2008 over the band ±45º latitude. Adapted from [Lovejoy and Schertzer, 2011].**



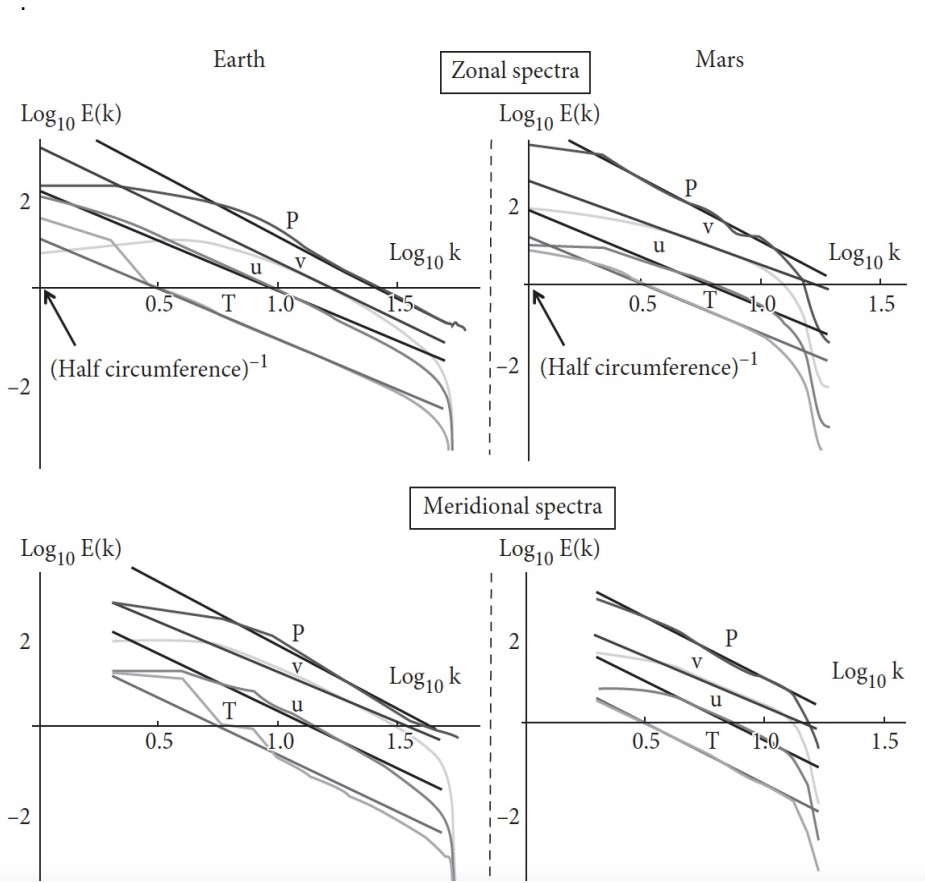

**Fig. 9: Earth (left), Mars (right). The zonal spectra (left) and (right) Mars as functions of the nondimensional wavenumbers for pressure ($p$, purple), meridional wind ($v$ (green), the zonal wind ($u$, blue), temperature ($T$, red). The data for Earth were taken at 69% atmospheric pressure for 2006 between latitudes ±45° latitude. The data for Mars were taken at 83% atmospheric pressure for Martian year 24 to 26 between latitudes ±45°. The reference lines have absolute slopes from top to bottom $\beta$ = 3.00, 2.05, 2.35 and 2.35 (for $p$, $v$, $u$, $T$ respectively). The spectra have been rescaled and an offset added for clarity. Wavenumber $k$ =1 corresponds to the half circumference of the respective planets. Reproduced from [*Chen et al.*, 2016].**

### 2.2.3 Which is more fundamental: scaling or isotropy?

In section 2.1, we discussed the debate between scaling and mechanistic, generally deterministic, "scalebound" approaches. But even in the statistical (turbulence) strand of atmospheric science there evolved an alternative to Richardson's wide range scaling: the paradigm of isotropic turbulence.

In the absence of gravity (or other strong source of anisotropy), the basic isotropic scaling property of the fluid equations has been known for a long time [*Taylor*, 1935], [*Karman and Howarth*, 1938]. The scaling symmetry justifies the numerous classical fluid dynamics similarity laws (e.g. [*Sedov*, 1959]) and it underpins models of statistically isotropic turbulence,





notably the classical turbulence laws of Kolmogorov [*Kolmogorov*, 1941], Bolgiano-Obukov (buoyancy driven, section 4.1) [*Bolgiano*, 1959], [*Obukhov*, 1959] and Corrsin-Obukov (passive scalar) [*Corrsin*, 1951], [*Obukhov*, 1949].

These classical turbulence laws can be expressed in the form:

$$\text{Fluctuation} \approx (\text{turbulent flux})^a (\text{scale})^H \tag{4}$$

where scale was interpreted in an isotropic sense, $H$ is the fluctuation exponent and physically, the turbulent fluxes are the

drivers (compare with eq. 2). The first and most famous example is the Kolmogorov law for fluctuations in the wind where the turbulent flux is the energy rate density ($\varepsilon$, $a = 1/3$) and $H = 1/3$. Eq. 4 is the same as eq. 2 except that the randomness is hidden in the turbulent flux that classically was considered to be quasi-Gaussian, the non-intermittent special case (section 3.2).

Theories and models of isotropic turbulence were developed to understand the fundamental properties of high Reynolds

number turbulence, and this, independently of whether or not it could be applied to the atmosphere. Since the atmosphere is a convenient very high Reynolds number laboratory ($Re \approx 10^{12}$), the question is therefore "Is isotropic turbulence relevant in the atmosphere"? (the title of [*Lovejoy et al.*, 2007]).

Fig. 10 graphically shows the problem: although the laws of isotropic turbulence are themselves scaling, they imply a break in the middle of the "mesoscale" at around 10km. To model the larger scales [*Fjortoft*, 1953], [*Kraichnan*, 1967], soon found

another isotropic scaling paradigm: 2D isotropic turbulence. Charney in particular adapted Kraichnan's 2-D isotropic turbulence to geostrophic turbulence [*Charney*, 1971], the result is sometimes called "layerwise" 2D isotropic turbulence. Whereas Kraichnan's 2D model was rigidly flat with strictly no vortex stretching - Charney's extension allowed for some limited vortex stretching. Fig. 10 shows the implied difference between the 2D isotropic and 3D isotropic regimes.

Even though isotropy had originally been proposed purely for theoretical convenience, armed with two different isotropic

scaling laws, it was now being proposed as the fundamental atmospheric paradigm. If scaling in atmospheric turbulence is always isotropic, then we are forced to accept a scale break. The assumption that isotropy is the primary symmetry, implies (at least) two scaling regimes with a break (presumably) near the 10km scale height i.e in the mesoscale. The 2D-3D model with its implied "dimensional transition" ([*Schertzer and Lovejoy*, 1985a]) already contradicted the wide range scaling proposed by Richardson.

An important point is that the implied scale break is neither physically nor empirically motivated, it is purely a theoretical consequence of assuming the predominance of isotropy over scaling. One is forced to choose: which of the fundamental symmetries is primary; isotropy or scaling?

By the time a decade later that the alternative (wide range) anisotropic scaling paradigm (see fig. 11 for a schematic) was proposed [*Schertzer and Lovejoy*, 1985a], [*Schertzer and Lovejoy*, 1985c], Charney's beautiful theory along with its 2D/3D

scale break had already been widely accepted, and even today it is still taught. More recently, [*Schertzer et al.*, 2012]





Generalized Scale Invariance was linked directly to the governing equations so that a clear anisotropic theoretical alternative to Charney's isotropic theory is available.

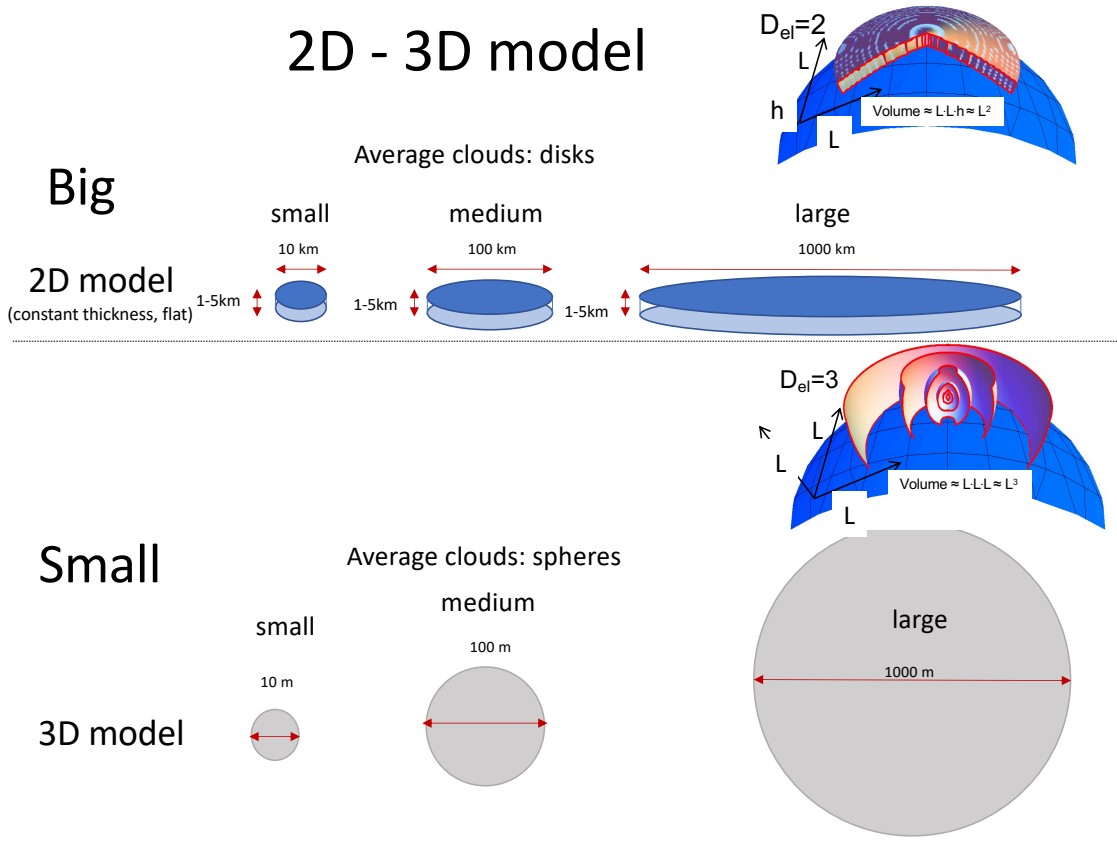


**Fig. 10: A schematic showing the geometry of isotropic 2-D models (top, for the large scales), the volumes of average structures (disks) increase as the square of the disk diameter. The isotropic is a schematic of 3D turbulence models for the small scales, with the volumes of the spheres increasing as the cube of the diameter. These geometries are superposed on the earth's curved surface (the blue spherical segments on the right). We see (the bottom right, earth surface) that - unless they are strongly restricted in range**
**- the 3D isotropic models quickly imply structures that extend into outer space.**

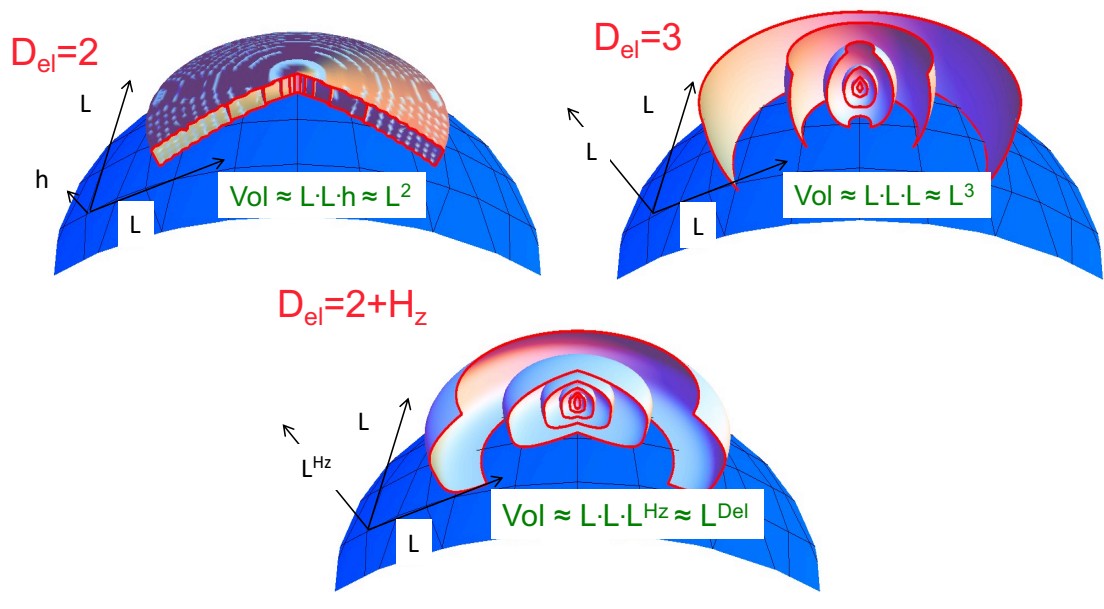

**Fig. 11: A schematic diagram showing the change in shape of average structures which are isotropic in the horizontal (slightly curved to indicate the earth's surface) but with scaling stratification in the vertical; $H_z$ increases from 0 (upper left) to 1 (lower right); $D_{el} = 2+H_z$. In order to illustrate the change in structures with scale, the ratio of tropospheric thickness to earth radius has been increased by nearly a factor of 1000. Note that in the $D_{el} =3$ case, the cross-sections are exactly circles; the small distortion is an effect of perspective due to the mapping of the structures on to the curved surface of the earth. Reproduced from [*Lovejoy and Schertzer, 2010b*].**

## 2.3 Aspects of scaling in one dimension

The basic signature of scaling is a power law relation of a statistical characteristic of a system as a function of space and / or time *scale*. In the empirical test of the Richardson 4/3 law (fig. 7, left), it is the turbulent viscosity as a function of horizontal scale that is a power law. In the right hand side, it is rather the complicated (fractal) perimeters of clouds and rain zones that are power law functions of the corresponding areas. These analysis methods lack generality, let's instead consider spectra (Fourier space) and then, fluctuations (real space, section 2.5, appendix B).

Following Mitchell we may consider variability in the spectral domain, for example, the power spectrum of the temperature $T(t)$ is:




$$E(\omega) \propto \left\langle \left| \widetilde{T}(\omega) \right|^2 \right\rangle \tag{5}$$

where $\widetilde{T}(\omega)$ is its Fourier Transform, $\omega$ is the frequency. A scaling process $E(\omega)$ has the same form if we consider it at a time scale $\lambda$ times smaller or equivalently, at a frequency $\lambda$ times larger:

$$E(\lambda\omega) = \lambda^{-\beta} E(\omega) \tag{6}$$

Where $\beta$ is the "spectral exponent". The solution of this functional equation is a power law:

$$E(\omega) \propto \omega^{-\beta} \tag{7}$$

Therefore, a log-log plot of the spectrum as a function of frequency will be a straightline, see fig. 12 for early quantitative applications to climate series.

Alternatively, we can consider scaling in real space. Due to "Tauberian theorems" (e.g. [*Feller*, 1971]) power laws in real space are transformed into power laws in Fourier space (and visa versa). This result holds whenever the scaling range is wide enough - i.e. even if there are high and or low frequency cut-offs (needed if only for the convergence of the transforms). If we consider fluctuations $\Delta T$ over time interval $\Delta t$, then if the system is scaling, we can introduce the ("generalized", $q^{\text{th}}$ order) structure function as:

$$S_q(\Delta t) = \left\langle \left( \Delta T(\Delta t) \right)^q \right\rangle \propto \Delta t^{\xi(q)} \tag{8}$$

Where the "< >" sign indicates statistical (ensemble) averaging (assuming statistical stationarity there is no $t$ dependence. Once again, classically fluctuations are defined simply as differences i.e. $\Delta T(\Delta t) = T(t) - T(t - \Delta t)$, although more general fluctuations are needed as discussed in section 2.5. For stationary scaling processes, the Wiener-Khintchin theorem implies a simple relation between real space and Fourier scaling exponents:

$$\beta = 1 + \xi(2) \tag{9}$$

If in addition, the system is "quasi-Gaussian" then $S_2$ gives a full statistical characterization of the process, therefore often only the second order structure function $S_2(\Delta t)$ is considered (e.g. fig. 8, top). However as discussed above, geoprocesses are typically strongly intermittent, rarely quasi-Gaussian and the full exponent function $\xi(q)$ is needed, (section 3.1). In the next section, we discuss this figure in more detail, including its physical implications, for the moment simply note the various linear (scaling) regimes on the log-log plots.



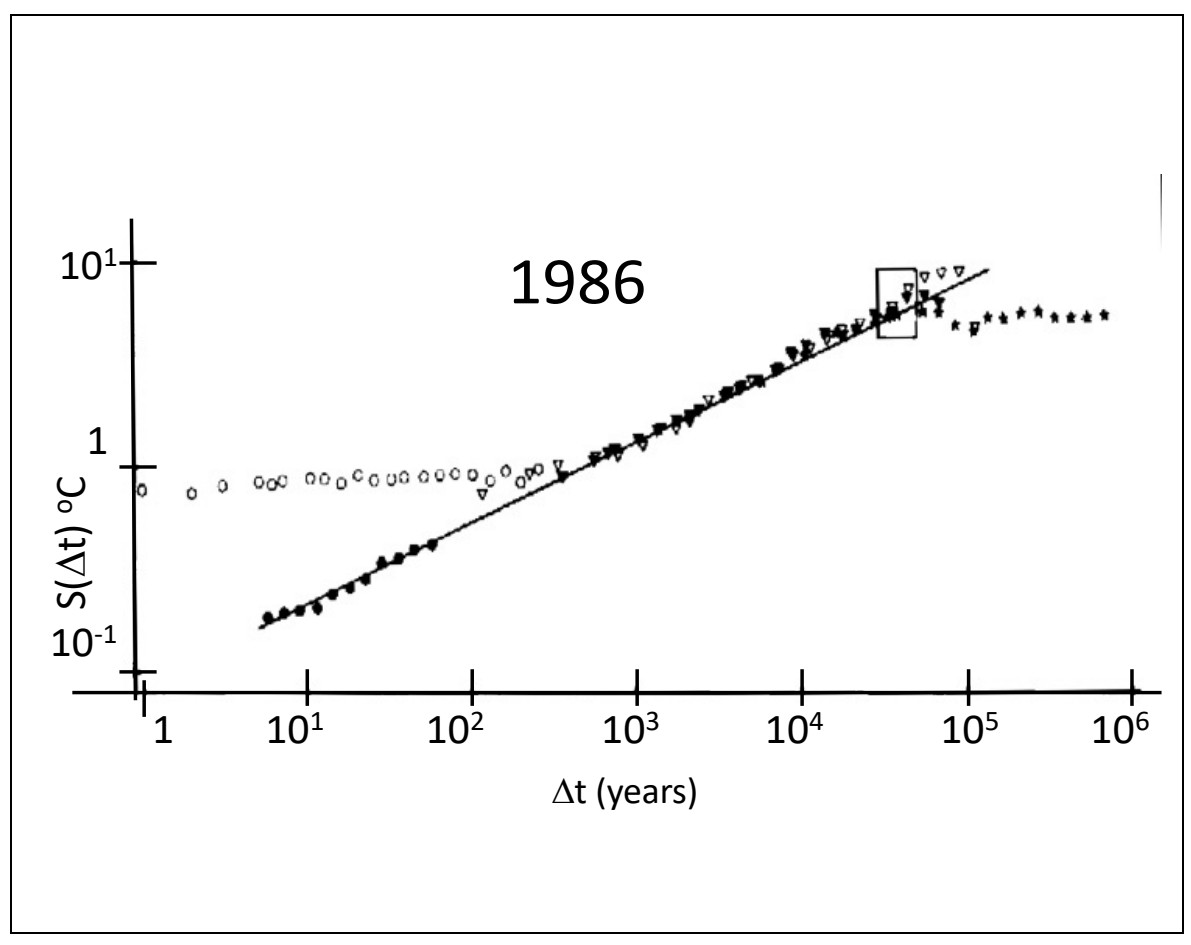




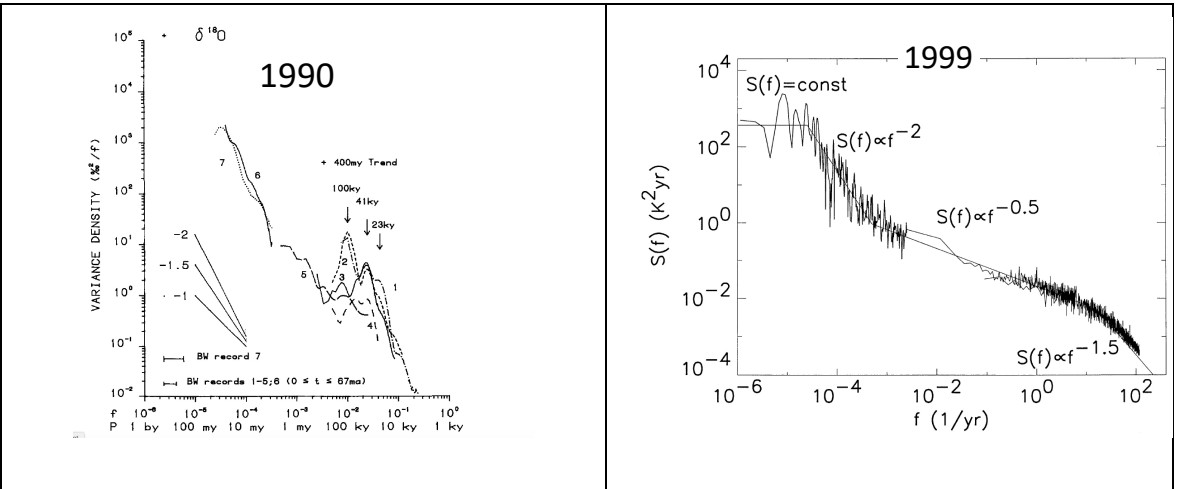

**Fig. 12: The evolution of the scaling picture, 1986 - 1999.**

**Top: The RMS difference structure functions estimated from local (Central England) temperatures since 1659 (open circles, upper left), northern hemisphere temperature (black circles), and from paleo temperatures from Vostok (Antarctic, solid triangles), Camp Century (Greenland, open triangles) and from an ocean core (asterixes). For the northern hemisphere temperatures, the (power law, linear on this plot) climate regime starts at about 10 years. The reference line has a slope $H = 0.4$. The rectangle (upper right) is the "glacial-interglacial window" through which the structure function must pass in order to account for typical variations of ±2 to ±3K for cycles with half periods ≈ 50 kyrs. Reproduced from [*Lovejoy and Schertzer*, 1986b].**

**Notice, the two essentially flat sets of points, one from the local central England temperate up to roughly three hundred years, and the other from an ocean core that is flat from scales 100,000 years and longer. These correspond to the macroweather and macroclimate regimes where $H<0$ so that the flatness is an artefact of the use of differences in the definition of fluctuations (appendix B2):**

**Bottom left: Composite spectrum of $\delta^{18}O$ paleo temperatures from [*Shackleton and Imbrie*, 1990].**

**Bottom right: Composite using instrumental temperatures (right) and paleotemperatures left) with piecewise linear (power law) reference lines. The composite is not very different from the more recent one shown in fig. 13, reproduced from [*Pelletier*, 1998].**

## 2.4 The impact of data on the scalebound view

In spite of its growing disconnect with modern data, Mitchell's figure and its scalebound updates continue to be influential. Yet, within fifteen years of Mitchell's famous paper, two scaling composites, over the ranges 1 *hr* to $10^5$ *yrs*, and

$10^3$ to $10^8$ *yrs*, already showed huge discrepancies [*Lovejoy and Schertzer*, 1986b], figure 12 (top) [*Shackleton and Imbrie*, 1990] (bottom left) see also [*Pelletier*, 1998] (bottom right) and [*Huybers and Curry*, 2006]. Returning to Mitchell's original figure, [*Lovejoy*, 2015] superposed the spectra of several modern instrumental and paleo series; the differences are literally astronomical (fig. 13). Whereas over the range 1 *hr* to $10^9$ *yrs*, Mitchell's background varies by a factor ≈ 150, the spectra from real data imply that the true range is a factor greater than a quadrillion ($10^{15}$).


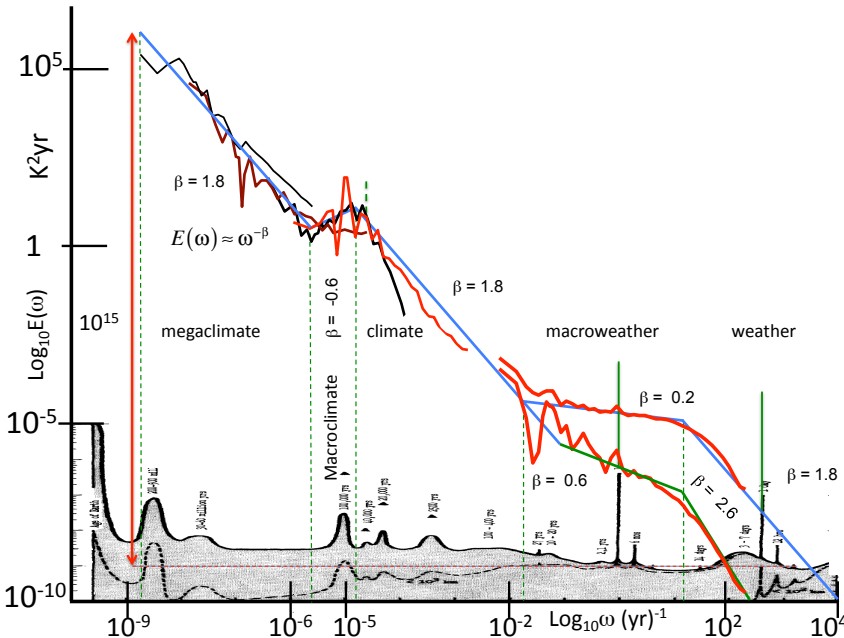

**Figure 13: A comparison of Mitchell's relative scale, "educated guess" of a log-log spectral plot (grey, bottom [Mitchell, 1976]) with modern evidence from spectra of a selection of the series described in table 1 and [Lovejoy, 2015] from which this figure is reproduced. On the far right, the spectra from the 1871-2008 20CR (at daily resolution) quantifies the difference between the globally averaged temperature (bottom right red line) and local averages (2ox2o, top right red line).**

The spectra were averaged over frequency intervals (10 per factor of ten in frequency), thus "smearing out" the daily and annual spectral "spikes". These spikes have been re-introduced without this averaging, and are indicated by green spikes above the red daily resolution curves. Using the daily resolution data, the annual cycle is a factor $\approx 1000$ above the continuum, whereas using hourly resolution data, the daily spike is a factor $\approx 3000$ above the background. Also shown is the other striking narrow spectral spike at $(41\ \text{kyrs})^{-1}$ (obliquity; $\approx$ a factor 10 above the continuum), this is shown in dashed green since it is only apparent over the period 0.8 - 2.56 Myr BP (before present).

The blue lines have slopes indicating the scaling behaviours. The thin dashed green lines show the transition periods that separate out the scaling regimes; these are (roughly) at 20 days, 50 yrs, 80,000 yrs, and 500,000 yrs. Reproduced from [Lovejoy, 2015].


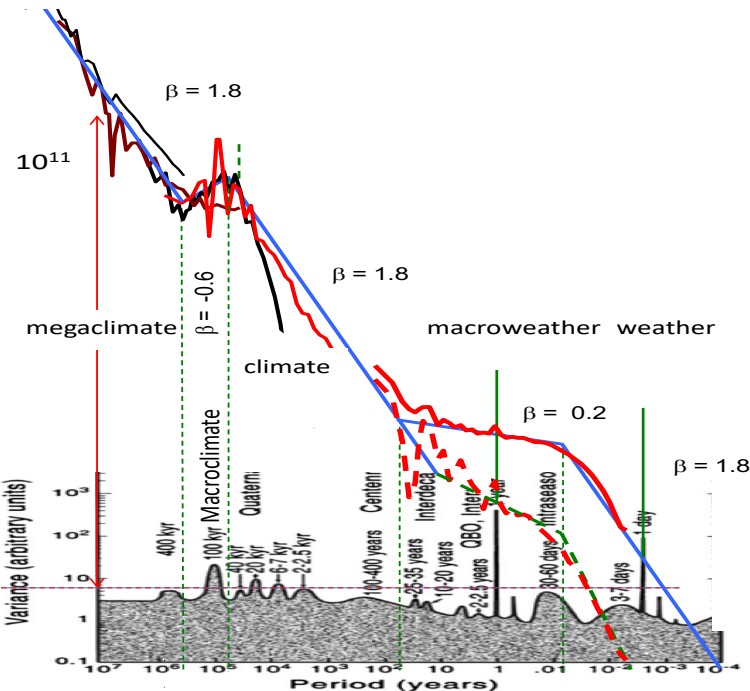

**540**   **Fig. 14 Artist's rendering with data superposed. Adapted from [*Ghil*, 2002], reprinted in [*Ghil and Lucarini*, 2020].**

Returning to the "Artist's Rendering", fig. 14 shows that when compared to the data, it fares no better than Mitchell's educated guess. The next update - NOAA's "mental model" - only specified that its vertical axis was proportional to "variability". If we interpret "variability" as the root mean square fluctuation at a given scale, and the flat "background" between the bumps

**545**   as white noise, then then we obtain the comparison fig. 15. Although the exact definition of these fluctuations is discussed in section 2.5, they give a directly physically meaningful quantification of the variability at a given time scale. In fig. 15, we see that the mental model predicts that successive million year average earth temperatures would differ by only tens of micro Kelvins! A closely similar conclusion would hold if we converted Mitchell's spectrum into RMS real space fluctuations.

The most recent scalebound update – the "conceptual landscape" - is compared with modern data in fig. 16. Although the

**550**   various scaling regimes proposed in [*Lovejoy*, 2013] (updated in fig. 18 and discussed below) are discretely indicated in the background, in many instances, there is no obvious relation between the regimes and the landscape. In particular, the word "macroweather" appears without any obvious connection to the figure, but even the landscape's highlighted scalebound features are not very close to the empirical curve (red). Although the vertical axis is only "relative", this quantitative empirical comparison was made by exploiting the equal area property mentioned above. The overlaid solid red curve was estimated by

**555**   converting the disjoint spectral power laws shown in the updated Mitchell graph (fig. 8). In addition, there is also an attempt to indicate the amplitudes of the narrow spectral spikes (the Green spikes in fig. 13) at diurnal, annual and – for the period 2.5





– 0.8 Myrs – the obliquity spectral peak at $(41 \text{kyrs})^{-1}$). In conclusion, the conceptual landscape bears little relation to the real world.

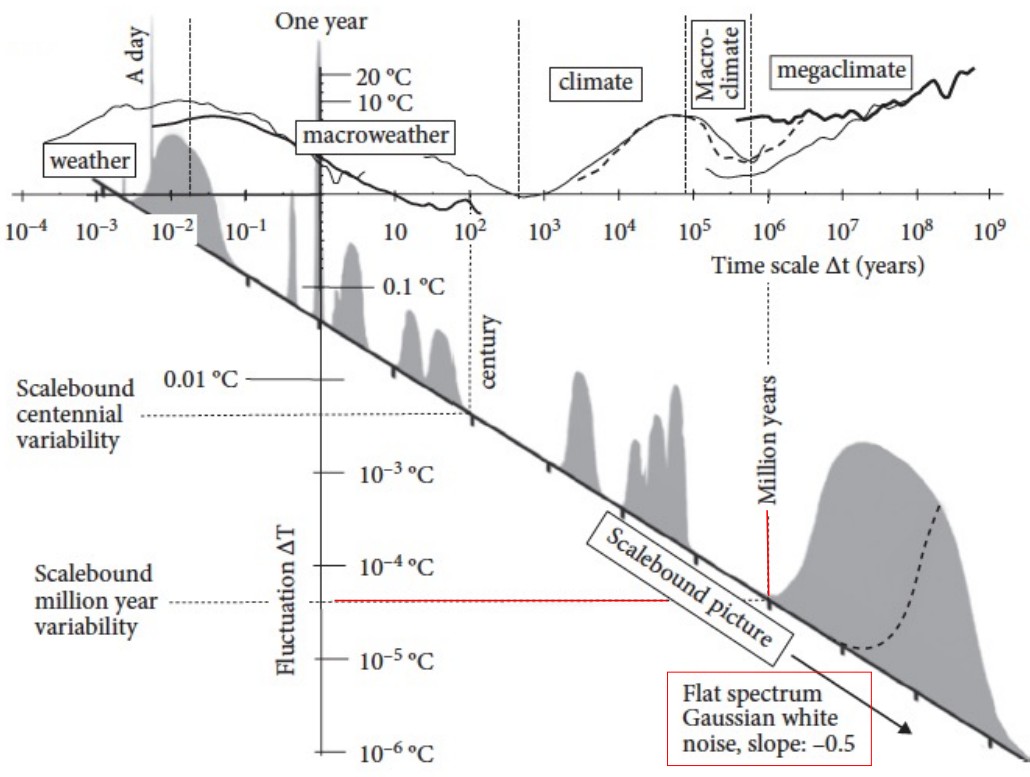

560

**Fig. 15: Mental model with data.** The data spectrum in fig. 13 is replotted in terms of fluctuations (grey, top, see fig. 17). The diagonal axis corresponds to the flat base line of fig. 6 (lower left) that now has a slope of -1/2 corresponding to an uncorrelated Gaussian "white noise" background. Since the amplitudes in fig. 6 (lower left) were not specified, the amplitudes of the transformed "bumps" are only notional. At the top is superposed the typical Haar fluctuations at time scale $\Delta t$ as estimated from various instrumental and paleo series, from fig. 17 (bottom, using the data displayed in fig. 2). We see (lower right) that consecutive 1 Myr averages would only differ by several μK. Reproduced from [Lovejoy 2019].

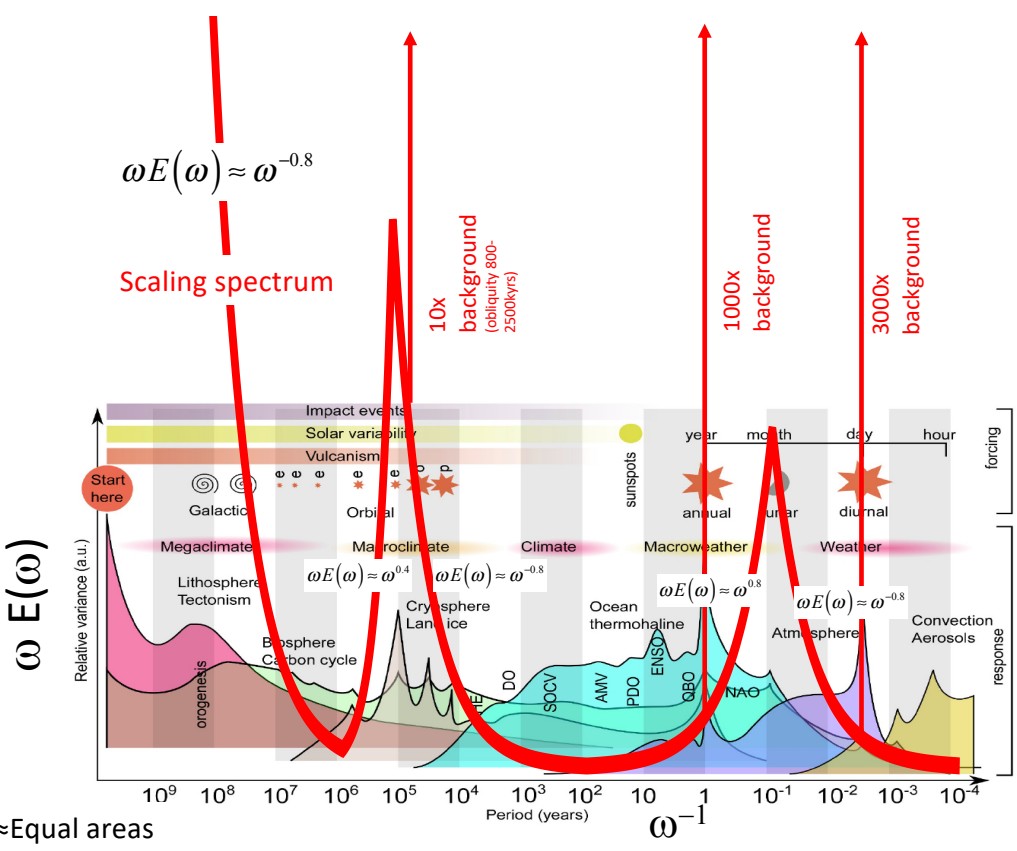

**Fig. 16: Conceptual landscape with data.** The superposed red curves use the data spectra in fig. 11 and adjusted the (linear) vertical scale for a rough match with the landscape. The vertical lines indicate huge periodic signals (the diurnal and annual cycles at the right and at the left, the obliquity signal seen in spectra between 0.8 and 2.5 Myrs ago. Adapted from [*von der Heydt et al.*, 2021].

### 2.5 Revisiting the atmosphere with the help of fluctuation analysis

The scalebound framework for atmospheric dynamics emphasized the importance of numerous processes occurring at well-defined time scales, the quasi periodic "foreground" processes illustrated as bumps – the signals - on Mitchell's nearly flat background. The point here is not that these processes, mechanisms are wrong or nonexistent, it is rather that they only explain a small fraction of the overall variability. This was also demonstrated quantitatively and explicitly over at least a significant part of the climate range by [*Wunsch*, 2003].

One of the lessons to be drawn from the educated guesses, artists' renderings, and conceptual landscapes is that although spectra can be calculated for any signal, the interpretations are often not obvious. The problem is that we have no intuition about the physical meaning of the units - $K^2s$, $K^2yr$, or even $K^2Myr$ - so that often (as here) the units used in spectral plots are not even given. It then becomes impossible to take data from disparate sources and at different time scales to make the spectral composites needed to make a meaningful check of the scalebound paradigm.





The advantage of fluctuations such as in fig. 12 (top) is that the numbers – for example the RMS temperature fluctuations at some scale - have a straightforward physical interpretations. However the differences used to define fluctuations (see fig 17, top) have a non-obvious problem: on average, differences cannot decrease with increasing time intervals (in appendix B, this problem is discussed more precisely in the Fourier domain). This is true for any series that has correlations that decrease with $\Delta t$ (as physically relevant series always do). A consequence is whenever the value of $\xi(2)$ is negative - implying that the mean fluctuations decrease with scale - that the differences fluctuations will at best give a constant result, the flat parts of Figure 12 (top).

But do regions of negative $\xi(2)$ exist? One way to investigate this is to try to infer the exponent $\xi(2)$ from the spectrum that does not suffer from an analogous restriction: its exponent $\beta$ can take any value. In this case we can use the formula $\beta = 1 + \xi(2)$ (Eq. 9). The latter implies that negative $\xi(2)$ corresponds to $\beta < 1$, and a check on the spectrum fig. 7 indicates that several regions (notably the macroweather regime) are indeed flat enough ($\beta < 1$) to imply negative $\xi(2)$. How do we fix the problem and estimate the correct $\xi(2)$ when it is negative?

It took a surprisingly long time to clarify this issue. To start with, in classical turbulence, $\xi(2) > 0$ (e.g. the Kolmogorov law), there was no motivation to look further than differences. Mathematically, the main advance came in the 1980's from wavelets. It turns out that technically, fluctuations defined as differences are indeed wavelets, but mathematicians mock them calling them the "poor man's wavelet" and they generally promote more sophisticated wavelets (see section Appendix B2): the simplicity of the physical interpretation is not their concern. This was the situation in the 1990's when scaling started to be systematically applied to geophysical time series involving negative $\xi(2)$ (i.e. to any macroweather series, although at the time this was not clear). A practical solution adopted by many was to use the Detrended Fluctuation Analysis (DFA) method [*Peng et al.*, 1994]. Unfortunately, but DFA fluctuations are difficult to interpret, so that typically only exponents are extracted: the important information contained in the fluctuation amplitudes not exploited (see appendix B).

New clarity was achieved with the help of the first "Haar" wavelet [Haar, 1910]. There were two reasons for this: the simplicity of its definition and calculation and the simplicity of its interpretation [*Lovejoy and Schertzer*, 2012b]. To determine the Haar fluctuation over a time interval $\Delta t$, one simply takes the average of the first half of the interval and from this, subtracts the average of the second half (Figure 17 bottom, see appendix B2 for more details). As for the interpretation, when $H$ is positive, then it is (nearly) the same as a difference, whereas whenever $H$ is negative, the fluctuation can be interpreted as an "anomaly" (in this context an anomaly is simply the average over a segment length $\Delta t$ of the series with its long term average removed, Appendix B2). In both cases we also recover the correct value of the exponent $H$. Although the Haar fluctuation is only useful for $H$ in the range -1 to 1, this turns out to cover most of the series that are encountered in geoscience.


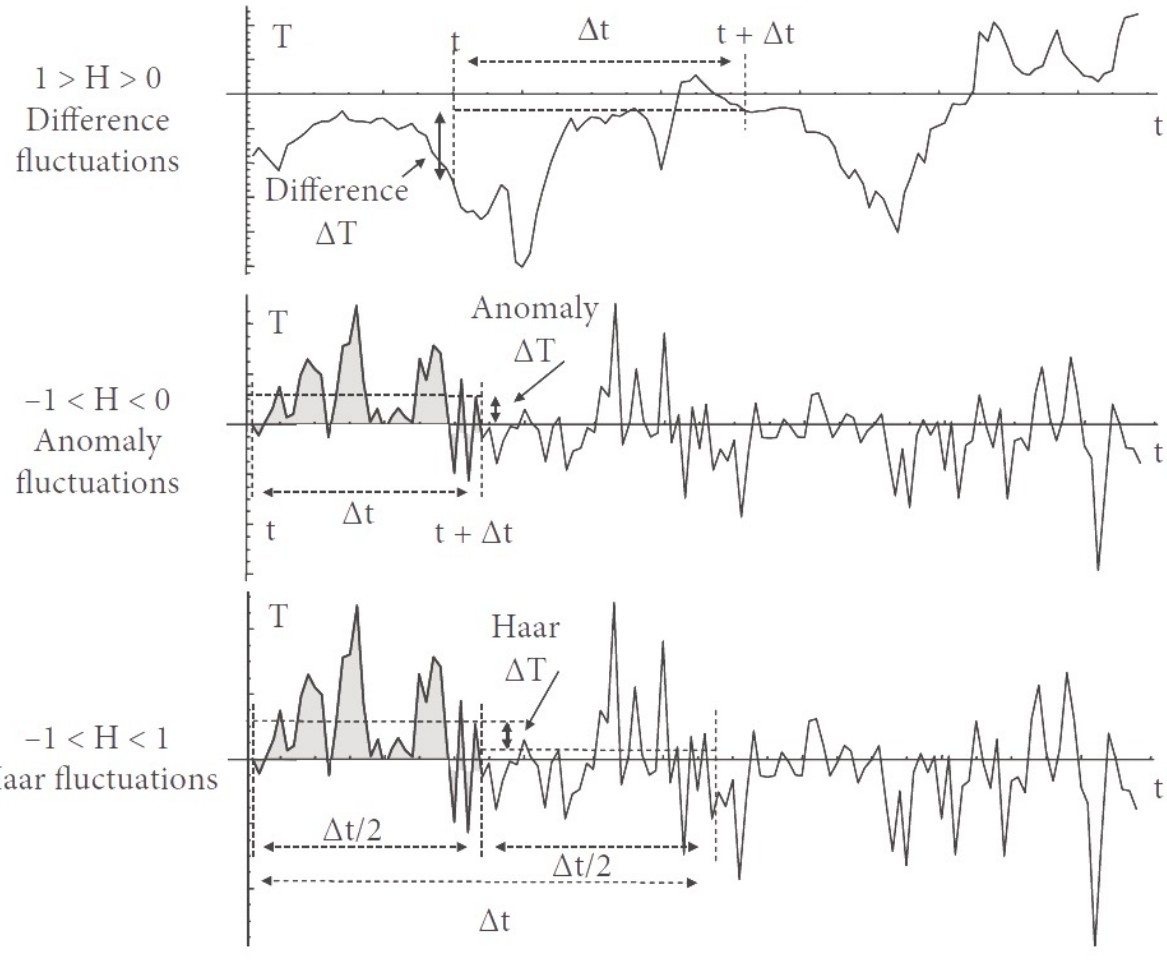

**Figure 17.** **Schematic illustration of difference (top) and anomaly (middle) fluctuations for a multifractal simulation of the atmosphere in the weather regime (0≤H≤1), top, and in the lower frequency macroweather regime (middle). Notice the wandering or drifting of the signal in the top figure and the cancelling behaviour in the middle. The bottom figure is a schematic illustration of Haar fluctuations (useful for processes with -1≤H≤1). The Haar fluctuation over the interval Δ*t* is the mean of the first half subtracted from the mean of the second half of the interval Δ*t*. Reproduced from [Lovejoy 2019].**

Figure 18 shows a modern composite using the root mean square Haar fluctuation, spanning a range of scales of $\approx 10^{17}$ (compare this with fig. 12 (top) for the earlier version using fluctuations as differences). The same five regimes as in fig. 13 are shown but now the typical variations in temperature over various time scales are very clear.

Also shown in fig. 16 are reference lines indicating the typical scale dependencies. These correspond to typical temperature fluctuations $\quad \Delta T \propto \Delta t^{\xi(2)/2} \approx \Delta t^{\xi(1)}$ where $\xi(1) = H$ is the "fluctuation exponent" (the exponent of the mean absolute fluctuation, the relationship $\xi(2) = 2H$ is valid if we ignore intermittency, it is the quasi-Gaussian relationship still often





invoked, see eq. 15). In the figure, we see that the character of the regimes alternates between regimes that grow ($H>0$) and ones that decrease ($H<0$) with time scale. The sign of $H$ has a fundamental significance; to see this, we can return to typical series over the various regimes, fig. 2, (left hand column, 3). In terms of their visual appearances, the $H>0$ regimes have signals that seem to "wander" or"drift" whereas for $H<0$ regimes fluctuations tend to cancel. In the former, waiting longer and longer typically leads to larger changes in temperature, whereas in the latter, longer and longer temporal scales leads to

convergence to well defined values.

With the help of the figure, we can now understand the problem with the usual definition of climate as "long term" weather. As we average from ten days to longer durations, temperature fluctuations do indeed tend to diminish – as expected if they converged to the climate. Consider for example the thick solid line in fig. 18 (corresponding to data at 75°N), that shows that at about 10 days, the temperature fluctuations are $\approx \pm 3\text{K}$ ($S_2(\Delta t)^{1/2} \approx 6\text{K}$) diminishing at 20 yrs to $\approx \pm 0.3\text{K}$. Since $H<0$, the

Haar fluctuations are nearly equivalent to the anomalies i.e. to averages of the series with the long time mean removed. Over this range, increasing the scale leads to smaller and smaller fluctuations about the point of apparent point of convergence: the average "climate" temperature. Fig. 18 also shows the longer scales deduced purely from paleodata (isotope ratios from either ice or cores).

The interpretation of the apparent point of convergence as the climate state is supported by the analysis of global data compared

with GCMs in"control runs"( i.e. with fixed external conditions, fig. 19). When averaged over long enough times, the control runs do indeed converge although the convergence is"ultra slow" (at a rate characterized by the exponent $H \approx -0.15$ for the GCMs). Extrapolating from the figure, shows that even after averaging over a million simulated years the GCMs would still typically be only within $\pm 0.01\text{K}$ of their respective climates.



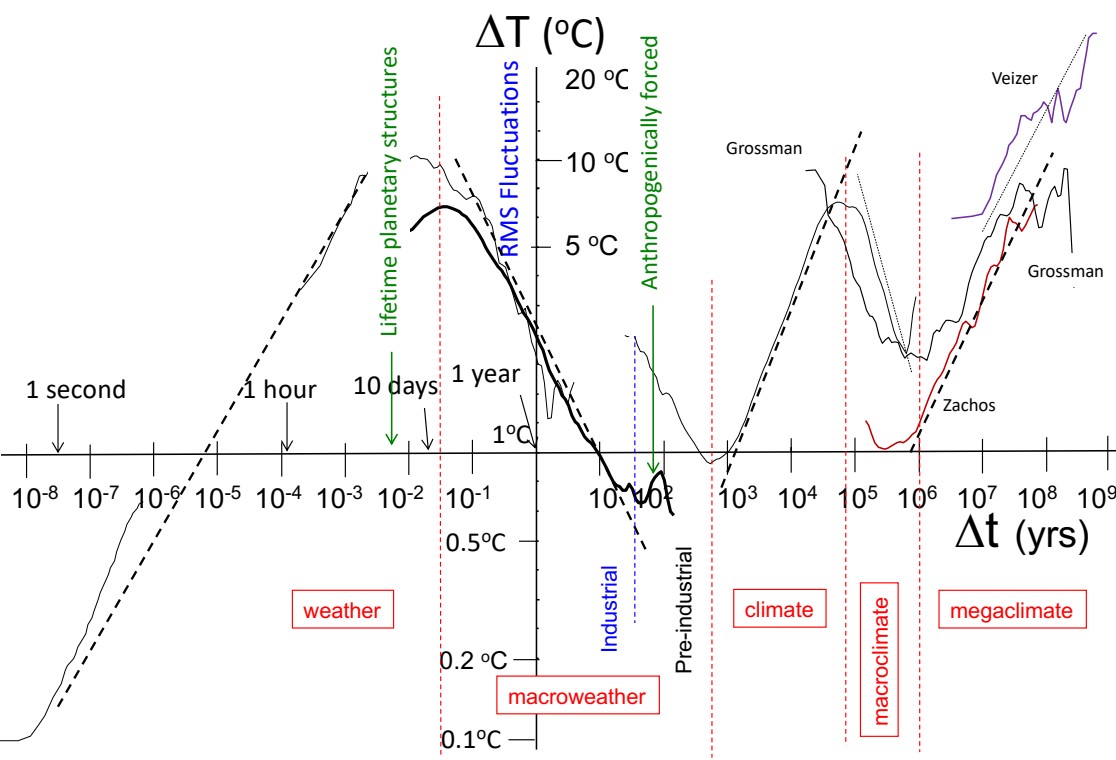

**Fig. 18:** **The broad sweep of atmospheric variability with root mean square (RMS) Haar fluctuations showing the various (roughly power law) atmospheric regimes, adapted and updated from the original [*Lovejoy*, 2013] and the update in Lovejoy (2015) where the full details of the data sources is given (with the exception of the paleo analysis marked "Grossman" which is from [*Grossman and Joachimski*, 2022]). The dashed vertical lines show the rough divisions between regimes; the macroweather-climate transition is different in the pre-industrial epoch. The high frequency analysis (lower left) from thermistor data taken at McGill at 15Hz was added. The thin curve starting at 2 hours is from a weather station, the next (thick) curve is from the 20th Century reanalysis, the next, "S" shaped curve is from the EPICA core. Finally, the three far right curves are benthic paleo temperatures (from "stacks"). The quadrillion estimate is for the spectrum, it depends somewhat on the calibration of the stacks. With the calibration in the figure, the typical variation of consecutive 50 million year averages is ±4.5K ($\Delta t = 10^8$ years, RMS $\Delta T = 9$K). If the calibration is lowered by a factor of ≈3 (to variations of ±1.5K), then the spectrum would be reduced by a factor of 32. On the other hand, the addition of the 0.017s resolution thermistor data increases the overall spectral range by another factor of $10^8$ for a total spectral range of a factor ≈$10^{17}$ for scales from 0.017s to 5x$10^8$ years.**





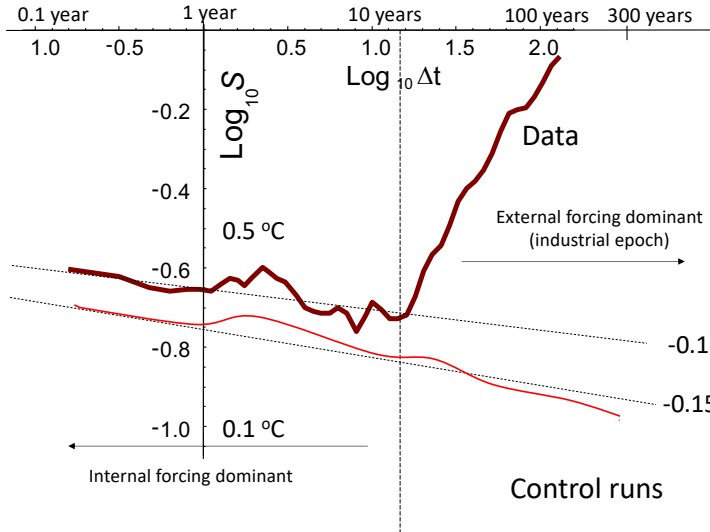

**Fig. 19: Top (brown): The globally averaged, RMS Haar temperature fluctuations averaged over three data sets (adapted from**
**where there are the full details, the curve for the corresponding time scale in fig. 19 is at 75ºN, it is a bit different). At small time scales, one can see reasonable power law behaviour with H ≈ -0.1. However for scales longer than about 15 years, the externally forced variability becomes dominant. Although in reality, the internal variability continues to larger scales, and the externally forced to variability to smaller ones. The two can roughly be separated at decadal scales as indicated by the vertical dashed line.**

**The curve is reproduced from [*Lovejoy*, 2017a]. Bottom: (red), the RMS Haar fluctuations for 11 control runs from the Climate**
**Model Intercomparison Project 5 (CMIP5). The reference slope *H* = -0.15, adapted from [*Lovejoy*, 2019].**

Returning to fig. 18 however, we see that beyond a critical time scale $\tau_c$, the convergence is reversed and fluctuations tend to

rather to increase with time scale. In the anthropocene (roughly since 1900), the ≈ 15 year time scale where fluctuations stop

decreasing and begin increasing with scale, is roughly the time that it has taken for anthropogenic warming (over the last

decades) to become comparable to the natural internal variability (about ±0.2K for these globally averaged temperatures).

However, for the preindustrial epoch (see the "S" shaped paleo-temperature curve from the EPICA ice core, fig. 18), the

transition time is closer to 300 years. The origin of this larger $\tau_c$ value is not clear, it is a focus of the PAGES - CVAS working

group [*Lovejoy*, 2017b].

As concerns the last 800kyrs, the key point about Fig. 18 is that we have three regimes – not two. Since the intermediate

regime is well produced by control runs, (fig. 19), it is termed "macroweather": it is essentially averaged weather.

If the macroweather regime is characterized by slow *convergence* of averages with scale, it is logical to define a climate state

as an average over durations that are long enough so that the maximum convergence has occurred – i.e. over periods $\Delta t > \tau_c$. In

the anthropocene, this gives an objective justification for the official World Meteorological Organization's otherwise arbitrary

climate averaging period of 30 years. Similarly, the roughly ten days to a month weather-macroweather transition at $\tau_w$ - gives



an objective justification for the common practice of using monthly average anomalies: these define analogous macroweather states. The climate regime is therefore the regime beyond $\tau_c$ where the climate state itself starts to vary. In addition to the analyses presented here, there are numerous papers claiming evidence for power law climate regimes: [*Lovejoy and Schertzer*, 1986b], [*Shackleton and Imbrie*, 1990], [*Schmitt et al.*, 1995], [*Ditlevsen et al.*, 1996], [*Pelletier*, 1998], [*Ashkenazy et al.*, 2003], [*Wunsch*, 2003], [*Huybers and Curry*, 2006], for a more comprehensive review, see discussion and table 11.4 in

[*Lovejoy and Schertzer*, 2013].

Again from fig. 18, we see that the climate state itself starts to vary in a roughly scaling way up until Milankovitch time scales (at about 50kyrs; half the period of the main 100kyr eccentricity frequency) over which fluctuations are typically of the order ±2 to ±4K: the glacial-interglacial "window" over typical variability that is quite clear in the figure (c.f. the most recent estimate is a total range of 6 K or ±3, [*Tierney et al.*, 2020]). At even larger scales there is evidence (from ice core and benthic paleodata,

notably updated with a much improved mega-climate series by [*Grossman and Joachimski*, 2022], bold curve at the right), that there is a narrow macroclimate regime and then a wide range megaclimate regime, but these are outside our present scope (see [*Lovejoy*, 2015] for more discussion).

**2.6: Lagrangian space-time relations, Stommel diagrams and the weather-macroweather transition time**

**2.6.1 Space-time scaling from the anisotropic Kolmogorov law**

Space-time diagrams, are log time - log space plots for the ocean ([Stommel, 1963], fig. 20 left) and atmosphere ([Orlanski, 1975], fig. 20 right). They highlight the conventional morphologies, structures and processes typically indicated by boxes or ellipses in the space-time regions in which they have been observed. Since the diagrams refer to the lifetimes of structure co-moving with the fluid, these are Lagrangian space-time relations. The Eulerian (fixed frame) relations are discussed in the

next section.

A striking feature of these diagrams - especially in Orlanski's atmospheric version (fig. 20 right) but also in the updates (fig. 21), is the near linear - i.e. power law - arrangement of the features. As pointed out in [*Schertzer et al.*, 1997a] in the case of Orlanski's diagram, the slope of the line is very close to the theoretically predicted value 3/2. This is the value that holds if the atmosphere respects (anisotropic) Kolmogorov scaling in the horizontal: $\Delta v(l) \approx \varepsilon^{1/3} l^{1/3}$ where $\varepsilon$ is the power per mass, $l$ is

the horizontal length scale and $\Delta v(l)$ is the typical velocity difference across a structure of size $l$. In the scaling "inertial" range where this relationship holds - if only on dimensional grounds - the lifetime $\tau$ of a structure is given by $\tau = l \, \Delta v(l)$. This implies the lifetime - size relation:

$$\tau = \varepsilon^{-1/3} \, l^{2/3} \tag{10}$$





In isotropic turbulence, this is a classical result, yet it was first applied to the anisotropic Kolmogorov law (and hence up to

planetary scales) in [*Schertzer et al.*, 1997a]. Eq. 10 predicts both the exponent (the log-log slope), and – if we know ε - the

prefactor (figs. 20 - 22, see [Lovejoy et al., 2001]).

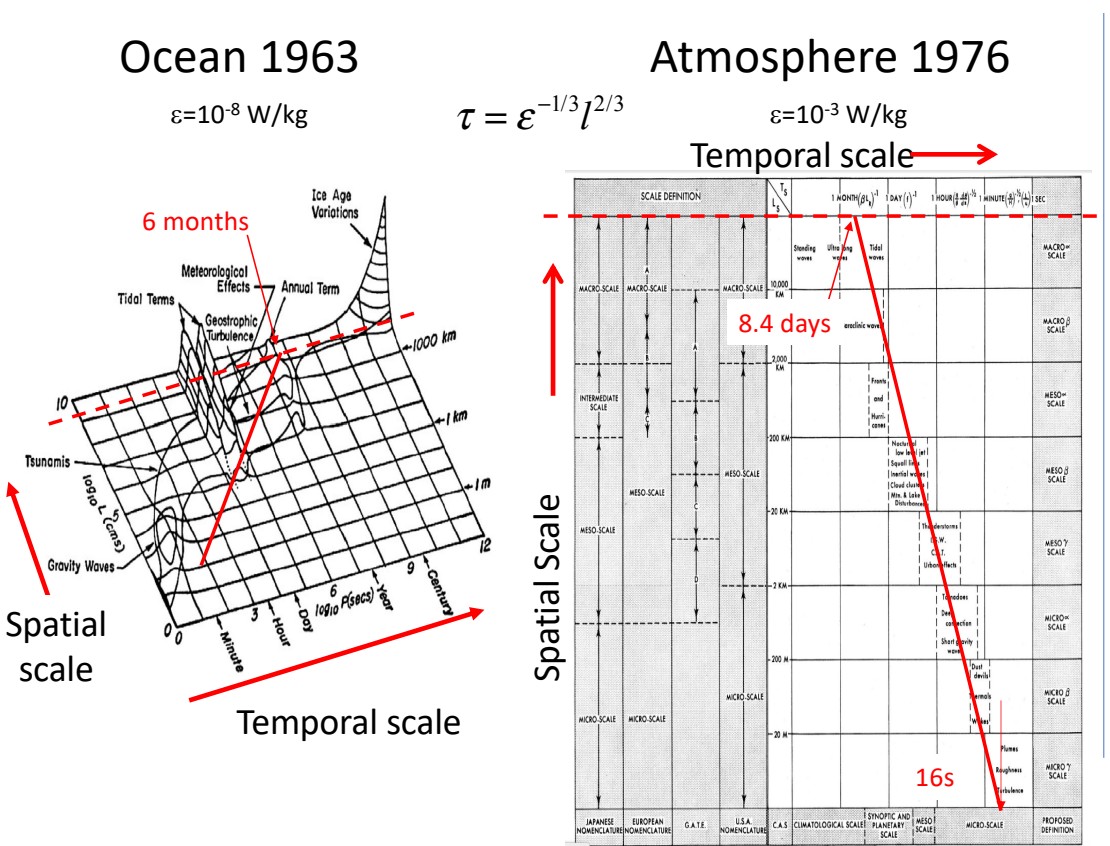


**Fig. 20: The original space-time diagrams ([*Stommel*, 1963] ocean, left), [*Orlanski*, 1975], the atmosphere, right). The solid red lines are theoretical lines assuming the horizontal Kolmogorov scaling with the measured mean energy rate densities indicated. The dashed red lines indicate the size of the planet (the half circumference, 20000km). Where the time scale at which they meet is the lifetime of planetary structures (≈ 10 days in the atmosphere, about 6 months in the ocean). It is equal to the weather-macroweather**

**and "ocean -weather" to "ocean macroweather" transition scales and it is also close to the corresponding deterministic predictability limits.**



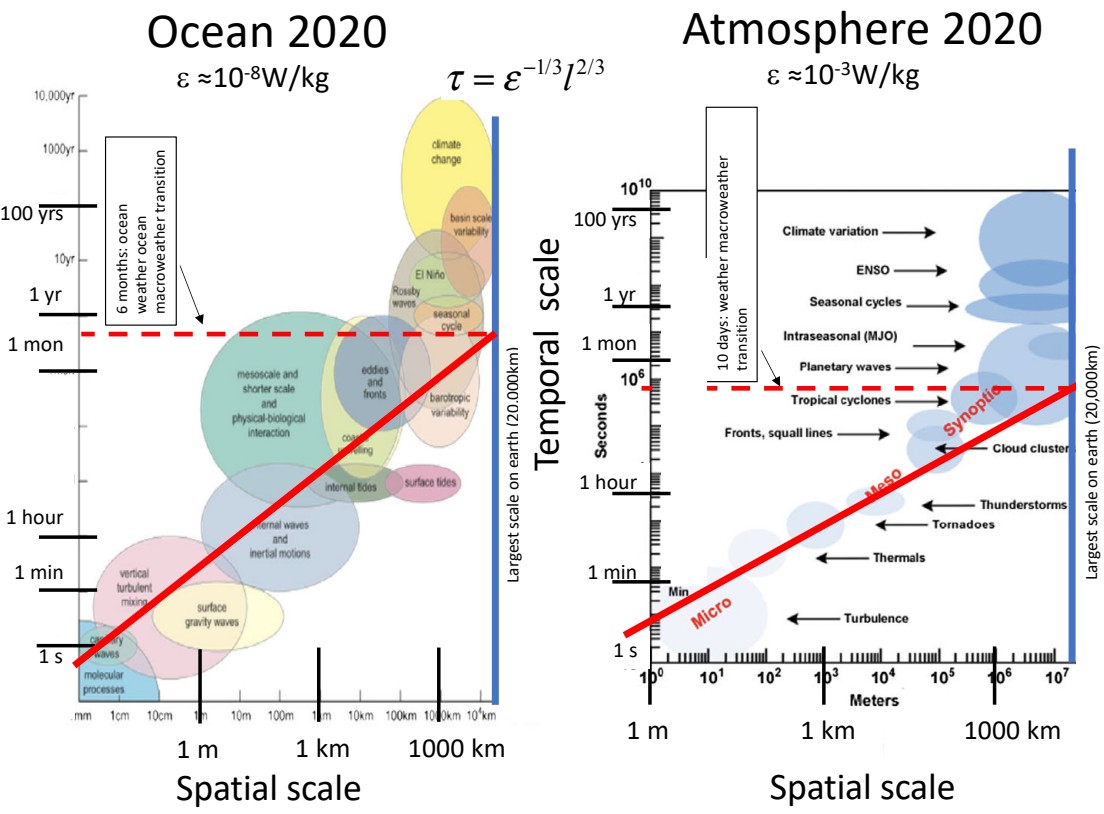

**Fig. 21: The basic figures are space-time diagrams for the ocean (left) and atmosphere (right), from [*Ghil and Lucarini*, 2020], note that space and time have been swapped as compared to fig. 20. As in fig. 20, solid red line has been added showing the purely theoretical predictions. At the right, a solid blue line was added showing the planetary scale. The dashed red line (also added) shows the corresponding lifetimes of planetary structures (the same as in fig. 20). We see once again that wide range horizontal Kolmogorov scaling is compatible with the phenomenology, especially when take into account the statistical variability of the space-time relationship itself, as indicated in fig. 22.**




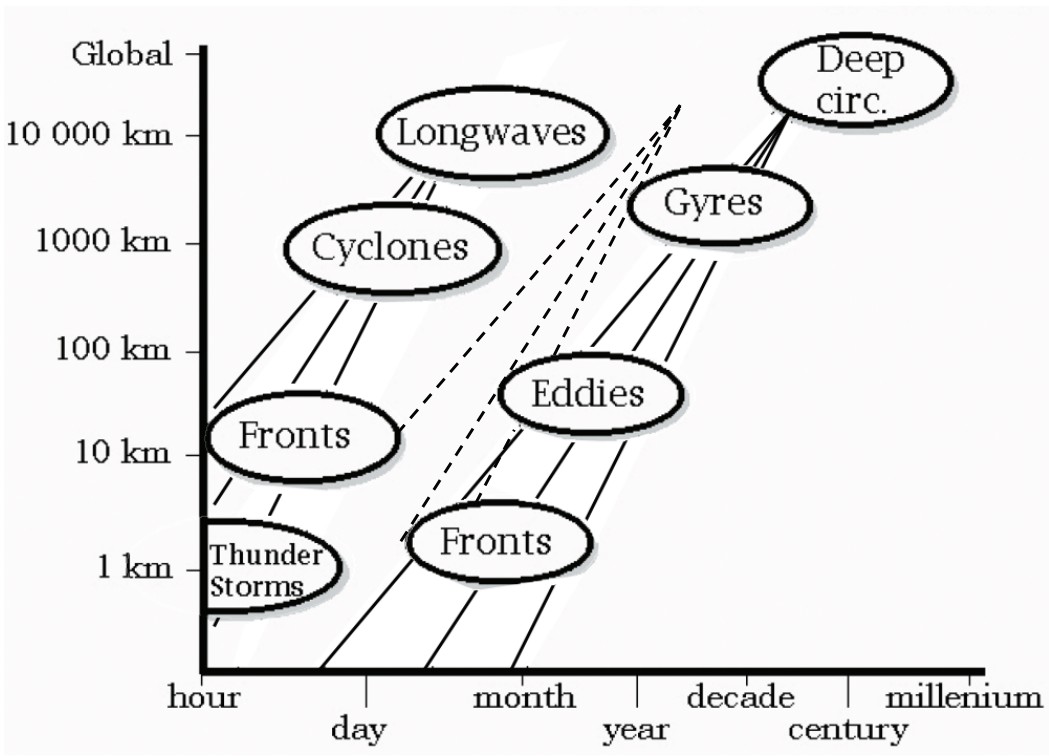

**Fig. 22 A space-time diagram showing the effects of intermittency and for the oceans, the deep currents associated with very low ε. The original was published in [*Steele*, 1995] with solid reference lines added in [*Lovejoy et al.*, 2001] and the dashed lines were added in a further update in [*Lovejoy and Schertzer*, 2013]. The central black lines indicate the mean theory (i.e. $\tau \approx \varepsilon^{-1/3} l^{2/3}$ with $\varepsilon = 10^{-3}$ W/kg left, $\varepsilon = 10^{-12}$ (right, appropriate for deep water). The central dashed lines represent $\varepsilon = 10^{-8}$ W/kg. The lines to the left and right of the central lines represent the effects of intermittency with exponent $C_1 = 0.25$ ( slopes $3/(2 \pm C_1) \approx 0.75, 0.59$), see section 3, this corresponds to roughly one standard deviation variation of the singularities in the velocity field) reproduced from [*Lovejoy and Schertzer*, 2013].**

**2.6.2 The atmosphere as a heat engine: Space-time scaling and the weather-macroweather transition scale**

Thinking of the atmosphere as a heat engine that converts solar energy into mechanical energy (wind), allows us to estimate ε directly from first principles. Taking into account the average albedo and averaging over day, night and over the surface of the globe, we find that solar heating is $\approx 238$ W/m². The mass of the atmosphere is $\approx 10^4$ kg/m² so that the heat engine operates with a total power of 23.8 mW/kg. However, heat engines are never 100% efficient and various thermodynamic models (e.g. [*Laliberté et al.*, 2015]) predict efficiencies of a few percent. For example, an engine at about 300K that operates over a range of 12K, has a Carnot efficiency of 4%.


On earth, direct estimates of ε from wind gradients (using $\varepsilon = \Delta v^3 / \Delta x$) find large scale average values of $\approx$ 1 mW/kg implying an efficiency of (1 mW/kg) /(23.8 mW/kg) $\approx$ 4%, confirming the theory [*Lovejoy and Schertzer*, 2010b] (the values 1 mW/kg, 23.8 mW/kg are for global averages, there are systematic latitudinal variations, fig. 23 confirms that the theory works well at each latitude).

Using the value ε $\approx$ 1mW/kg and the global length scale $L_e$ gives the maximum lifetime τ $\approx$ 10 days (this is where the lines in the Stommel diagrams intersection the earth scales in the atmospheric Stommel diagrams). For the surface ocean currents, as reviewed in ch. 8 of [*Lovejoy and Schertzer*, 2013], ocean drifter estimates yield ε $\approx$ $10^{-8}$ W/kg, implying a maximum ocean gyre lifetime of about a 1 year. Deep ocean currents have much smaller values ε $\approx$ $10^{-12}$ - $10^{-15}$ W/kg (or less) that explain the right hand side of the Stommel diagram fig. 22. This diagram indicates these values with the theoretical slope 3/2 well fit the

phenomenology. The figure also shows the effect of intermittency (section 3.3) that implies a statistical distribution about the exponent 3/2 (this is simply the exponent of the mean), the width of which is also theoretically estimated and shown in the plot, thus potentially explaining the statistical variations around the mean behaviour.

In space, up to planetary scales, the basic wind statistics are controlled by ε, hence, up to $\tau_w$, it also determines the corresponding temporal statistics. Beyond this time scale, we are considering the statistics of many planetary scale structures.

That the problem becomes a statistical one is clear since the lifetime in this anisotropic 23/9D turbulence is essentially the same as its predictability limit, the error doubling time for the $l$ - sized eddies (e.g. ch. 2 of [*Lovejoy and Schertzer*, 2013]). If the atmosphere had been perfectly flat (or "layerwise flat" as in quasi-geostrophic 2-D turbulence) –then its predictability limit would have been much longer. Therefore at this transition scale, even otherwise deterministic GCMs become effectively stochastic. Since the longer time scales are essentially large scale weather, it has been dubbed "macroweather".

Fig. 24 shows atmospheric and oceanic spectra clearly showing the weather macroweather transition and ocean weather - ocean macroweather transitions at the theoretically calculated time scales. It also shows the only other known weather-macroweather transition, this time on Mars using Viking lander data. The Martian transition time may be theoretically determined by using the Martian value ε $\approx$ 40mW/Kg, and a 4% Carnot efficiency. Using the Martian solar insolation and atmospheric mass, the theory predicts a Martian weather/ macroweather transition at about 1.8 sols (1 sol $\approx$ 25 hours), a

prediction confirmed in fig. 24 [Lovejoy et al., 2014].



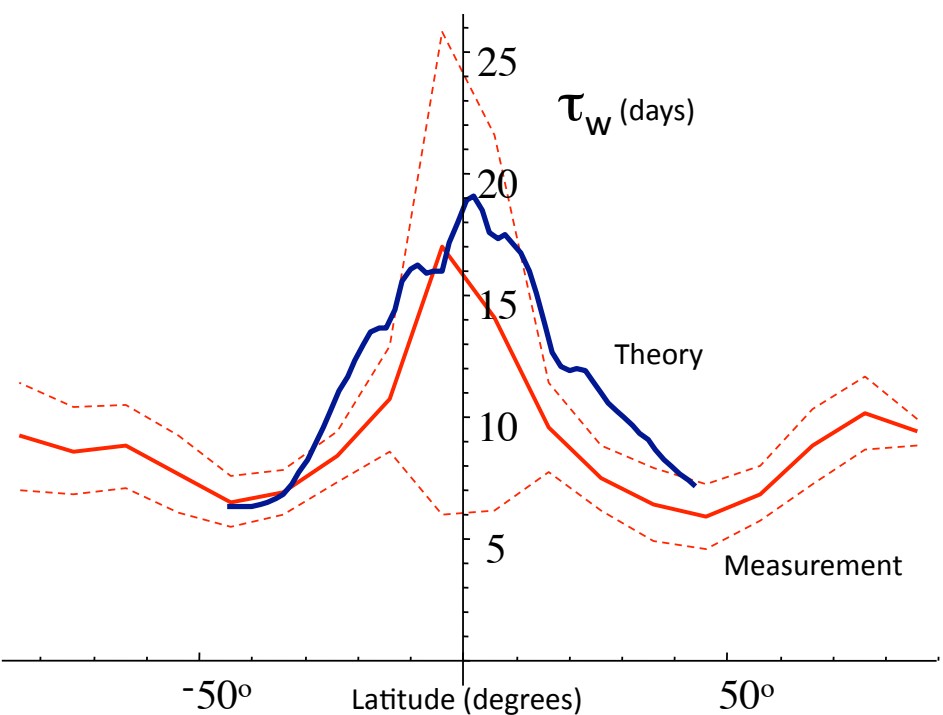

**Fig. 23: The weather-macroweather transition time $\tau_w$ as a function of latitude. Blue shows the theoretical curve ($\pm 45°$ latitude only)**
**estimated from the horizontal wind field at 700 mb (blue, using $\varepsilon = \left\langle \Delta v^3 / \Delta x \right\rangle$, $\tau_w \approx \varepsilon^{-1/3} L^{2/3}$, data from the ECMWF**
**reanalysis), and (red) direct estimates from breaks in the spectra of 700 mb temperature series at 2° resolution from the 20th century**
**reanalysis (the solid line is the mean, the dashed lines are the one standard deviation spread along each latitude line). Adapted from**
**[*Lovejoy and Schertzer*, 2013].**

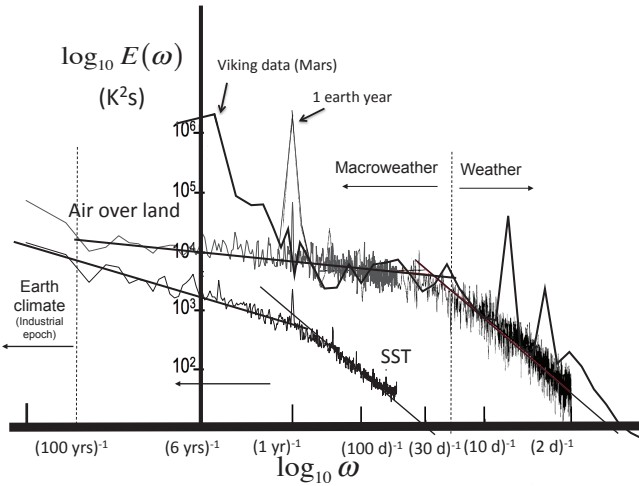


**Fig. 24: The three known weather - macroweather transitions: air over the Earth (black and upper left grey), the Sea Surface Temperature (SST, ocean) at 5° resolution (lower thin black) and air over Mars (thick, solid, black). The air over earth curve is from 30 years of daily data from a French station (Macon, black) and from air temps for last 100 years (5°x5° resolution NOAA NCDC), the spectrum of monthly averaged SST is from the same data base. The Mars spectra are from Viking lander data. The**
**strong "spikes" at the right are the Martian diurnal cycle and its harmonics. At the far left, the spectral rise (Earth) is low frequency response to anthropogenic forcing. Reproduced from [*Lovejoy*, 2019].**

### 2.7 Eulerian space-time relations

In the previous section, we discussed the space-time relations of structures of size $l$ that maintained their identities over a lifetime $\tau$ - these space time diagrams are Lagrangian. Unsurprisingly, it turns out to be much simpler to empirically check
the corresponding fixed-frame Eulerian relations, we consider this in this sub-section.

The key difference between the Eulerian and Lagrangian statistics is that the former involves an overall mean advection velocity V. When studying laboratory turbulence, [*Taylor*, 1938] proposed that the turbulence is "frozen" such that the pattern of turbulence blows past the measuring point sufficiently fast so that it does not have time to evolve. Frozen turbulence requires the existence of a scale separation between small and large scales so that the large (nearly "frozen") structures really
do blow the small ones structures without much evolution. When this approximation is valid, the spatial statistics may be obtained from time series by the deterministic transformation $V\Delta t \rightarrow \Delta x$ where $V$ is a constant.

In Taylor's laboratory turbulence, $V$ is determined by the fan and by the wind-tunnel geometry and within limits, the approximation is valid. However, although the transformation has been frequently used to interpret meteorological data, due to the horizontal scaling, there is no scale separation between large and small scales so that atmospheric tubulence is not frozen.
However, we are only interested in the statistical relations between time and space, and if the system is scaling, then advection can be taken into account using the Gallilean transformation $r \rightarrow r\text{-}Vt$, $t \rightarrow t$. Since V is now a random (turbulent) velocity, its effects must then be averaged, this is discussed in section 4.1.5. The full theory of space-time scaling requires the consideration




of anisotropic space-time and was developed in [*Schertzer et al.*, 1997a], [*Lovejoy et al.*, 2008b], [*Pinel et al.*, 2014] and reviewed in [*Lovejoy and Schertzer*, 2013].

In order to test the space-time scaling on real world data, the best sources are remotely sensed data such as the space-time lidar data  discussed in [*Radkevitch et al.*, 2008] or the global scale data from geostationary satellites in the IR whose spectra are shown in fig. 25 [*Pinel et al.*, 2014].  The figure uses 1440 consecutive hourly images at 5km resolution over the region 30N to 30S and 80E to 200E.   A full analysis based on the 3D (*x,y,t*) space data is given in [*Pinel et al.*, 2014] the figure shows only 1-D subspaces (EW, NS and time).

There are two remarkable aspects of the figure. The first is that in spite of an apparently slight curvature (normally a symptom of deviations from perfect scaling) it is in reality largely a "finite size effect" on otherwise excellent scaling. This can be seen by comparison with the black curve that shows the consequences of the averaging over the (roughly rectangular) geometry of the observing region combined with the "trivial" anisotropy of the spectrum (the matrix $\underline{C}$ is eq. 11).  (This is clearly visible in the various subspaces *(x, y), (x, t), (y,t)*, analyzed in [*Pinel et al.*, 2014] where the full theory and analysis is given).

Comparing the spectra to the theoretical black curve, we see that there are only small deviations from scaling, and this holds over the range in space from 60 km to $\approx$ 5000km (space) and from $\approx$ 2 hours to $\approx$ 7 days.

 The second remarkable aspect is the near perfect superposition of the 1-D spectra $E(\omega), E(k_x), E(k_y)$ over the same range

(obtained by successively integrating out the conjugate variables (e.g. $E(\omega) = \int P(k_x, k_y, \omega) dk_x \, dk_y$ ).   Writing

$\underline{K} = (k_x, k_y, \omega)$, the overall result is that the full 3D, horizontal space-time spectrum $P(\underline{K})$ respects the symmetry:

$P(\lambda^{-1} \underline{K}) = \lambda^s P(\underline{K})$, (see section 4.1.5 For the theory).  The full relationship between the Lagrangian and Eulerian statistics is derived in full in [*Pinel et al.*, 2014] and ch. 8 of [*Lovejoy and Schertzer*, 2013].  By averaging over the advection is turbulent advection the final theoretical  result is:

$$P(\underline{K}) \propto \left| \underline{\underline{C}}^{-1} \underline{K} \right|^{-s}; \quad \underline{\underline{C}} = \begin{pmatrix} 1 & 0 & -\mu_x \\ 0 & 1 & -\mu_y \\ 0 & 0 & \sqrt{1 - (\mu_x^2 + a^2 \mu_y^2)} \end{pmatrix} \tag{11}$$

Where the nondimensional $\mu_x$, $\mu_y$ are related to the average zonal, meridional advection velocities and their variances, and *a*

is the average "trivial" zonal to meridional aspect ratio. Empirically $s = 3.4$, $a \approx 1.6$. Eq. 11 implies that the 1-D spectra in fig. 25 respect $E(\omega) \propto E(k_x) \propto E(k_y)$ as shown.
Given the space-time scaling, one can use the real space statistics to define Eulerian space-time diagrams. Using the same data, this is shown in fig. 26. where we see that the relationship is nearly linear on a linear-linear plot (i.e. with a constant velocity) up to about 10 days, corresponding to near planetary scales as indicated in the figure. Note some minor differences between EW and NS directions.

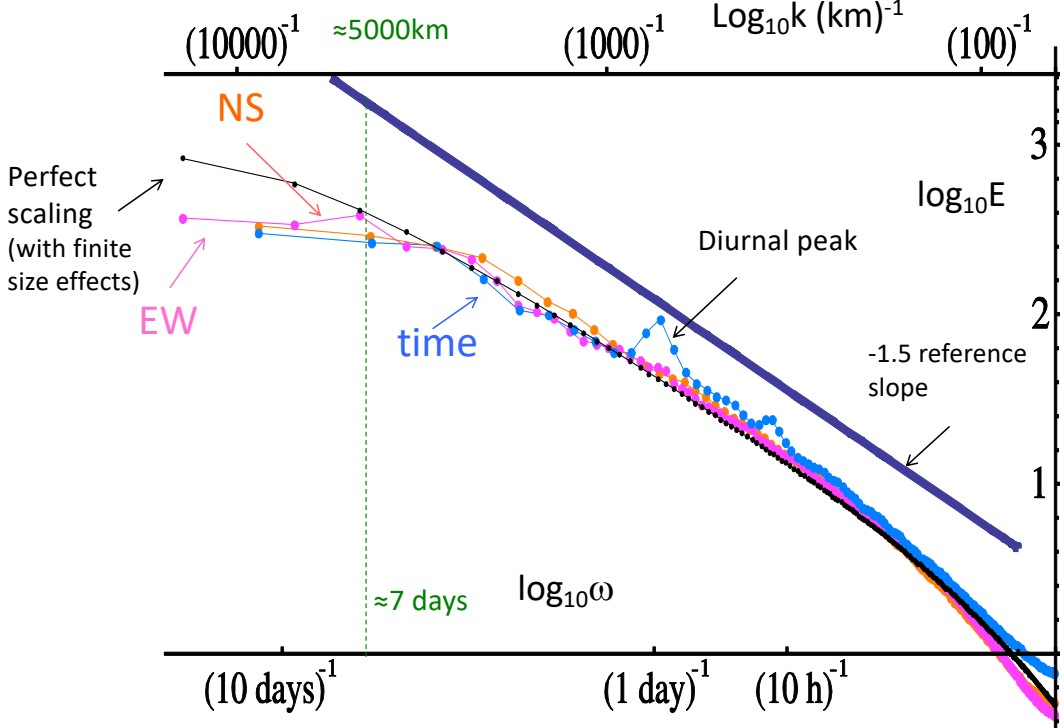

**Fig. 25: The zonal, meridional and temporal spectra of 1386 images (~ two months of data, September and October 2007) of radiances fields measured by a thermal infrared channel (10.3-11.3 μm) on the geostationary satellite MTSAT over south-west Pacific at resolutions 30 km and 1 hr over latitudes 40°S – 30°N and longitudes 80°E – 200°E. With the exception of the (small) diurnal peak (and harmonics), the rescaled spectra are nearly identical and are also nearly perfectly scaling (the black line shows exact power law scaling after taking into account the finite image geometry. Adapted from [Pinel et al., 2014].**
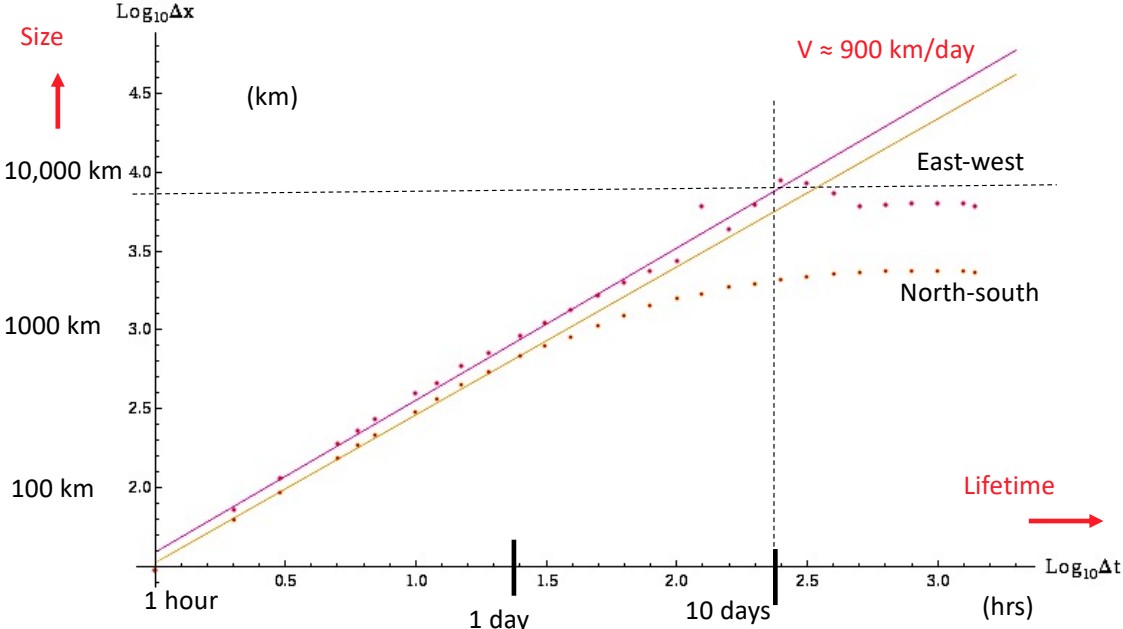

**Fig.26: The Eulerian (fixed frame) space-time diagram obtained from the satellite pictures analyzed in fig. 25, lower left, reproduced from [*Pinel et al.*, 2014]. The slopes of the reference lines correspond to averages winds of 900 km/day, i.e. about 11 m/s. The dashed reference lines show the spatial scales corresponding to 1 and 10 days respectively.**

## 3. Scaling and multifractals

### 3.1 Scaling in one dimension: Time series and spatial transects

Up until now, we have discussed scaling at a fairly general level as an invariance under scale changes, contrasting it with scaleboundedness and emphasizing its indispensable role in understanding the atmosphere, the ocean and more generally, the geosphere. There are two basic elements that must be considered: a) the definition of the notion of scale and scale change, and b) the aspect of the system or process that is invariant under the corresponding change.

We have seen that in general terms, a system is scaling if there exists a power law relationship (possibly deterministic, but usually statistical) between fast and slow (time) or small and large (space, space-time). If the system is a geometric set of points – such as the set of meteorological measuring stations [*Lovejoy et al.*, 1986], then the set is a fractal set and the density of its points is scaling – it is a power law – whose exponent is its fractal codimension. Geophysically interesting systems are





typically not sets of points but rather scaling fields such as the temperature $T(\underline{r}, t)$ at the space-time point $(\underline{r}, t)$. Although we will generally use the term scaling"fields", for multifractals, the more precise notion is of singular densities of multifractal

measures.

In such a system, some aspect – most often a suitably defined fluctuation – $\Delta T$ - has statistics whose small and large scales are related by a scale changing operation that involves only the scale ratios: the system has no characteristic size.  In one dimension the scaling of temporal series or spatial transects can be expressed as:

$$\Delta T\left(\Delta t\right)\overset{d}{=}\varphi_{\Delta t}\Delta t^{H}$$

860                                                                                                     (12)

where $\Delta t$ is the time interval (scale) over which the fluctuations are defined (for transects, replace $\Delta t$ by the spatial interval $\Delta x$), $H$ is the fluctuation exponent and $\varphi_{\Delta t}$ is a random variable.  Dimensionally, the units of $\varphi$ determine $H$, physically $\varphi$ is a turbulent flux that drives the process (or a power of flux, see eq. 2).  In turbulence theory, the  fluxes in Fourier space, from small to large wavenumbers.  The subscript $\Delta t$ indicates that $\varphi$ generally depends on the scale (resolution).  The equality sign

$\overset{d}{=}$ is in the sense of random variables; this means that the random fluctuation $\Delta T\left(\Delta t\right)$ has the same probability distribution

as the random variable $\varphi_{\Delta t}\Delta t^{H}$ .  We suppressed the $t$ dependence since we consider the case where the statistics are independent of time or space - statistical stationarity or statistical homogeneity (see appendix B4).  Physically, this is the assumption that the underlying physical processes are the same at all times and everywhere in space.  Eq. 12 is the more formal expression of eq. 1 or  of the classical turbulence laws eq. 4.   For example, if we consider space rather than time, the

Kolmogorov law has $H = 1/3$ with $\varphi = \varepsilon^{1/3}$.

The simplest case is where the fluctuations in a temporal series $T(t)$ follow eq. 12, but with $\varphi$ a random variable independent

of resolution: $\varphi_{\Delta t}\overset{d}{=}\varphi$ .  This is the classical special case of nonintermittent, quasi-Gaussian turbulence.  Examples are fractional Gaussian noise (fGn, -1<$H$<0), and fractional Brownian motion (fBm, 0<$H$<1) with special cases Gaussian white noise ($H$ = -1/2), and standard Brownian motion ($H$ = ½).

Eq. 12 relates the probabilities of small and large fluctuations, it is usually easier  to deal with the deterministic equalities that follow by taking $q^{th}$ order statistical moments of Eq. 12 and then averaging over a statistical ensemble:

$$\left\langle \left(\Delta T\left(\Delta t\right)\right)^{q}\right\rangle = \left\langle \varphi_{\Delta t}^{q}\right\rangle \Delta t^{qH}$$                                                                                  (13)

where "< >" indicates statistical (ensemble) averaging.

Eq. 13 is the general case where the resolution $\Delta t$ is important for the statistics of $\varphi$; indeed, quite generally,  $\varphi_{\Delta t}$  is a random

function (1-D) or field averaged at resolution $\Delta t$.  If $\varphi_{\Delta t}$  is scaling, its statistics will follow:





$$\left\langle \varphi_\lambda^q \right\rangle = \lambda^{K(q)}; \quad \lambda = \tau / \Delta t \geq 1 \tag{14}$$

where $\tau$ is the largest, "outer" scale of the scaling regime satisfied by the equation, the $\Delta t$ resolution is $\lambda$ times smaller. $K(q)$ is a convex ($K''(q) \geq 0$) exponent function; since the mean fluctuation is independent of scale ($\left\langle \varphi_\lambda \right\rangle = \lambda^{K(1)} = const$), we have $K(1) = 0$ (see eq. 2). This is the generic statistical behaviour of cascade processes, it displays general "multiscaling"

behaviour - a different scaling exponent $K(q)$ for each statistical moment $q$. Since large $q$ moments are dominated by large, extreme values, and small $q$ moments by common, typical values, $K(q) \neq 0$ implies that fluctuations of various amplitudes change scale with different exponents - multiscaling - and each realization of such a process is a multifractal. In general, $K(q) \neq 0$ is associated with intermittency, a topic we treat in more detail in sections 3.2, 3.3.

Combining Equations 13, 14, we obtain:

$$S_q(\Delta t) = \left\langle \left(\Delta T(\Delta t)\right)^q \right\rangle = \left\langle \varphi_{\Delta t}^q \right\rangle \Delta t^{qH} \propto \Delta t^{\xi(q)}; \quad \xi(q) = qH - K(q) \tag{15}$$

where $S_q$ is the $q^{th}$ order ("generalized") structure function and $\xi(q)$ is its exponent defined in eq. 15. Since $K(q)$ is convex, eq. 15 shows that in general, $\xi(q)$ will be concave ($\xi'' < 0$). The structure functions are scaling since the small and large scales are related by a power law:

$$S_q(\lambda \Delta t) = \lambda^{\xi(q)} S_q(\Delta t) \tag{16}$$

As discussed in section 2.3 the spectra are power laws $E(\omega) \approx \omega^{-\beta}$ with the exponents related as $\beta = 1 + \xi(2) = 1 + 2H - K(2)$, eq. 9.

In the case of "simple scaling" where $\varphi$ has no scale dependence (e.g. fGn, fBm), we find $\left\langle \varphi_{\Delta t}^q \right\rangle = B_q$ where $B_q$ is a constant independent of scale $\Delta t$, hence we have $K(q) = 0$ and $S_q(\Delta t) \propto \Delta t^{qH}$ so that:

$$\xi(q) = qH \tag{17}$$

i.e. $\xi(q)$ is a linear function of $q$. Simple scaling is therefore sometimes called "linear scaling" and it respects the simpler relation $\beta = 1 + 2H$. Linear scaling arises from scaling linear transformations of noises; the general linear scaling transformation is a power law filter (multiplication of the Fourier Transform by $\omega^{-H}$) - or equivalently - fractional integrals ($H > 0$) or fractional derivatives ($H < 0$). Appropriate fractional integrals of Gaussian white noises yield fBm ($1 > H > 0$) and fractional derivatives





yield fGn ($-1 < H < 0$). The analogous Levy motions and noises are obtained by the filtering of independent Levy noises (in this
case, $\xi(q)$ is only linear for $q < \alpha < 2$; for $q > \alpha$, the moments diverge so that both $\xi(q)$ and $S_q \to \infty$).

The more general "nonlinear scaling" case where $K(q)$ is nonzero and convex, is associated with fractional integrals or
derivatives of scaling multiplicative (not additive) random processes (cascades, multifractals), these pure multiplicative
processes ($\varphi$ in eq. 12) have $H = 0$, they are sometimes called "conservative multifractals" since their exponent of the mean
$\xi(1) = H = 0$ (i.e. the mean is independent of scale). The exponent H in eq. 15 still refers to the order of fractional integration
($H > 0$) or differentiation ($H < 0$) of $\varphi$ that adds the extra term $qH$ in the structure function exponent: $\xi(q) = qH - K(q)$. Note
that while the symbol $H$ is in honour of Edwin Hurst; the interpretation of $H$ as the "Hurst exponent" is only valid for Gaussian
processes, more generally it is a fluctuation exponent describing the behaviour of the mean ($q = 1$) fluctuation.
We could mention that here and in section 3.3 where we discuss the corresponding multiscaling probability distributions, we
use the $c(\gamma)$, $K(q)$ codimension multifractal formalism that is appropriate for stochastic multifractals [*Schertzer and Lovejoy*,
1987]. Another commonly used multifractal formalism is the $\alpha$, $f(\alpha)$, $\tau(q)$ dimension formalism of [*Halsey et al.*, 1986] that
was developed for deterministic chaos applications. The relationship between the two formalisms is $f(\alpha) = d - c(\gamma)$
where $d$ is the dimension of the space in which the multifractal process is defined. and $\alpha = d - \gamma$ is the singularity of the
measure of the multifractal - not it's density whose singularity is $\gamma$. For the moment exponent functions, we have

$$\tau(q) = d(q-1) - K(q).$$

The $\alpha$, $f(\alpha)$, $\tau(q)$ "dimension" formalism was developed to characterize deterministic chaos in low ($d$) dimensional spaces,
Here we are interested in stochastic multifractal processes that are defined on probability spaces with $d = \infty$. Therefore, the
codimension formalism - independent of $d$ – is required. Another difference between the formalisms is that the singularity of
the multifractal measure $\alpha$ is assumed to be defined at a mathematical point (it is a "Holder" exponent) whereas in the
codimension formalism, $\gamma$ is the singularity of the *density* of the multifractal measure, and only a looser convergence in the
neighbourhood of a point is assumed. This lack of pointwise convergence of the singularities is a general feature of (stochastic)
cascade processes and is hence more relevant in the atmosphere.

### 3.2 Spikiness, Intermittency and multifractals

### 3.2.1 Spikes, singularities, co-dimensions

Atmospheric modeling is classically done using the deterministic equations of thermodynamics and continuum mechanics.
However, in principle, one could have used a more fundamental (lower level) approach - statistical mechanics - but this would
have been impossibly difficult. Yet, in strongly nonlinear fluid flow, the same hierarchy of theories continues to higher level
turbulent laws. These laws are scaling and may – depending on the application – be simpler and more useful. A concrete
example is in the macroweather regime where (strongly nonlinear, deterministic) GCMs are taken past their deterministic




predictability limit of about 10 days, effectively becoming stochastic. To some degree of approximation, since the
intermittency is low (the spikiness in the right hand side of fig. 2, bottom of fig. 3), this stochastic behaviour is amenable to
modelling by *linear* stochastic processes, in this case, the Half-Order and Fractional Energy Balance Equations (HEBE, FEBE,
[*Lovejoy*, 2021a; b; *Lovejoy et al.*, 2021], [*Lovejoy*, 2022a]). The key issue – of whether linear or nonlinear *stochastic*
processes can be used thus depends on their "spikiness" or intermittency (multifractality).

Classically, intermittency was first identified in laboratory flows as "spottiness" [*Batchelor and Townsend*, 1949]; in the
atmosphere by the concentration of atmospheric fluxes in tiny, sparse regions. In time series, it is associated with turbulent
flows undergoing transitions from "quiescence" to "chaos". Quantitative intermittency definitions developed originally
for fields (space) are of the "on-off" type, the idea being that when the energy or other flux exceeds a threshold then it
is "on" i.e. in a special state - perhaps of strong/violent activity. At a specific measurement resolution, the on-off
intermittency can be defined as the fraction of space that the field is "on" (where it exceeds the threshold). In a scaling
system, for any threshold the "on" region will be a fractal set and both the fraction and threshold will be characterized
by exponents (by $c$ and $\gamma$, introduced shortly) that describe the intermittency over all scales and all intensities
(thresholds). In scaling time series, the same intermittency definition applies; note however that other definitions are
sometimes used in series in deterministic chaos.

With the help of multifractals we can now quantitatively interpret the spike plots. Recall that in eq. 12, $\varphi_{\Delta t}$ is the flux driving
the process normalized so that $\langle \varphi_{\Delta t} \rangle = 1$. If we estimate the ensemble mean flux by the (temporal) average flux over the
entire time series, and then averaged over all the available series (indicated by an overbar), then the normalized spikes
$\Delta T / \overline{\Delta T}$ are estimates of the nondimensional, normalized driving fluxes:

$$\Delta T(\Delta t) / \overline{\Delta T(\Delta t)} = \varphi_{\Delta t,un} / \overline{\varphi_{\Delta t,un}} = \varphi_\lambda; \quad \lambda = \tau / \Delta t \tag{18}$$

where $\varphi_{\Delta t,un}$ is the raw, unnormalized flux, the outer scale of the scaling regime is $\tau$ so that the normalized flux $\varphi_\lambda$ is over
scale ratio $\lambda$. In the weather regime in respectively time and space, the squares and cubes of the wind spikes are estimates of
the turbulent energy fluxes. This spikiness is because most of the dynamically important events are sparse, hierarchically
clustered, occurring mostly in storms and the centre of storms.

As long as *H*<1 (true for nearly all geo-processes), the differencing that yields the spikes acts as a high pass filter, the spikes
are dominated by the high frequencies. Smoothed Gaussian white noises such as the scaling fractional Gaussian noise (fGn)
and fractional Brownian motion (fBm) processes, or nonscaling processes such as autoregressive and moving average
processes will have spikes that look like the weak macroweather spikes in fig. 2, roughly bounded by the solid horizontal line
in the figure.





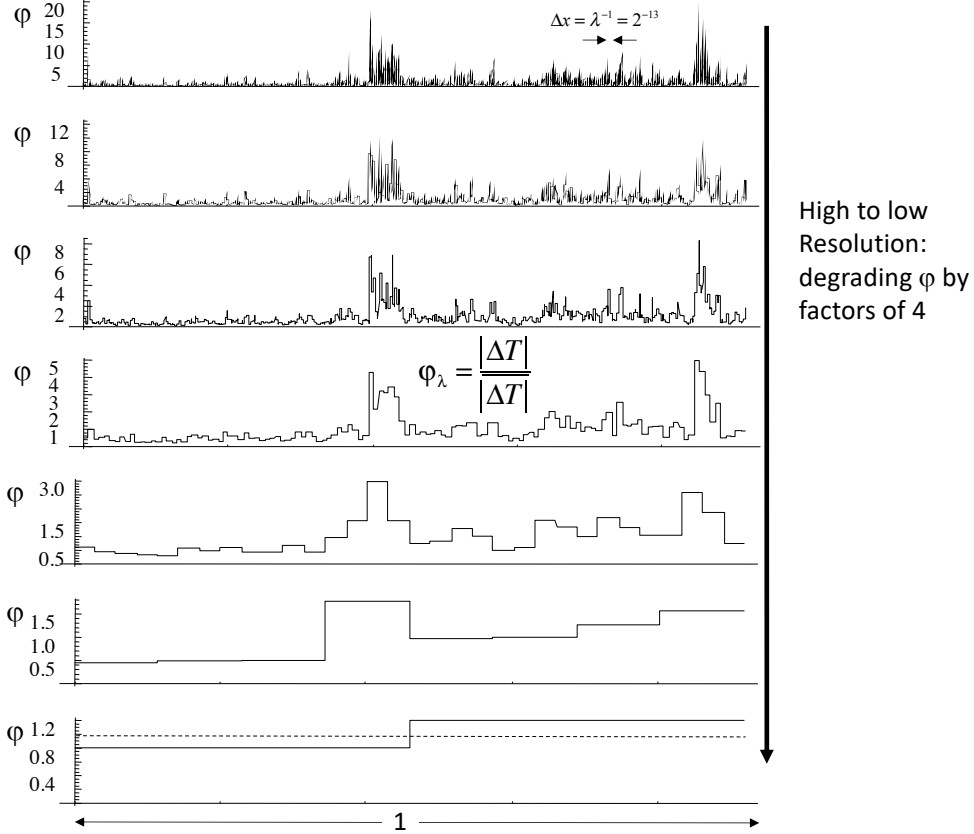

**Fig. 27: The top row is a reproduction of the intermittent spikes taken from the gradients in the aircraft data at the bottom of fig. 3. The original series is 2294 km long with resolution 280m hence it covers a scale range of a factor of $\lambda = 2^{13}$. Here we use nondimensional units so that the length is 1 with resolution $\lambda^{-1} = 2^{-13}$. Moving from top to bottom, each row degrades (by averaging) the resolution of the previous is by a factor of 4. Note the scale on the left is constantly changing. At the bottom, the dashed line indicates the mean which is unity since φ is a normalized process. Reproduced from [Lovejoy 2019].**





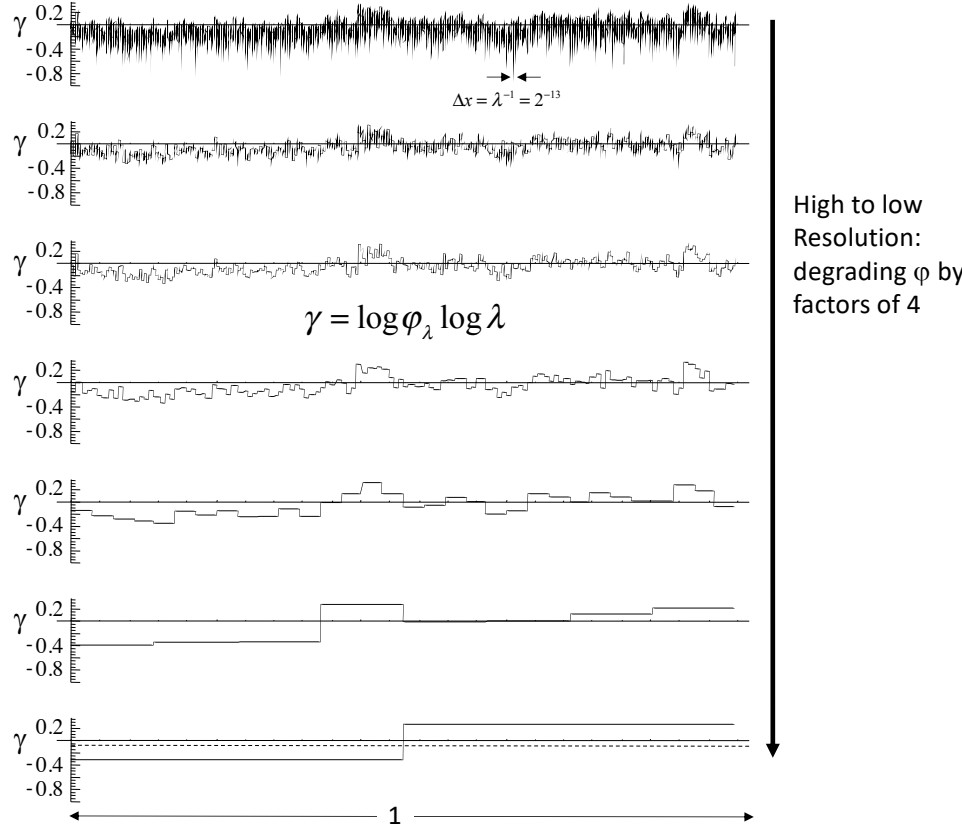

**Fig. 28: The same as fig. 27 but in terms of the corresponding singularities obtained through the transformation of variables** $\gamma = \log\varphi/\log\lambda$. **Notice that while the range of variation of the $\varphi$ in the previous figure rapidly diminishes as the resolution is lowered, on the contrary the amplitude of the fluctuations of the $\gamma$'s (above) is roughly the same at all scales. Note that the dashed horizontal line on the bottom plot shows the mean singularity, here =-0.06. It is <0 since it is the mean of the log of the normalized flux and the logarithm function is concave. Adapted from [*Lovejoy and Schertzer*, 2013].**

What happens if we change the resolution of $\varphi_\lambda$ by averaging it over larger and larger scales $\Delta t$ (smaller $\lambda$)? Fig. 27 shows this using the example of the spatial (aircraft) transect in fig. 3. The top plot is identical to the bottom of fig. 3 except that nondimensional units are used so that the top spike transect $\varphi_\lambda$ has length 1 with $\lambda = L / \Delta x = 2^{13} =$ where $L = 2294\ km$ is the actual length and $\Delta x = 280\ m$ is the transect resolution. As we move from top to bottom, the resolution is successively degraded by factors of 4 ($\lambda$ decreases by factors of 4). Since the flux is normalized by its mean ($\lambda = 1$), the fluctuations are about unity (the dashed line at the bottom where $\lambda = 2$).





Examine now the vertical axes. We see that - as expected – the amplitude of the spikes systematically decreases with
resolution, the plots are clearly *not* scale invariant. We would like to have a scale invariant description of the spikes, a scale
invariant probability distribution of the spikes. For this, each spike is considered to be a singularity of order γ:

$$\lambda^{\gamma} = \frac{|\Delta T|}{\overline{|\Delta T|}} \qquad (19)$$

This is simply a transformation of variables from the spikes $|\Delta T|/\overline{|\Delta T|}$ to singularities $\gamma = \log\left(|\Delta T|/\overline{|\Delta T|}\right)/\log\lambda$.

Figure 27 shows the same spikes but now in terms of the orders of singularity. Now we see that the vertical range is pretty
much independent of the resolution. It is therefore plausible that the characterization of the spikes by γ is scale invariant. To
obtain a scale invariant characterization of the probabilities, introduce the codimension function c(γ), the spike probability
distribution may be written:

$$\Pr\left(\frac{|\Delta T|}{\overline{|\Delta T|}} > s\right) = P(s)\lambda^{-c(\gamma)}; \quad \gamma = \frac{\log s}{\log \lambda} \qquad (20)$$


$\Pr\left(\dfrac{|\Delta T|}{\overline{|\Delta T|}} > s\right)$ is the probability that a randomly chosen normalizedspike $|\Delta T|/\overline{|\Delta T|}$ exceeds a fixed threshold $s$ (it is equal

to one minus the more usual cumulative distribution function), *P(s)* is a nonscaling prefactor that depends on *s* and weakly on
γ. To leading order (i.e. putting the prefactor ≈1), we obtain:

$$c(\gamma) \approx -\log\Pr/\log\lambda; \quad \gamma = \log\left(|\Delta T|/\overline{|\Delta T|}\right)/\log\lambda \qquad (21)$$

We see while γ gives a scale invariant characterization of the spikes, c(γ) does the same for the probability distributions. *c(γ)*
characterizes sparseness because it quantifies how the probabilities of spikes of different amplitudes change with resolution λ.

*c(γ)* corresponds to the sparse set of spikes that exceed the threshold $s = \lambda^{\gamma}$. Increasing the spike amplitude (γ) defines a sparse

exceedance set with large *c*. A series is intermittent whenever it has spikes with *c>0*.

In the general scaling case, the set of spikes that exceed a given threshold form a fractal set whose sparseness is quantified by
the fractal codimension *c(γ) = d - D(γ)* where *d* is dimension of the space (*d = 1* for series and transects) and *D(γ)* is the
corresponding fractal dimension. The codimension is fundamental since it is the exponent of the probability distribution:





Gaussian series are not intermittent since $c(\gamma) = 0$ for all the spikes. To see this, note that for Gaussian processes the cumulative probability of a spike exceeding a fixed threshold $s$ is independent of the resolution $\lambda$, $\Pr\left(\left|\Delta T\right| / \overline{\left|\Delta T\right|} > s\right) \approx P(s)$, where here, $P(s)$ is related to the error function. Comparing this with Eq. 20, we see that $c = 0$.

Returning to Figure 2, we have $\lambda = 1000$. The extreme spikes ($\left|\Delta T\right| / \overline{\left|\Delta T\right|}$) in this 1000 point long series have a probability $\lambda^{-1} \approx 1/1000$. For Gaussian processes, the spikes with this probability are $\left|\Delta T\right| / \overline{\left|\Delta T\right|} = 4.12$, this is shown by the solid lines in figure 2, the line therefore corresponds to $\gamma = \gamma_{\max} = \log\left(\left|\Delta T\right| / \overline{\left|\Delta T\right|}\right) / \log\lambda \approx \log 4.12 / \log 1000 \approx 0.20$. If the series in fig. 2 were generated by multifractal processes, what is the maximum $\gamma$ (and hence spike) that we would expect? The extreme value would still correspond to $\lambda^{-1}$ hence from eq. 20, we have $c\left(\gamma_{\max}\right) = 1$. More generally, in a space of dimension

$d$, there would be $\lambda^d$ spikes, the probability of the maximum would be $\lambda^{-d}$ so that $c\left(\gamma_{\max}\right) = d$. Since the fractal dimension of the spikes is $D(\gamma) = d - c(\gamma)$, this is simply the result that $D\left(\gamma_{\max}\right) = 0$. Since $c(\gamma)$ is a monotonically increasing function, this is just the simple geometric result that the fractal dimension of the exceedance sets cannot be negative. Appendix A, table 2 gives the both the observed maximum $\gamma$ for each series in Figure 2 as well as the generally comparable theoretically expected maxima for the multifractal processes with the parameters estimated for the series in question.

1020        Whereas $c(\gamma)$ quantifies the way the probability distributions of spikes change with scale, the moment scaling exponent $K(q)$ (eq. 14) quantifies the way the statistical moments change with scale. Since the process can be equivalently characterized by either probabilities or moments, $c$ and $K$ must be related. Indeed, the relationship is beautiful and simple: via a Legendre transformation:

$$K(q) = \max_{\gamma}\left(q\gamma - c(\gamma)\right); \quad c(\gamma) = \max_{q}\left(q\gamma - K(q)\right) \tag{22}$$

[*Parisi and Frisch*, 1985].

These equations imply one to one relationships between the spike singularities $\gamma$ (and amplitudes $\lambda^{\gamma}$) and the exponent of the order of moments $q$, they imply: $K'(q) = \gamma$ and $c'(\gamma) = q$. $K'(q=1) = \gamma_1$ is therefore the singularity that gives the dominant contribution to the mean ($q = 1$) of the process. At the same time, $K(1) = 0$ so that (eq. 21) $K(1) = 0 = \gamma_1 - c(\gamma_1)$ (where $\gamma_1$ is the value that give the maximum of $\gamma - c(\gamma)$) so that we obtain $\gamma_1 = c(\gamma_1)$ and since $K'(1) = \gamma_1$, we have $K'(1) = c(\gamma_1)$.

Defining $C_1 = c(\gamma_1) =$ the codimensions of the singularity $\gamma_1$ that gives the dominant contribution to the mean, we have $K'(1) = C_1$. Thus $C_1$ plays the dual role of being the order of singularity that gives the dominant contribution to the mean while also





being equal to the codimension of the set of the corresponding singularities. Since we see that $\gamma_1 = C_1 = c(C_1)$, this justifies the interpretation of $C_1 = \gamma_1 =$ the codimension of the mean.

### 3.2.2 Universal Multifractals

At first sight, general (multifractal) scaling involves an entire exponent function – either $K(q)$ or $c(\gamma)$ - for its statistical characterisation, the equivalent of an infinite number of parameters (e.g. one for each statistical moment). This would be unmanageable – either from the point of view of empirical parameter estimation or from the point of view of model construction. Fortunately, one can avail oneself of a multiplicative version of the central limit theorem, that leads to "universal multifractals" with:

$$K(q) = \frac{C_1}{\alpha - 1}(q^\alpha - q)$$

(23)

and (via Legendre transform):

$$c(\gamma) = C_1 \left( \frac{\gamma}{C_1 \alpha'} + \frac{1}{\alpha} \right)^{\alpha'}; \quad \frac{1}{\alpha} + \frac{1}{\alpha'} = 1; \quad 0 \le \alpha \le 2$$

(24)

[Schertzer and Lovejoy, 1987]. $C_1$ is the codimension of the mean introduced earlier and $0 \le \alpha \le 2$ is the Levy index, $\alpha'$ is an auxiliary variable introduced for convenience.

Fig. 28 and 29 show the universal $K(q)$, $c(\gamma)$ functions for various values of $\alpha$ in the relevant range $0 \le \alpha \le 2$. The lower limit $\alpha = 0$ corresponds to the on/off, "monofractal" "$\beta$ model" [*Novikov and Stewart*, 1964], [*Frisch et al.*, 1978] where all the fluxes are concentrated on a fractal set with codimension $C_1$ and the upper limit $\alpha = 2$ to the "log normal" multifractal [*Kolmogorov*, 1962], [*Yaglom*, 1966]. Note that due to the divergence of moments discussed in section 3.5, the multifractals with the above $K(q)$, $c(\gamma)$ are only approximately "log-Levy" (or when $\alpha = 2$, "log-normal").

Table 1 shows various empirical estimates relevant to atmospheric dynamics. We see that generally $1.5 \lessapprox \alpha < 2$ and $C_1 \approx 0.1$, the main exception being precipitation. As quantified in table 1, precipitation is the most strongly intermittent atmospheric field ($C_1 \approx 0.4$) quantitatively confirming the subjective impression of extreme precipitation intermittency. The multifractal properties of precipitation have been the subject of numerous studies. Early analyses include spatial analyses by [*Tessier et al.*, 1993], Olsson and Niemczynowicz, 1996, [*de Montera et al.*, 2009] and [*Verrier et al.*, 2010]),

[*Veneziano et al.*, 2006] and temporal analyses by [*Tessier et al.*, 1993], [*Hubert et al.*, 1993], [*Ladoy et al.*, 1993], [*Harris et al.*, 1996]; [*De Lima*, 1998], [*De Lima and Grasman*, 1999], [*Hubert et al.*, 2002; *Kiely and Ivanova*, 1999], [*Hubert et al.*, 2002]; [*Pathirana et al.*, 2003], [*Venugopal et al.*, 2006], [*Garcia-Marin et al.*, 2008; *Pathirana et al.*, 2003], [*Serinaldi*,




2010]; [*Sun and Barros*, 2010]; [*Schertzer et al.*, 2011]; [*Verrier et al.*, 2011]. There is more discussion of table 1 in section 3.4.

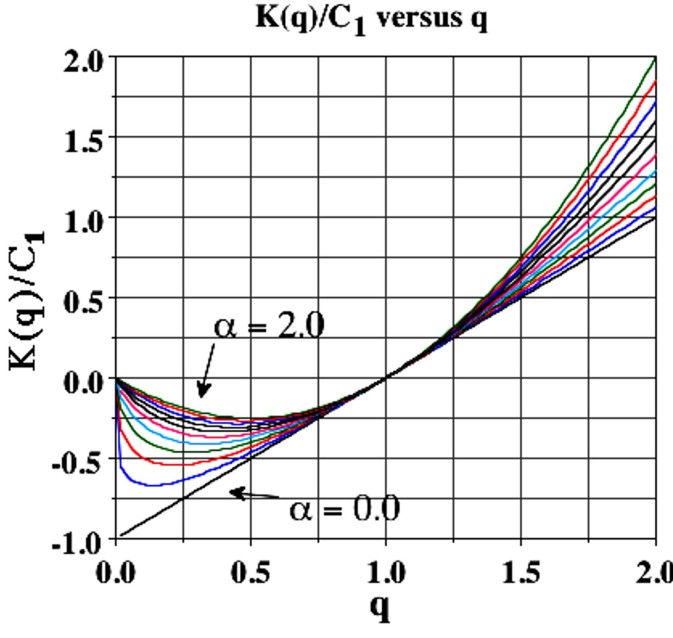


**Fig. 29: Universal *K(q)* as a function of *q* for different a values from 0 to 2 in increments of 0.2. Adapted from [*Schertzer and Lovejoy*, 1989].**


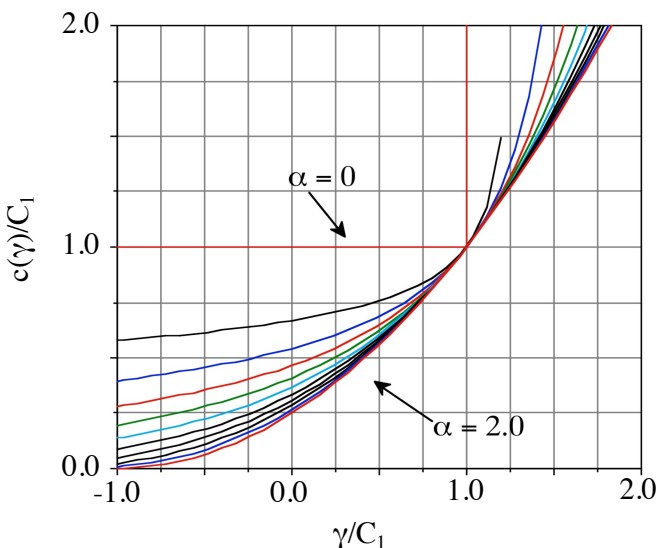

l065 **Fig. 30: Universal c(γ) for α in the range 0 to 2 in increments of 0.2. Adapted from [*Schertzer and Lovejoy*, 1989].**





| | | $C_1$ | α | $H$ | β | $L_{eff}$ |
|---|---|---|---|---|---|---|
| State variables | $u, v$ | 0.09 | 1.9 | 1/3, (0.77) | 1.6, (2.4) | (14000) |
| | $w$ | (0.12) | (1.9) | (-0.14) | (0.4) | (15000) |
| | $T$ | 0.11, (0.08) | 1.8 | 0.50, (0.77) | 1.9, (2.4) | 5000 (19000) |
| | $h$ | 0.09 | 1.8 | 0.51 | 1.9 | 10000 |
| | $z$ | (0.09) | (1.9) | (1.26) | (3.3) | (60000) |
| Precipitation | $R$ | 0.4 | 1.5 | 0.00 | 0.2 | 32000 |
| Passive Scalars | Aerosol concentration | 0.08 | 1.8 | 0.33 | 1.6 | 25000 |
| Radiances | Infra Red | 0.08 | 1.5 | 0.3 | 1.5 | 15000 |
| | visible | 0.08 | 1.5 | 0.2 | 1.5 | 10000 |
| | Passive microwave | 0.1-0.26 | 1.5 | 0.25-0.5 | 1.3-1.6 | 5000-15000 |
| Topography | Altitude | 0.12 | 1.8 | 0.7 | 2.1 | 20000 |
| Sea Surface Temperature | SST | 0.12 | 1.9 | 0.50 | 1.8 | 16000 |

**Table 1: We compare various horizontal parameter estimates, attempting to give summarize categories of values (radiances) or approximate values ($u, v, w$ are the zonal, meridional and vertical wind, $T$ the temperature, $h$ the humidity, $z$ the pressure height). $C_1$ is the codimension of the mean, a convenient intermttency parameter, a, the multifractal index, $H$ the fluctuation exp0ooent, b the spectral exponent. $L_{eff}$ is the effective outer cascade scale (determined from the crossing scales of the different moments). When available (and when reliable), the aircraft data were used in precedence over the reanalysis values with the latter given in parentheses in those cases where there was no comparable in situ value or when it was significantly different from the in situ value. For $L_{eff}$ where the anisotropy is significant, the geometric mean of the north-south and east-west estimates are given the average ratio is 1.6 : 1 EW/NS (although for the precipitation rate, the along-track TRMM estimate was used). Finally, the topography estimate of $L_{eff}$ is based on a single realization (one earth!) so that we only verified that there was no obvious break below planetary scales. The aerosol concentration was estimated from the lidar backscatter ratio from the data in fig. 45.**

## 3.3 Quantifying Intermittency with structure functions

Using spike plots, we can simply demonstrate the unique character of the macroweather regime: low intermittency in time, but high intermittency in space. We introduced the $c(\gamma)$ function that for each spike level $\lambda^\gamma$, characterizes the probability (fraction)





of a transect or series (or more generally, space) whose spikes exceed the threshold. In this section we discuss a particularly simple way to analyze the intermittency.

Consider the data shown in fig. 31 (macroweather time series and spatial transects, top and bottom respectively). Fig. 32 compares the root mean square (RMS) fluctuations with exponent $\xi(2)/2$ and the mean fluctuations with exponent $H = \xi(1)$

from macroweather temperature time series (bottom) and for the spatial transects (top). When the system is Gaussian (so that $K(q) = 0$), we obtain $\xi(2)/2 = \xi(1)$; so that moments of the mean and RMS fluctuations are in a constant ratio, in this case, log $\langle\Delta T(\Delta t)\rangle$ is parallel to log $\langle\Delta T(\Delta t)^2\rangle^{1/2}$. Figure 32 (bottom) shows that to a good approximation this is indeed true of the nonspiky temporal series (fig. 32, top). However, the corresponding statistics of the spatial transect (the top lines in fig. 32) tend to converge at large $\Delta x$ corresponding to the highly spikey transect (fig. 32, bottom). To a first approximation, it turns

out that $\xi(2)/2 - \xi(1) \approx K'(1) = C_1$ which characterizes the intermittency near the mean. However, there is a slightly better way to estimate $C_1$, using the intermittency function (see fig. 33 and caption) whose theoretical slope (for ensemble averaged statistics) is exactly $K'(1) = C_1$. As a point of comparison, we could note that fully developed turbulence in the weather regime typically has $C_1 \approx 0.09$. The temporal macroweather intermittency ($C_1 \approx 0.01$) is indeed small whereas the spatial intermittency is large ($C_1 \approx 0.12$).

For many applications, the exceptional smallness of macroweather intermittency makes the "monoscaling" approximation (i.e. $\xi(q) \approx Hq$) acceptably accurate so that macroweather processes are relatively easy to statistically characterize. In this case, the fluctuation exponent $H$, the spectral exponent $\beta$ and the Detrended Fluctuation Analysis exponent $a$ (appendix A5) are equivalent and sufficient (the general relations are: $H = (\beta - 1 + K(2))/2 = a - 1 + K(2)/2$ and here, with no intermittency, $K(2) = 0$, and these simplify to $H = (\beta - 1)/2 = a - 1$). For examples of macroweather scaling, see [*Tessier*

*et al.*, 1996], [*Pandey et al.*, 1998], [*Koscielny-Bunde et al.*, 1998], [*Bunde et al.*, 2004; *Eichner et al.*, 2003], [*Blender et al.*, 2006], [*Huybers and Curry*, 2006], [*Rybski et al.*, 2006], [*Lennartz and Bunde*, 2009], [*Lanfredi et al.*, 2009], , [*Fraedrich et al.*, 2009], [*Franzke*, 2010], [*Franzke*, 2012], [*Varotsos et al.*, 2013], [*Varotsos et al.*, 2009], [*de Lima and Lovejoy*, 2015].



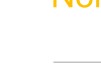 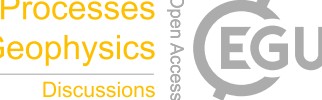
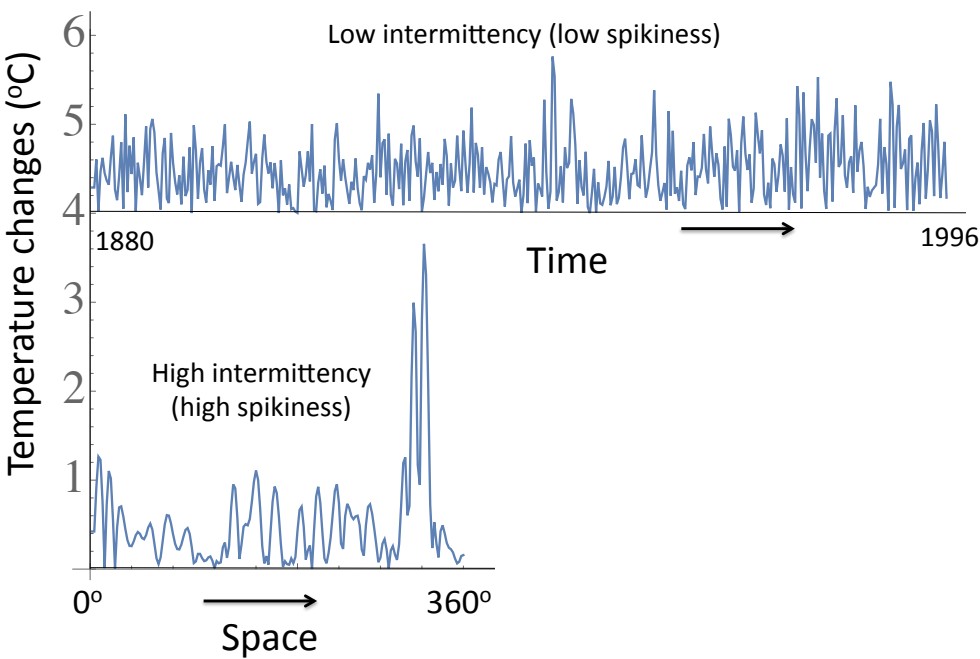

**Fig. 31: A comparison of temporal and spatial macroweather series at $2^{o}$ resolution. The top are the absolute first differences of a temperature time series at monthly resolution (from $80^{o}$ E, $10^{o}$ N, 1880 -1996, displaced by 4K for clarity), and the bottom is the series of absolute first differences of a spatial latitudinal transect (annually averaged, 1990 from $60^{o}$ N), as a function of longitude. Both use data from the 20CR. One can see that while the top is noisy, it is not very "spikey". Quantitatively, the intermittency parameter near the mean is $C_1 \approx 0.01$ (time), $C_1 \approx 0.12$ (space). Reproduced from [*Lovejoy*, 2022c].**





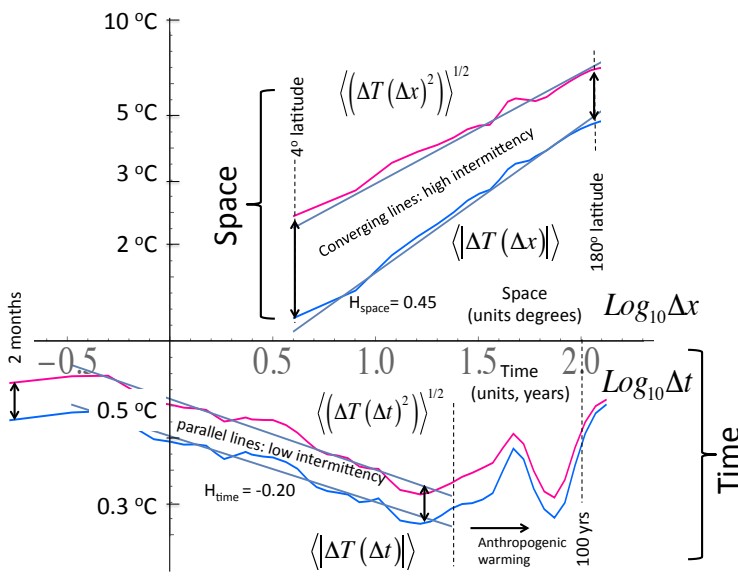

**Fig. 32. The first order and RMS Haar fluctuations of the series and transect in fig. 29. One can see that in the spikey transect, the fluctuation statistics converge at large lags ($\Delta x$), the rate of the converge is quantified by the intermittency parameter $C_1$. The time series (bottom) is less spikey, converges very little and has low $C_1$ (see fig. 30 top). The break in the scaling at ≈ 20 years is due to the dominance of anthropogenic effects at longer time scales. Reproduced from [*Lovejoy*, 2022c].**

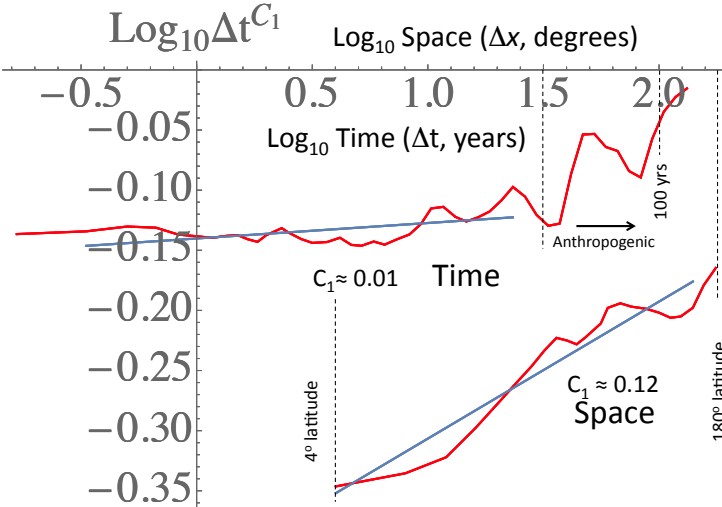





**Fig. 33: A comparison of the intermittency function** $F = \left\langle |\Delta T| \right\rangle \left( \left\langle |\Delta T|^{1+\Delta q} \right\rangle \right) / \left( \left\langle |\Delta T|^{1-\Delta q} \right\rangle \right)^{1/\Delta q}$ **for the series and transect in**

**the fig. 27, quantifying the difference in intermittencies: in time $C_1 \approx 0.01$, in space, $C_1 \approx 0.12$. Since $K'(1) = C_1$, when $\Delta q$ is small**
**enough (here, $\Delta q = 0.1$ was used), we have $F(\Delta t) = \Delta t^{C_1}$. The break in the temporal scaling at about $20 - 30$ years ($\log_{10} \Delta t \approx 1.5$) is**
**due to anthropogenic forcings. Reproduced from [*Lovejoy*, 2022c].**

**3.4 Multifractal analyses of geofields: Direct (trace moment) estimates of outer scales and $K(q)$ for Earth and Mars**

In the preceding, we gave evidence that diverse atmospheric fields are scaling up to planetary scales. In addition, we argued

that they generally were multifractal, with each statistical moment $q$ having a different exponent ($K(q)$, eq. 14). In section 3.2,

we saw that this nonlinear, convex part of the structure function exponent $\xi(q)$ arises due to variability building up scale by

scale from a large external scale $L$ to smaller scales $\Delta x$, (ratio $\lambda = L/\Delta x$ or in time, $\lambda = \tau/\Delta t$). By analyzing the statistics of the

125 fluxes $\left\langle \varphi_\lambda^q \right\rangle$, this gives us the possibility of directly determining the effective outer scale of the process i.e. the scale at which

the variability starts to grow. As a bonus, our method is based on isolating the flux, in the exponent, it yields $K(q)$ rather than

$\xi(q)$ (fluctuations). By effectively removing the $qH$ term (i.e. $\xi(q)-K(q)$), it is only sensitive to $K(q)$ which for small $q$ is often

a small correction to $\xi(q)$.

Before proceeding to empirical analyses of the fluxes, a few comments are required. The flux in Eq. 14 is assumed to be

normalized i.e. $\left\langle \varphi_\lambda \right\rangle = 1$. For empirical estimates one starts with unnormalized fluxes and one doesn't know a priori the

effective outer scale of the variability $L$ that is needed to estimate the ratio $\lambda$. The normalization problem is easy to solve, see

eq. 18. For empirical estimates, one therefore starts with these normalized fluxes at the smallest available resolution (i.e. $\Delta x$

$= 1$ pixel); using this, lower resolution estimates (i.e. larger $\Delta x$) are obtained simply by averaging. However, to verify Eq. 14,

we need the scale ratio $\lambda$ which is the ratio of the (a priori unknown) outer scale $L$ to the resolution $\Delta x$. The simplest procedure

is to use the largest planetary scale $L_{ref}$ (half the Earth circumference) as an initial reference scale; a guess hopefully not far

from the true outer scale $L$, a kind of "bootstrap". When this is done, the statistics of the various moments as functions of the

reference scale ratio (i.e. with $\lambda_{ref} = L_{ref}/\Delta x$ in place of $\lambda$: $\left\langle \varphi_{L_{ref}/\Delta x}^q \right\rangle$) are plotted on a log-log plot. For each moment order of

$q$, the regressions (with slopes $K(q)$) all converge to the true outer scale; this is because at that scale $\lambda = 1$ and (Eq. 14) shows

that $\left\langle \varphi_{\lambda=1}^q \right\rangle = 1$ for all $q$.

Fig. 34 shows the first trace moment estimate [*Schertzer and Lovejoy*, 1987]. It was applied to data from a land based radar

whose 3 km altitude reflectivity maps were 128 km wide with a 1 km resolution. The vertical axis is $Log_{10} M_q$ where

$M_q = \left\langle \varphi_\lambda^q \right\rangle$ and $\lambda = L_{ref} / \Delta x$ with $L_{ref} = 128$ km. Although this gives a tantalizing hint that atmospheric cascades start at





planetary scales, it wasn't until ten years of satellite radar data were released over the internet that this was confirmed directly (fig. 35, [*Lovejoy et al.*, 2009e]). The poor scaling (curvature) for the low $q$ values (bottom) were quantitatively explained as artefacts of the fairly high minimum detectable signal. Fig. 36 shows similar results, but this time using the same geostationary satellite data whose spectra was analysed in fig. 25. An interesting comparison of horizontal and vertical cascade structures from vertical sections of lidar aerosol backscatter is shown in fig. 37. Although this is discussed in section 4.1, we can already note that the outer scales are roughly the largest available ($\approx 20000$ km in the horizontal and $\approx 10$ km in the vertical), but also analysis shows that the slopes ($K(q)$) are the theoretically predicted ratio $K_{hor}/K_{vert} = H_z = 5/9$ (for all $q$, section 4).

The trace moments characterize a fundamental aspect of the atmosphere's nonlinear dynamics - its the intermittency (they would all have zero slopes for quasi-Gaussian fields), in fully developed turbulence, it is expected to be a "universal" feature i.e. found in all high Reynold's number flows. In our case, the closest to universality is to compare Earth to Mars (using the same reanalyses as in fig. 9). Fig. 38 shows the result when this technique is applied to both terrestrial and Martian reanalyses for pressure, wind and temperature (for both planets, the analyses were at altitudes corresponding to about 70% of surface pressure). One can note a) as predicted, the turbulence is universal, i.e. not sensitive to the forcing mechanisms and boundaries so that the behaviour is nearly identical on the two planets, b) there is clear multiscaling (the logarithmic slopes $K(q) \neq 0$), c) the effective outer scales (where the lines converge) is indeed nearly the size of the planet. For more detailed discussion and analyses (including spectra and horizontal anisotropy), see [*Chen et al.*, 2016].

Table 1 shows typical values of multifractal parameters estimated from trace moments (section 3.4) of various atmospheric fields. Over the decades, many multifractal analyses of geofields have been performed, including of atmospheric boundary conditions (notably the topography on Earth [*Lavallée et al.*, 1993], [*Gagnon et al.*, 2006], and Mars [*Landais et al.*, 2015], and the sea surface temperature, [*Lovejoy and Schertzer*, 2013]). We can remark that the universal multifractal index ($\alpha$) is typically fairly close to the log normal value ($\alpha = 2$ ), although due to divergence of moments even when $\alpha = 2$, the statistics of the extremes are power laws, (not log-normal, see the next section on divergence of moments). In the table we also see that with the notable exception of the highly intermittent precipitation field, that the parameter for the intermittency near the mean ($C_1$) might seem small. However, it should be remarked that since the cascades operate over huge ranges of scale, that the resulting fields are nevertheless highly intermittent. In addition, it should be recalled that since $\alpha \approx 2$ the intermittency increases very rapidly for the higher order moments so for example the kurtosis ($q = 4$) has an "effective" intermittency 12 times larger ($K(4) = C_1 (4^2-4) = 12C_1$).



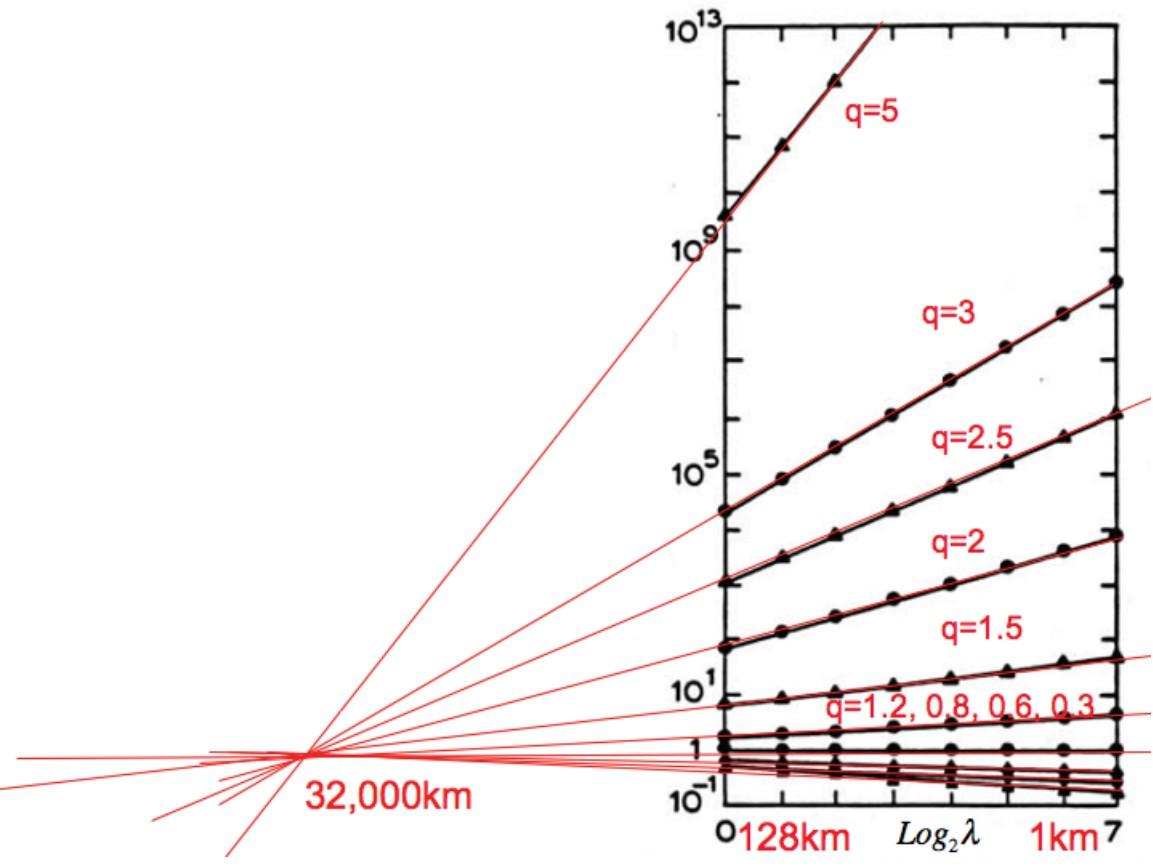

**Fig. 34: The moments** $M_q = \left\langle \varphi_\lambda^q \right\rangle$ **of the normalized radar reflecitivty facto for 70 constant altitude radar maps at 3km altitude from the McGill weather radar (10cm wavelength). As can be seen, the outer scale (where the lines cross) is at roughly 32000 km. This scale is a bit larger than the Earth half- circumference because even at the largest scale there is some precipitation variability due to the interaction of precipitation with the other atmospheric fields. From [Schertzer and Lovejoy, 1987], adapted in [Lovejoy et al., 2008a].**

1175


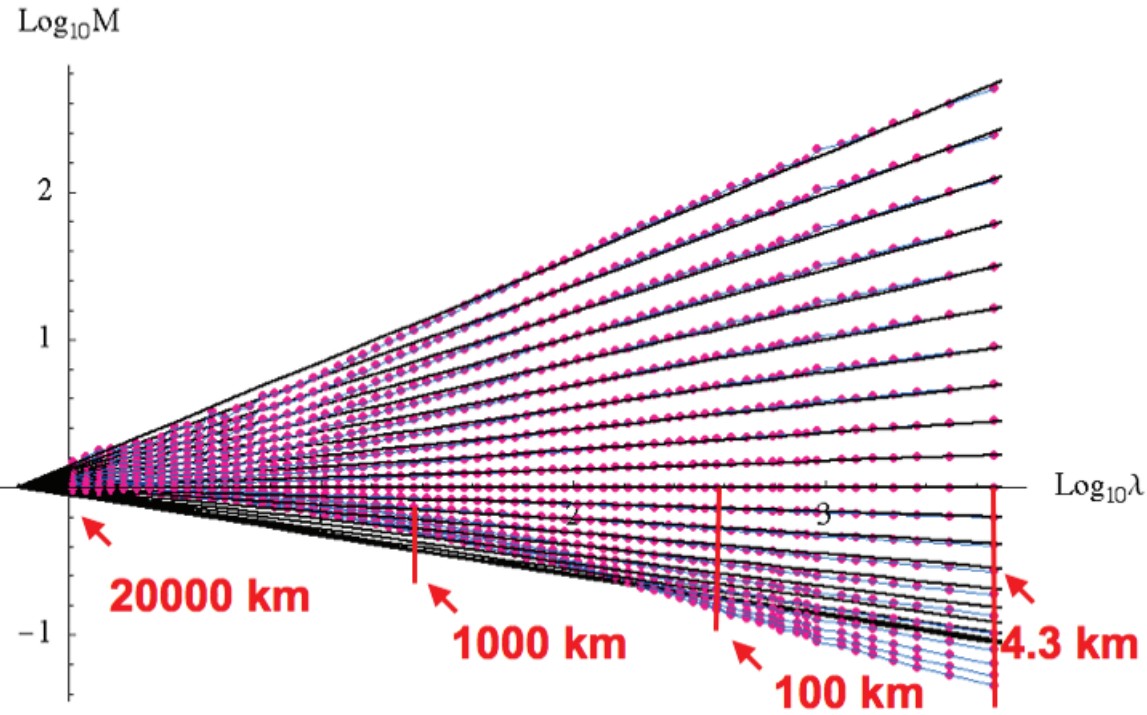

⌐180

**Fig. 35: The same as fig. 32 except for TRMM reflectivities (4.3 km resolution). The moments are for *q* = 0., 0.1, 0.2, …2, taken along the satellite track. The poor scaling (curvature) for the low *q* values (bottom) can be explained as artefacts of the fairly high minimum detectable signal. The reference scale used as a first estimate of the outer cascade scale was  $L_{ref}$ = 20000 km, the outer scale (where the lines cross) was 32000km (as in fig. 31), [*Lovejoy et al.*, 2009e].**

⌐185

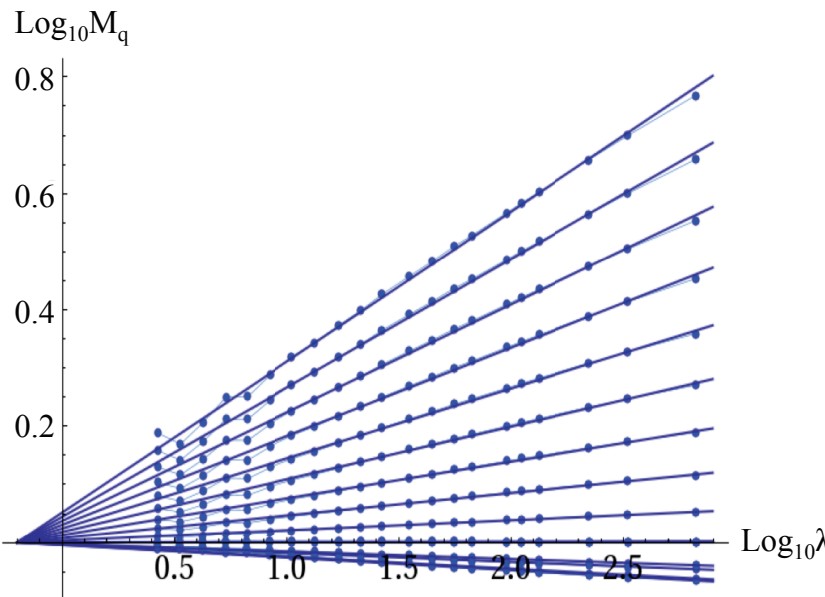

**Fig. 36: Trace moments from** $M_q = \left\langle \varphi_\lambda^q \right\rangle$ **(for** $q$ = 0, 0.1, 0.2, …2.) **for the 1440 hourly geostationary MTSAT data at a 30km resolution, over the region 40ºN to 30ºS covering 130º of longitude over the Pacific ocean.** $L_{ref}$ = 20000km, **the lines cross at an outer scale of 32000km. For the space-time spectra, see fig. 25. Reproduced from [***Pinel et al.***, 2014].**



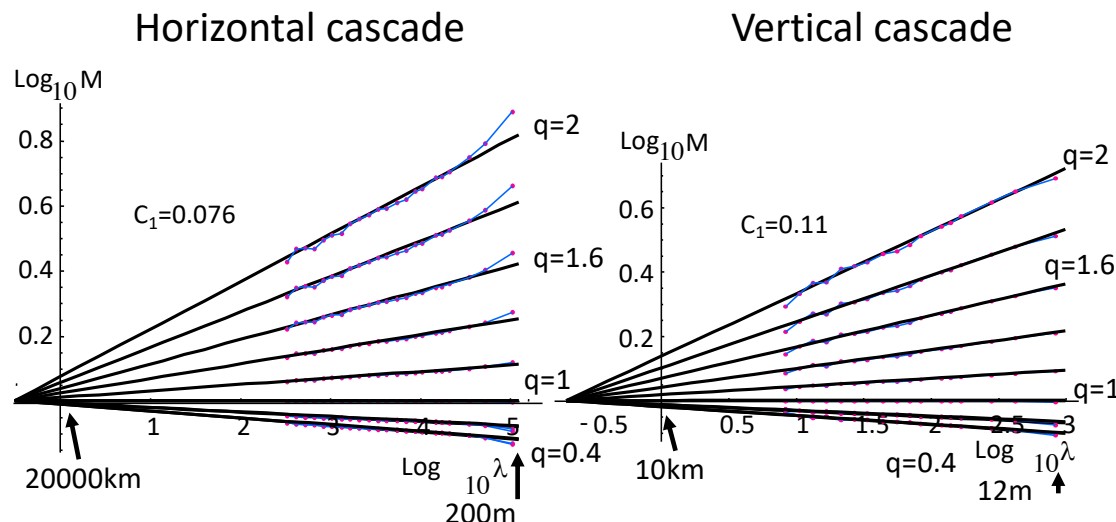

**Fig. 37: The cascade structure of lidar aerosol backscatter, see the example in fig. 45. Moments of normalized fluxes (indicated as** *M***). We show the moments of order** *q* **= 0., 0.2, . 0.4, …2. Notice how the lines converge at effective outer scales that are close to the half circumference (left) and tropopause height (right). Also, the ratio of the intermittency parameters C₁ is ≈ 0.70±0.15 are compatible with the theoretical ratio H$_z$. Reproduced from [***Lovejoy et al.***, 2009c].**



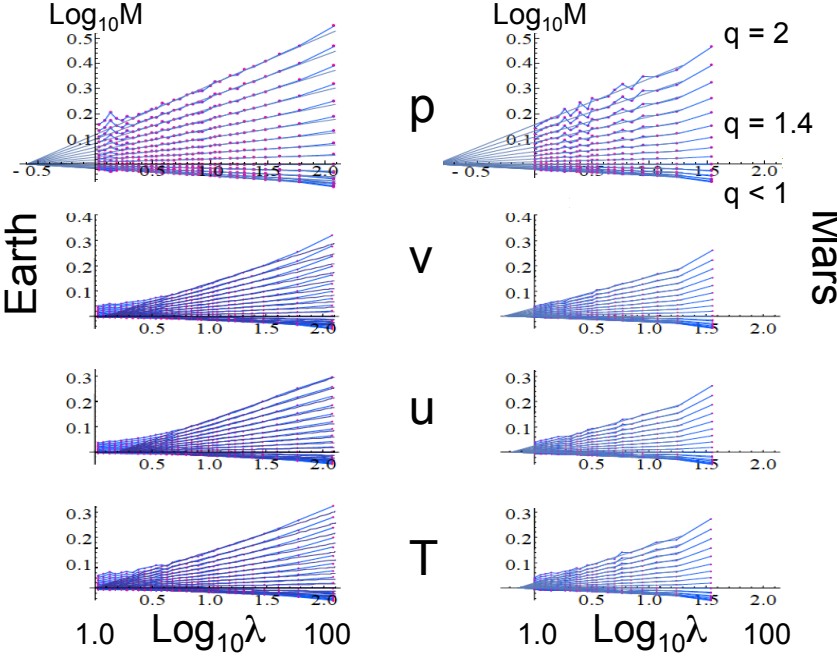

**Fig. 38: A comparison of the scaling of the normalized fluxes ($M_q(\lambda)$) as functions of the scale ratio $\lambda = L_{ref}/\Delta x$ for both Earth (left) and Mars (right) and for surface pressure anomalies ($p$, top), north-south wind ($v$), east-west wind ($u$) and temperature ($T$, bottom). These were estimated from the fluctuations by using eq. 18: . $L_{ref}$ is the half circumference of each planet so that the scale ratio $\lambda$ (denoted $\lambda_{ref}$ in the text) is the inverse of a nondimensional distance. For each individual plot (called "trace moments"), the moments of order $q = 2, 1.9, 1.8… 0.1$ are shown as indicated (upper right). For the Earth the data was at 1º resolution from daily data from the ECMWF interim reanalysis for the year 2006, whereas for Mars, it was from the MACDA (Mars Analysis Correction Data Assimilation) reanalysis at 3.75º resolution over 3 Martian years (roughly 8 terrestrial years). The regression lines are fits to Eq. 4; the slopes are the exponents $K(q)$ and the point of convergence is at the outer scale ratio, it indicates the scale at which the variability starts to build up. In all cases, it is nearly the size of the planet ($\lambda = 1$). Adapted from [*Chen et al.*, 2016].**

### 3.5 Bare and dressed multifractals and Multifractal extremes:

### 3.5.1 Power law tails, divergence of high order moments, multifractal phase transitions

The multifractal process $\varphi_\lambda$ in fig. 27 is shown at various resolutions generated from data at 280m resolution that was then

systemically degraded. How could we model such a process? Let's first consider a multiplicative cascade process by starting

at the bottom and (multiplicatively) adding details as we move to the top. To go from one level (resolution) to the next (i.e.

$\lambda \rightarrow 4\lambda$ in this example), we would use the same rule: pick 4 random multipliers from a unique probability distribution. In

order to prevent the mean tending to either zero or infinity, these multipliers should be normalized so that their mean is constant.

At each step, the spikes would become generally sharper depending only on the lower resolution spikes. At each level ($\lambda$),




the statistics would be characterized by the same $K(q)$ (moments) or $c(\gamma)$ (probability), this is a multifractal cascade, the generic multifractal process.

However, we can already see a problem with this naïve construct. When we reach the top (corresponding to data at 280m resolution), we are still far from the turbulent dissipation scale that is roughly a million times smaller: the top line is better modelled by continuing the cascade down to very small (dissipation) scales and then – imitating the aircraft sensor - averaging the result over 280m. A multifractal process at scale $\lambda$ can thus be produced in two ways: either by a cascade that proceeds over a finite range $\lambda$ and then stops, or alternatively, one that proceeds to very small scales and then is averaged to the same scale. Using renormalization jargon, the former is a "bare" multifractal process whereas the latter – the typical empirical multifractal - is a "dressed" process. What is the difference between the statistics of the bare and dressed resolution $\lambda$ multifractal processes?

Mathematically, we can represent the dressed process as:

$$\varphi_{(d),\lambda} = \lim_{\Lambda \to \infty} \frac{\Pi_\Lambda(B_\lambda)}{vol(B_\lambda)}; \quad \Pi_\Lambda(B_\lambda) = \int_{B_\lambda} \varphi_\Lambda d^D \underline{x}; \quad vol(B_\lambda) = \int_{B_\lambda} d^D \underline{x} \qquad (25)$$

The "flux" $\Pi_\Lambda(B_\lambda)$ and "volume" $vol(B_\lambda)$ are $D$ dimensional measures over a $\lambda$ resolution "ball" $B_\lambda$. In the $D = 1$ dimensional process considered here it is an interval (length $\lambda^{-1}$) for isotropic processes in $D = 2$ or $D = 3$, it is a square or cube (areas $\lambda^{-2}$, volumes $\lambda^{-3}$ respectively, for anisotropic scaling and balls, see section 4). The "$\lambda$ resolution dressed" process $\varphi_{(d),\lambda}$ is thus the small scale cascade limit ($\Lambda \to \infty$) of the $\lambda$ scale average.

A basic result going back to [*Mandelbrot*, 1974] and generalized in [*Schertzer and Lovejoy*, 1985a], [*Schertzer and Lovejoy*, 1987], [*Schertzer and Lovejoy*, 1992], [*Schertzer and Lovejoy*, 1994] shows that the statistical moments are related as:

$$\left\langle \varphi_{(d),\lambda}^q \right\rangle \quad \begin{matrix} \approx \left\langle \varphi_\lambda^q \right\rangle; & q < q_D \\ \to \infty; & q \geq q_D \end{matrix} \qquad (26)$$

i.e. the dressed moments greater than a critical order $q_D$ diverge, but below this, the bare and dressed moments are nearly the same. In terms of the moment scaling exponents:





$$K_d(q) \quad \begin{aligned} &= K(q); \quad q < q_D \\ &= \infty; \quad\quad q \geq q_D \end{aligned}$$

(27)

("$d$" for "dressed").

The critical moment for divergence $q_D$ is the solution of the implicit equation:

1245
$$K(q_D) = D(q_D - 1)$$

(28)

i.e. the $q^{th}$ moment converges if $K(q) < D(q-1)$.

We can now briefly consider the conditions under which there are nontrivial solutions to eq. 28 with finite $q_D$. First, note that $q_D > 1$ since otherwise the process cannot be normalized. Then recall that $K(q)$ is convex ($K'' > 0$) and $K(1) = 0$ (the mean is independent of scale, it is "conserved"). There is therefore a trivial solution at $q = 1$, the solution we require – if it exists – is for $q > 1$. Such solutions $q_D$ to eq. 28 are found at the intersection of $K(q)$ with the line slope $D$ passing through axis at $q = 1$ for $q > 1$.

It is now convenient to define the strictly increasing "dual" codimension function $C(q)$:

$$C(q) = \frac{K(q)}{q-1}$$

(29)

(see fig. 39 for a graphical representation of this relationship). The equation for divergence is now $C(q_D) = D$ and the condition $K'(1) < D$ is $C(1) = K'(1) = C_1 < D$. Since $C(1) < D$ and $C(q)$ is increasing, whenever $C(\infty) > D$ there will be a nontrivial $q_D$. For universal multifractals, this is always the case when the Levy index $\alpha \geq 1$.

To find the corresponding dressed probability exponent $c_d(\gamma)$, we can now take the Legendre transform (eq. 22) of $K_d(q)$:

$$c_d(\gamma) = \quad \begin{aligned} &c(\gamma); &\quad \gamma < \gamma_D \\ &c(\gamma_D) + q_D(\gamma - \gamma_D); &\quad \gamma \geq \gamma_D \end{aligned}$$

(30)




Where $\gamma_D$ is the singularity corresponding to the critical moment $q_D$: $\gamma_D = K'(q_D)$. Finally, with this dressed $c_d(\gamma)$, we

easily find that the tails of the probability distributions are power laws:

$$\Pr(\Delta T > s) \approx s^{-q_D}; \quad s \gg 1 \tag{31}$$

Such tails are sometimes called "fat" or "Pareto". Note that unlike additive processes (e.g. sums of random variables) that generally give rise to stable Levy distributions with divergence order restricted to exponents $q_D < 2$, in these

multiplicative cascade processes, $q_D$ can have any value >1. Finally, since the exponent functions $K(q)$, $c(\gamma)$ have thermodynamic analogues, the discontinuities in the dressed quantities can be theorized as "multifractal phase transitions", of various orders [*Schertzer and Lovejoy*, 1992]). Finally, since no empirical value is infinite – infinite moments occur only in the limit of averages over an infinite number of realizations - the moments $q > q_D$ will be finite but spurious, see the full theory in [*Schertzer and Lovejoy*, 1992], summarized in ch. 5 of [*Lovejoy and Schertzer*, 2013].

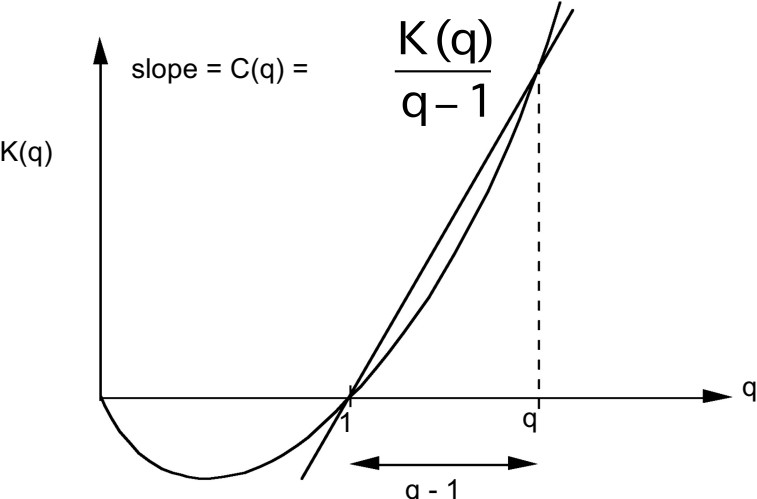

**Fig. 39: A schematic illustration of the relation between *K(q)* and *C(q)*. Reproduced from [*Lovejoy and Schertzer*, 2013].**





### 3.5.2 Power law tails, outliers, black swans and tipping points

To get an idea of how extreme the extremes can be, consider the temperature fluctuations with $q_D = 5$, fig. 40 and Table 2. For a Gaussian, temperature fluctuations 10 times larger than typical fluctuations would be $\approx 10^{23}$ times less likely; if observed, they would be classified as outliers. However, with a power law tail and $q_D = 5$, such extremes occur only $10^5$ times less frequently; so that although rare, they are no longer outliers. In the context of temperatures, understanding the nature of the extremes is fundamental since it determines our interpretation of large events as either extreme - but nevertheless within the expected range, and hence "normal" - or outside this range, hence an "outlier" or perhaps – notably in climate applications - even a "tipping point".

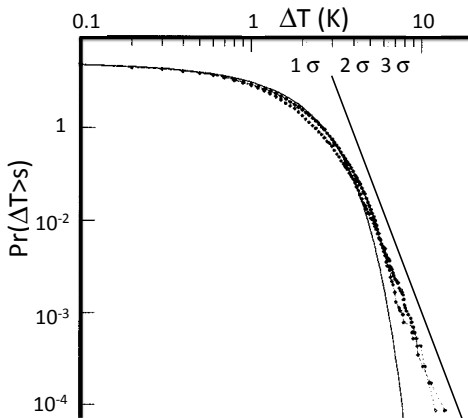

**Fig. 40: The probability distribution of daily temperature differences in daily mean temperatures from Macon France for the period 1949-1979 (10,957 days). Positive and negative differences are shown as separate curves. A best fit Gaussian is shown for reference indicating that the extreme fluctuations correspond to more than 7 standard deviations, for a Gaussian this has a probability of $10^{-20}$. The straight reference line (added) has an absolute slope of $q_D$ with $q_D = 5$. Adapted from [*Ladoy et al.*, 1991].**

A relevant example of the importance of the power law extremes, is global warming. Over about a century, there has been 1°C warming of the globally averaged temperature - a roughly a 5 standard deviation event (with Gaussian probability $\approx 3\times10^{-6}$). In spite of this, climate skeptics claim that it is no more than a Giant Natural Fluctuation (GNF) i.e. a change that might nevertheless be normal - albeit extreme. The relevant extreme centennial changes are indeed non-Gaussian, and bounding the probability tail between power laws with $4<q_D<6$, [*Lovejoy*, 2014] showed that the probability of extremes were enhanced by a factor of as much a factor of 1000. Yet the GNF hypothesis could nevertheless be statistically rejected with more than 99.9% confidence.

There are now numerous atmospheric fields whose extremes have been studied and power tail exponents ($q_D$) estimated. Some of these are shown in table 2 (reproduced from [*Lovejoy and Schertzer*, 2013]) and the wind field is discussed in the next section.





| Field | Data source | type | $q_D$ | reference |
|---|---|---|---|---|
| **Horizontal wind** | Sonic | 10Hz, time | 7.5 | [*Schmitt et al.*, 1994b] |
| | Hot wire probe | Inertial range | 7.7 | [*Radulescu et al.*, 2002] |
| | Hot wire probe | Dissipation range | 5.4 | [*Radulescu et al.*, 2002] |
| | anemometer | 15 minutes | 7 | [*Tchiguirinskaia et al.*, 2006] |
| | anemometer | daily | 7 | [*Tchiguirinskaia et al.*, 2006] |
| | Aircraft, stratosphere | Horizontal, 40m | 5.7 | [*Lovejoy and Schertzer*, 2007a] |
| | Aircraft, troposphere | Horizontal, 280m- 36 km | ≈5 | [*Lovejoy and Schertzer*, 2013] |
| | Aircraft, troposphere | Horizontal, 40m-20 km | ≈7±1 | [*Chigirinskaya et al.*, 1994] |
| | Aircraft, troposphere | Horizontal, 100m | ≈5 | [*Schertzer and Lovejoy*, 1985b] |
| | radiosonde | Vertical, 50m | 5 | [*Schertzer and Lovejoy*, 1985b], [*Lazarev et al.*, 1994] |
| **Potential temperature** | radiosonde | Vertical, 50m | 3.3 | [*Schertzer and Lovejoy*, 1985b] |





| Humidity | Aircraft, troposphere | Horizontal, 280m- 36 km | ≈5 | [*Lovejoy and Schertzer*, 2013] |
|---|---|---|---|---|
| Temperature | Aircraft, troposphere | Horizontal, 280m- 36 km | ≈5 | [*Lovejoy and Schertzer*, 2013] |
| Paleotemperatures | Ice cores | 200 years, time | 4 | [*Lovejoy and Schertzer*, 1986a] |
| Geopotential anomalies | Reanalyses | 500 mb, daily | 2.7 | [*Sardeshmukh and Sura*, 2009] |
| Vorticity anomalies | Reanalyses | 300 mb, daily | 1.7 | [*Sardeshmukh and Sura*, 2009] |
| Seveso pollution | Ground Concentrations | In situ measurements | 2.2 | Salvadori et al., 1993 |
| Chernobyl fallout | Ground Concentrations | In situ measurements | 1.7 | [*Chigirinskaya et al.*, 1998; *Salvadori et al.*, 1993] |

**Table 2: A summary of various estimates of the critical order of divergence of moments ($q_D$) for various meteorological fields (reproduced from [*Lovejoy and Schertzer*, 2013]). The numerous estimates of $q_D$ in precipitation are not included, they are more fully reviewed in [*Lovejoy and Schertzer*, 2013], typical estates are $q_D \approx 3$.**

### 3.5.3 The divergence of high order velocity moments

While the temperature is of fundamental significance for the climate, the wind is the dynamical field, so that it is analogously important at weather scales (as well as in mechanically forced turbulence). For example, numerous statistical models of fully developed turbulence are based on "closure" assumptions that relate high order statistical moments to lower order ones thus allowing the evolution of the statistics in high Reynold's number turbulence to be modelled. Closures thus postulate the finiteness of some (usually all) high order statistical moments of the velocity field.

In fully developed turbulence, in the inertial (scaling) range, ε is conserved by the nonlinear terms in the Navier-Stokes equations, this is the basis of the Kolmogorov law and of cascade theories. In this range, the Kolmogorov law gives $\varepsilon \propto \Delta v^3$. However at small enough (dissipation) scales, where the dynamics are dominated by viscosity, dimensional analysis shows that $\varepsilon \propto \Delta v^2$. For the probability exponents, these relations imply: $q_{D,\varepsilon,IR} = q_{D,v,IR} / 3$ (inertial range), and



$q_{D,\varepsilon,diss} = q_{D,v,diss} / 2$ (dissipation range). Since ε is expected to be constant throughout the inertial range – and in the dissipation range to be equal to the dissipation - we expect $q_{D,\varepsilon,IR} = q_{D,\varepsilon,diss}$ hence the ratio of velocity exponents is

$q_{D,v,diss} / q_{D,v,IR} = 3/2$.

Before discussing this further, let us consider the evidence for the divergence of high order moments in the velocity/wind field. The earliest evidence is shown in fig. 41, ( left), it comes from radiosondes (balloons) measuring the changes in horizontal wind velocity in the vertical direction, [*Schertzer and Lovejoy*, 1985a] found $q_{D,v} \approx 5$ as shown in the plot that extends over layers of thicknesses varying from 50m – 3200m. In the right (from the same paper) in fig. 3.16, probability distributions of ε are shown from aircraft data, with $q_{D,\varepsilon} \approx 1.67 \approx 5/3$, we see that these exponents approximately satisfy the above inertial range theory: $q_{D,\varepsilon,IR} = q_{D,v,IR} / 3$.

These early results had only of the order $10^2$ - $10^3$ measurements which only allows the determination of probabilities down to levels of $10^{-2}$ - $10^{-3}$ , this is barely enough to robustly characterize the exponents. More recent aircraft results with nearly the same horizontal exponent ($q_{D,v} \approx 5.7$) was obtained from another aircraft data set, this time with $\approx 10^6$ data points (fig. 42, right, [*Lovejoy and Schertzer*, 2007b]). Also with $\approx 10^6$ points is probability distribution in the time domain (Fig. 42 left) obtained using a sonic anemometer at 10Hz [*Schmitt et al.*, 1994b]). Here, the exponent is $q_{D,v} \approx 7.5$ i.e. a bit larger than in space.

Results from a much larger sample and from a more controlled laboratory setting (a wind tunnel) also in the temporal domain, are shown in Fig. 43 (data taken by [*Mydlarski and Warhaft*, 1998] analyzed in [*Radulescu et al.*, 2002] and [*Lovejoy and Schertzer*, 2013]). In this case, by placing sensors at varying separations, one can estimate the exponents in both the inertial and dissipation ranges. In the inertial range, the result ($q_{D,vIR} \approx 7.7$) is very close to the earlier temporal result (fig. 43 left, the truncation at large Δv is explained by experimental limitations, [*Lovejoy and Schertzer*, 2013]) whereas in the dissipation range, it has the lower value $q_{D,vdiss} \approx 5.4$. The ratio $q_{D,vdiss} / q_{D,vIR} \approx 1.43$ is very close to the theoretical ratio 3/2 noted above with the value $q_{D,\varepsilon} \approx 2.7$. This good verification of the theoretical result lends credence to the theory and to the reality of the divergence itself.

The previous results from the wind and laboratory turbulence allowed estimates of the probability tails down to levels of only about $10^{-6}$. A more recent result (fig. 44) is about a billion times larger. This is from the largest Direct Numerical Simulation (DNS) to date, using $(2^{13})^3$ discrete volume elements. This high Reynolds number incompressible Navier-Stokes turbulence [*Yeung et al.*, 2015]) allows us to reach much lower probability levels ($p \approx 10^{-15}$). Fig. 43 shows that over $\approx 6$ orders of magnitude, the probability tails of ε (and of enstrophy) have $q_{D,\varepsilon} \approx 5/3$. Surprisingly, although the plot was made explicitly to reveal the nature of extreme fluctuations - and the theory predicting the divergence of moments in turbulence ( [*Mandelbrot*, 1974] , [*Schertzer and Lovejoy*, 1987], eq. 26, 30), this striking power law behaviour of the extremes was not even mentioned by [*Yeung et al.*, 2015]; it was apparently only first noted in [*Lovejoy*, 2019]!

We could note that values $q_{D,\varepsilon} < 2$ imply the divergence of the second moment (i.e. the variance) of $\varepsilon$. This divergence is theoretically significant since - following [*Kolmogorov*, 1962] who proposed a log-normal distribution of $\varepsilon$ (i.e. with all moments finite) - the variance of $\varepsilon$ is regularly used to characterize turbulent intermittency but we now see that due to the divergence, this characterization is problematic. In practice no empirical result is ever infinite. What diverging moments imply is rather that when one attempts to empirically estimate them, that the estimates get larger and larger as the sample size increases. Different experiments can thus readily get quite different results, the parameters are not at all robust.

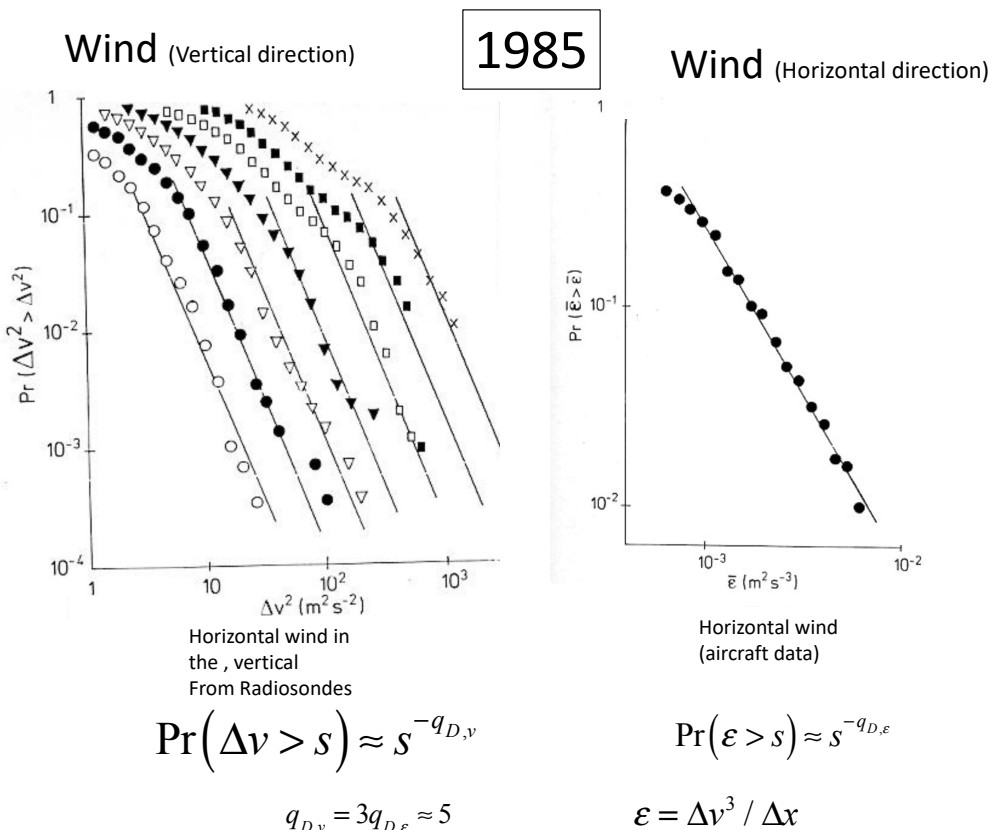

**Fig. 41: The left hand plot shows the probability distribution of the squares of horizontal wind differences in the vertical direction, estimated from radiosondes. The curves from left to right are for layers thickness 50m, 100m, …3200m. The curves straight reference lines have slopes corresponding to $q_D = 5$. The separation of each curve is $2^{Hv}$ with $H_v = 3/5$, the Bolgiano-Obukhov value, see section 4.**

**The plot on the right is for the probability distribution of $\varepsilon$ estimated in the horizontal from aircraft spectra. The straight line has slope -5/3, since $\varepsilon \propto \Delta v^3$ this corresponds to $q_D = 5$ for the wind. Both figures are adapted from [*Schertzer and Lovejoy*, 1985a].**


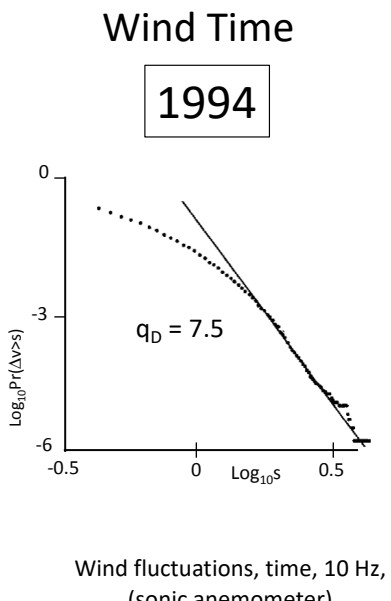

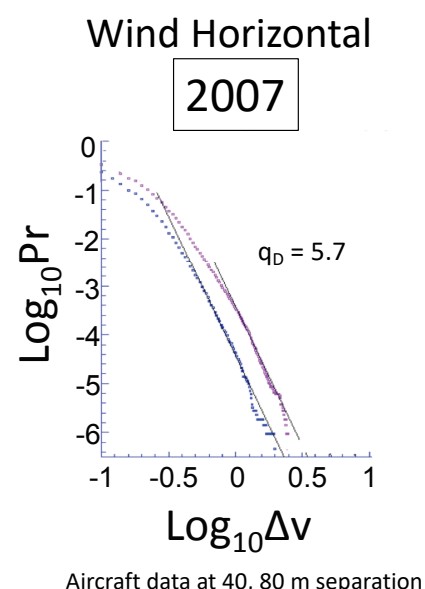

**Fig. 42:** The left hand figure shows the probability distribution of changes $\Delta v$ in the horizontal wind as measured by a sonic anemometer at 10Hz. The reference slope corresponds to $q_D$ =7.5 (adapted from [*Schmitt et al.*, 1994a]).

The figure on the right shows the differences in horizontal wind from 24 aircraft trajectories flying near 12km altitude results are shown for separations of 40 and 80 m, reference slopes corresponding to $q_D$ = 5.7 are shown. Adapted from [*Lovejoy and Schertzer*, 2007b].





1360

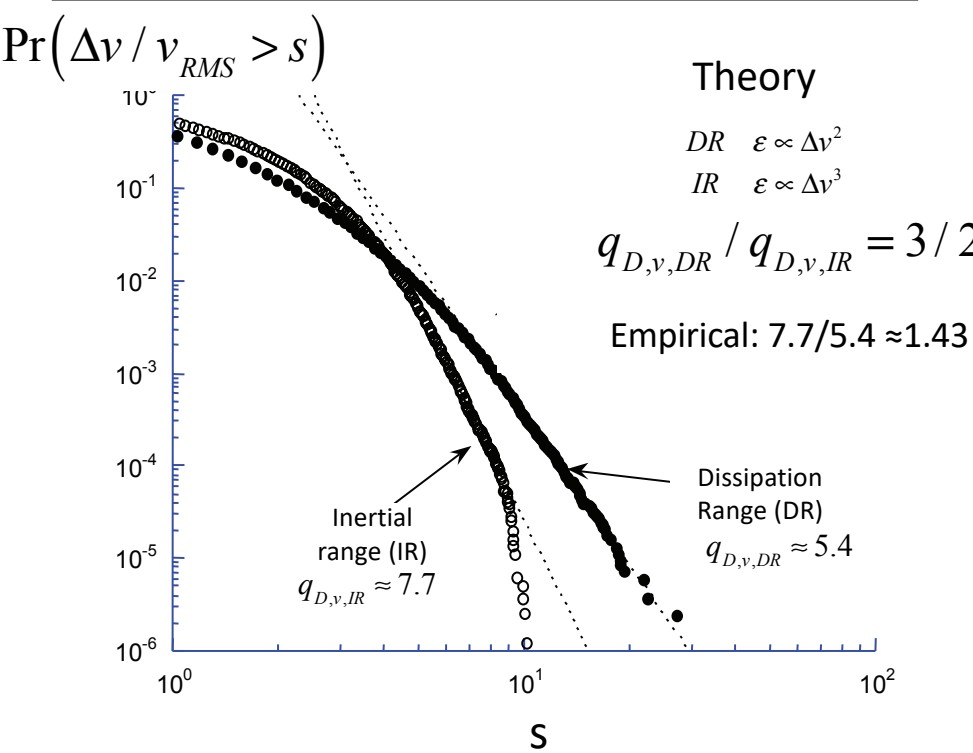

**Fig. 43:** Probability distributions from laboratory turbulence from pairs of anemometers separated by small (dissipation range) distances (DR) and larger, inertial range distances (IR). Slopes corresponding to $q_D$ = 5.4, 7.7 respectively are shown. Their theoretical ratio is 3/2 close to the empirical ratio 1.43. Reproduced from [*Lovejoy and Schertzer*, 2013].

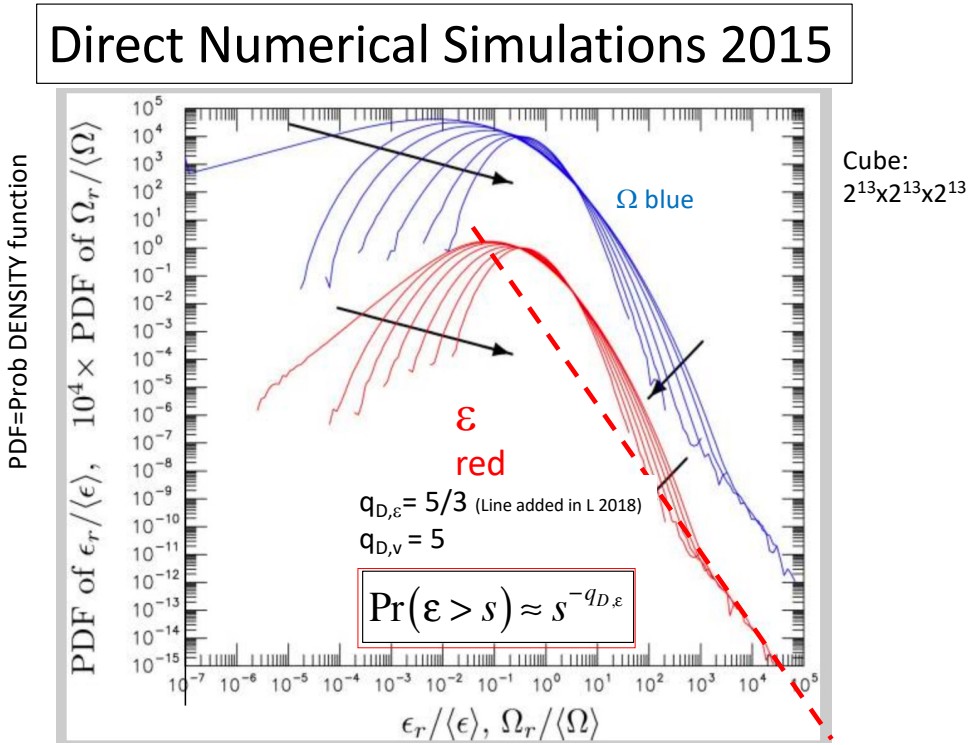

**Fig. 44: Probability distributions of enstrophy (W) and energy (e) fluxes from a large Direct Numerical Simulation of incompressible hydrodynamic turbulence $((2^{13})^3)$, adapted from [Yeung et al., 2015] by adding the dashed reference line corresponding to $q_D = 5$ for the velocity.**

### 3.5.4 Power law probabilities may be more common than we think

In these log-log plots of probability densities, we see that most of the distributions show evidence of log-log linearity near the extremes. When judging possible deviations, it could be recalled due to inadequate instrumental response times, post-processing noise reduction procedures (e.g. smoothing) or via "outlier" elimination algorithms, that extremes can easily be underestimated. Since physically, the extremes are consequences of variability building up over a wide range of spatial scales, we expect that numerical model outputs (including reanalyses) will underestimate the extremes. For example, [*Lovejoy*, 2018]

argued that the models' small hyperviscous scale range (truncated at $\approx 10^5$m rather than at the viscous scale of $\approx 10^{-3}$m), effectively truncates the extreme tails. Any of these effects may explain deviations from perfect power law tails or might explain some of the larger (i.e. less extreme) $q_D$ values in table 2. Finally, while power law probabilities arise naturally in scaling systems, the distributions are not *necessarily* power laws; non-power law (curved tails) may simply correspond to the special cases where $q_D \rightarrow \infty$ (as are the nonintermittent Gaussian special cases).





### 3.5.5 The Noah effect, Black Swans and Tipping points

The power law fluctuations in Figures 41-44 are so large that according to classical assumptions, they would be outliers. In atmospheric science thanks to the scaling, very few processes are Gaussian, extremes occur much more frequently than expected, a fact that colleagues and I regularly underscored starting in the 1980's (see table 2 and for a review, ch. 5 of [Lovejoy and Schertzer, 2013]).

At best, Gaussians can be justified for additive processes, with the added restriction that the variance is finite. However, once this restriction is dropped, we obtain "Levy distributions" with power law extremes, but with exponents $q_D < 2$ (see however [Ditlevsen, 1999]). [Mandelbrot and Wallis, 1968] called the Levy case the "Noah effect" after the Biblical Flood. The Gaussian assumption also fails for the additive but scaling $H$ model [Lovejoy, 2015], [Lovejoy and Mandelbrot, 1985]. Most importantly, Gaussians are irrelevant for multiplicative processes: these generally lead to power law extremes but without any restriction on the value of $q_D$ [Mandelbrot, 1974; Schertzer and Lovejoy, 1987]. Related models include Self-Organized Criticality [Bak et al., 1987] and Correlated Additive and Multiplicative noise [Sardeshmukh and Sura, 2009]. We could also mention that power law distributions also appear as the special (Frechet) case of Generalized Extreme Value Distributions although due to long range statistical dependencies, standard Extreme Value theory does not generally apply to scaling processes.

To underscore the importance of nonclassical extremes, Taleb introduced the terms "grey and black swans" [Taleb, 2010]. Originally, the former designated Levy extremes, and the latter was reserved for extremes that were so strong that they were outliers with respect to any existing theory. However, the term "grey swan" never stuck, and the better-known expression "black swan" is increasingly used for any power law extremes.

All of this is important in climate science where extreme events are often associated with tipping points. The existence of black swan extremes leads to a conundrum: since black swans already lead to exceptionally big extremes, how can we distinguish "mere" black swans from true tipping points?



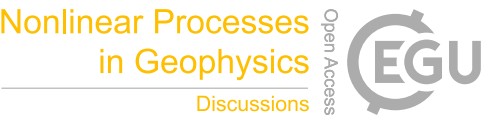

## 4. How Big is a Cloud?  Scaling in two or higher dimensional spaces

### 4.1 Generalized Scale Invariance

#### 4.1.1 Discussion

So far, we have only discussed scaling in 1-D (series and transects), so that the notion of scale itself can be taken simply as an interval (space) or lag (time), and large scales are simply obtained from small ones by multiplying by their scale ratio λ.  But series and transects are only 1-D subspaces of the full ($\underline{r}$,t) space-time where atmospheric processes are defined.   In order to answer the question "how big is a cloud?" - i.e. for more general atmospheric applications - we need to define scale in three dimensional space, and for its evolution, in four dimensional space-time.

The most obvious problem is stratification in the horizontal (see figs.  4,  5).  This is graphically shown in fig. 45 of airborne lidar backscatter from aerosols. At low resolution (bottom), one can see highly stratified layers.  Yet zooming in (top) shows that the layers have small structures that are in fact quite "roundish" and hinting that at even higher resolutions, there might be stratification instead in the vertical. If we determine the spectra in the horizontal and compare with that in the vertical, we obtain fig. 46, the spectra show power laws in both directions but with markedly different exponent.  As shown below, it turns out that the key ratio is:

$$H_z = \frac{\xi_h(2)}{\xi_v(2)} = \frac{\beta_h - 1}{\beta_v - 1} \tag{32}$$

Where "h" indicated horizontal, "v", "vertical", see eq. 9, with the value $H_z = 5/9$  predicted by the 23/9D model (discussed below).  In the figure we see that this prediction is well satisfied by the data.  If the atmosphere is scaling but stratified, then the transects and series must more generally have different exponents ($\xi(q)$, H, K(q)), but for any q, the ratio of horizontal to vertical exponents = $H_z$.

The difference in horizontal and vertical exponents is a consequence of scaling stratification: the squashing of structures with scale.  In the simplest case called "self-affinity", the squashing is along orthogonal directions that are the same everywhere in space – for example along the y axis in an x-y space (e.g. [Varotsos, 2004]). More generally, there is also rotation of structures with scale and the anisotropy depends not only on scale but also on position.  We need more general nonclasssical (non Euclidean) notions of scale and scale change: this is Generalized Scale Invariance (GSI; [*Schertzer and Lovejoy*, 1985c], for a review, see ch. 7 of [Lovejoy and Schertzer, 2013], or for a nontechnical overview, ch. 3 of [*Lovejoy*, 2019]).   Note that the following presentation is based on scale functions, and these can be used to define anisotropic Hausdorff measures, hence providing a (mathematical) measure based definition of size ([*Schertzer and Lovejoy*, 1985c]).



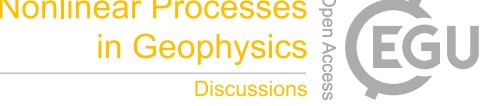

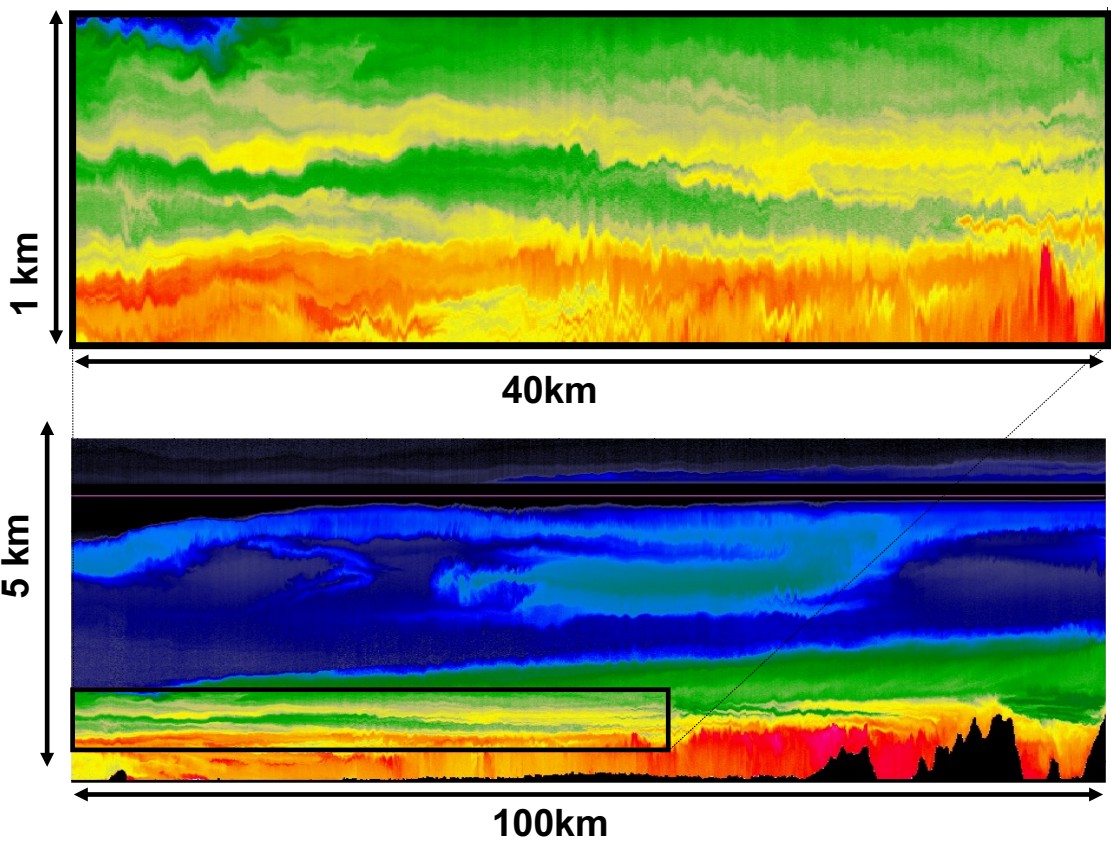

**Fig. 45 Bottom: A vertical section of laser backscatter from aerosols (smog particles) taken by an airborne lidar (laser) flying at 4.5 km altitude (purple line) over British Columbia near Vancouver (the topography shown in black; the lidar shoots two beams, one up and one down), [Lilley et al., 2004]. The resolution is 3 m in the vertical, 96 meters in the horizontal. The above is at a fairly coarse resolution and we mostly see a layered structure.**

**Top: The black box at the lower left is shown blown up in top of the figure. We are now starting to discern vertically aligned and roundish structures. The aspect ratio is about 40:1, reproduced from [Lilley et al., 2004].**

1435


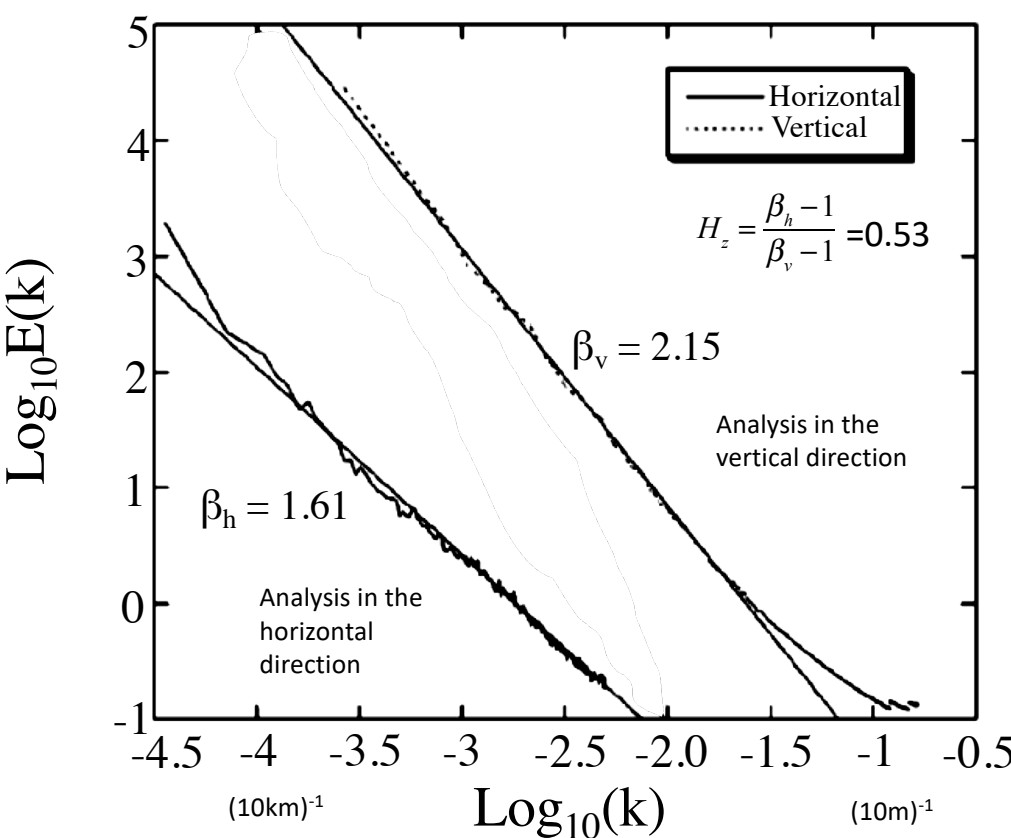

**Fig. 46: The lower curve is the power spectrum for the fluctuations in the lidar backscatter ratio, a surrogate for the aerosol density (*B*) as a function of horizontal wave number *k* (in $m^{-1}$) with a line of best fit with slope $\beta_h$ =1.61. The upper trace is the power spectrum for the fluctuations in *B* as a function of vertical number *k* with a line of best fit with slope $\beta_v$ = 2.15. Adapted from (Lilley et al., 2004).**

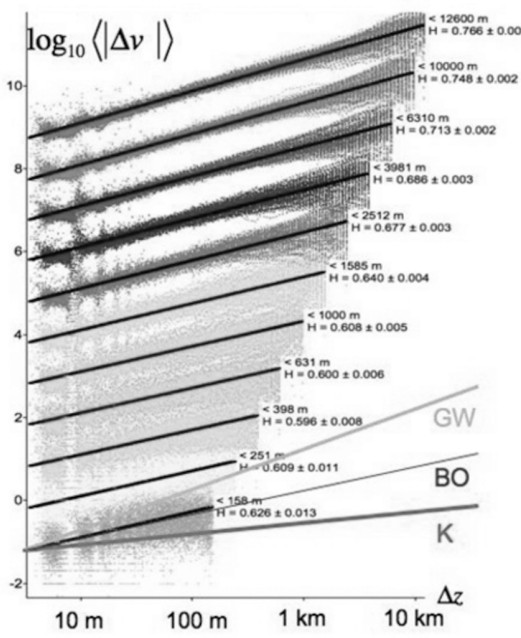

**Fig. 47: The average mean absolute difference in the horizontal wind from 238 drop sondes over the Pacific Ocean taken in 2004. The data were analyzed over regions from the surface to higher and higher altitudes (the different lines from bottom to top, separated by a factor of 10 for clarity). Layers of thickness Δz increasing from 5m to the thicknesses spanning the region were estimated, and lines fit corresponding to power laws with the exponents as indicated. At the bottom reference lines with slopes 1/3 (Kolmogorov, K), 3/5 (Bolgiano-Obhukov, BO), and 1 (Gravity waves, GW and quasi-geostrophic turbulence) are shown for reference.**
**Reproduced from [*Lovejoy et al.*, 2007].**

### 4.1.2 Scale functions and scale changing operators: From self-similarity to self-affinity

To deal with anisotropic scaling, we need an anisotropic definition of the notion of scale itself.

The simplest scaling stratification is called "self-affinity": the squashing is along orthogonal directions whose directions are the same everywhere in space – for example along the $x$ and $z$ axes in an $x$-$z$ space, e.g. a vertical section of the atmosphere or

455 solid earth. More generally, even horizontal sections will not be self-similar: as the scale changes, structures will be both squashed and rotated with scale. A final complication is that the anisotropy can depend not only on scale but also on position. Both cases can be dealt with by using the GSI formalism corresponding respectively to linear (scale only) and nonlinear GSI (scale and position); ([*Lovejoy and Schertzer*, 2013], ch. 7; [*Lovejoy*, 2019], ch. 3).

The problem is to define the notion of scale in a system where there is no characteristic size. Often, the simplest (but usually
unrealistic) "self-similar" system is simply assumed without question: the notion of scale is taken to be isotropic. In this case,





it is sufficient to define the scale of a vector $\underline{r}$ by the usual vector norm (in a vertical section $\underline{r} = (x,y)$, the length of the vector

$\underline{r}$ denoted by $\left| \underline{r} \right| = \left( x^2 + z^2 \right)^{1/2}$ ). $\left| \underline{r} \right|$ satisfies the following elementary scaling rule:

$$\left| \lambda^{-1} \underline{r} \right| = \lambda^{-1} \left| \underline{r} \right| \tag{33}$$

Where again, $\lambda$ is a scale ratio. When $\lambda > 1$, this equation says that the scale (here, length) of the reduced, shrunken vector $\lambda^{-1} \underline{r}$ is simply reduced by the factor $\lambda^{-1}$, a statement that holds for any orientation of $\underline{r}$.

To generalize this, we introduce a scale function $\left\| \underline{r} \right\|$ as well as a more general scale changing operator $T_\lambda$; together they satisfy the analogous equation:

$$\left\| T_\lambda \underline{r} \right\| = \lambda^{-1} \left\| \underline{r} \right\| \tag{34}$$

For the system to be scaling, a reduction by scale ratio $\lambda_1$ followed by a reduction $\lambda_2$ should be equal to first reduction by $\lambda_2$ and then by $\lambda_1$ and both should be equivalent to a single reduction by factor $\lambda = \lambda_1 \lambda_2$. The scale changing operator therefore satisfies multiplicative group properties so $T_\lambda$ is a one parameter Lie group with generator $G$:

$$T_\lambda = \lambda^{-G} \tag{35}$$

When $G$ is the identity operator ($I$), then $T_\lambda \underline{r} = \lambda^{-1} \underline{r} = \lambda^{-1} I \underline{r} = \lambda^{-1} \underline{r}$ so that the scale reduction is the same in all directions (an isotropic reduction): $\left\| \lambda^{-1} \underline{r} \right\| = \lambda^{-1} \left\| \underline{r} \right\|$. However a scale function that is symmetric with respect to such isotropic changes isn't necessarily equal to the usual norm $\left| \underline{r} \right|$ since the vectors with unit scale (i.e. those that satisfy $\left\| \underline{r} \right\| = 1$) may be any (nonconstant, hence anisotropic) function of the polar angle – they are not necessarily circles (2D) or spheres (3D). Indeed, in order to complete the scale function definition, we must specify all the vectors whose scale is unity – the "unit ball". If in addition to $G = I$, the unit scale is a circle (sphere), then the two conditions imply $\left\| \underline{r} \right\| = \left| \underline{r} \right|$ and we recover eq. 33. In the more general – but still linear case where $G$ is a linear operator (a matrix) - $T_\lambda$ depends on scale, but is independent of location. In this case, the qualitative behaviour of the scale functions depends on whether the eigenvalues of $G$ are real or complex. In the former case there is only a finite rotation of structures with scale, the latter, structures rotate and infinite number of times as the scale $\lambda$ goes from 1 to infinity. More generally - nonlinear GSI - $G$ also depends on location and scale, figs. 48 and 52-55 gives some examples of scale functions and figs. 56-66, 69 show some of the corresponding multifractal cloud, simulations. For simulations of the earth's magnetization, rock density, gravity and topography see [*Lavallée et al.*, 1993],[*Pecknold et al.*, 2001] [*Lovejoy and Schertzer*, 2007c; *Lovejoy et al.*, 2005].





### 4.1.3 Scaling stratification

GSI is exploited in modelling and analyzing many atmospheric fields (wind, temperature, humidity, precipitation, cloud density, aerosol concentrations, see [*Lovejoy and Schertzer*, 2013]). To give the idea, we can define the "canonical" scale function for the simplest stratified system representing a vertical $(x,z)$ section in the atmosphere or solid earth:

$$\left\| (x,z) \right\| = l_s \left| \left( \frac{x}{l_s} \right), sign(z) \left| \frac{z}{l_s} \right|^{1/H_z} \right| = l_s \left[ \left( \frac{x}{l_s} \right)^2 + \left| \frac{z}{l_s} \right|^{2/H_z} \right]^{1/2} \tag{36}$$

$H_z$ characterizes the degree of stratification (see below) and $l_s$ is the "sphero-scale", so-called because it defines the scale at which horizontal and vertical extents of structures are equal (although they are generally not exactly circular):

$$\left\| (l_s, 0) \right\| = \left\| (0, l_s) \right\| = l_s \tag{37}$$

It can be seen by inspection that $\left\| (x,z) \right\|$ satisfies:

$$\left\| T_\lambda (x,z) \right\| = \lambda^{-1} \left\| (x,z) \right\|; \qquad T_\lambda = \lambda^{-G}; \quad G = \begin{pmatrix} 1 & 0 \\ 0 & H_z \end{pmatrix} \tag{38}$$

(note that matrix exponentiation is simple only for diagonal matrices - here $T_\lambda = \begin{pmatrix} \lambda^{-1} & 0 \\ 0 & \lambda^{-H_z} \end{pmatrix}$ - but when $G$ is not diagonal it can be calculated by expanding the series: $\lambda^{-G} = e^{-G \log \lambda} = 1 - G \log \lambda + \left( G \log \lambda \right)^2 / 2 - \ldots$, or alternatively by transforming to a diagonal frame). Notice that in this case, the ratios of the horizontal/vertical statistical exponents (i.e. $\xi(q)$, $H$, $K(q)$, $c(\gamma)$) are equal to $H_z$. We could also note that linear transects taken any direction other than horizontal or vertical will have two scaling regimes (with a break near the sphero-scale). However the break is spurious; it is a consequence of using the wrong notion of scale.

Fig. 48 shows some examples of lines of constant scale function defined by eq. 36 with varying $H_z$ values. Successive ellipses are related by the operator $T_\lambda$ with $\lambda = 10^{0.1} = 1.26$ in the illustration. It can be seen that while horizontal scales are changed by a factor $\lambda$, vertical scales are changed by $\lambda^{H_z}$, hence cross-sectional areas are changed by:

$$Areas \propto \lambda^{-D_{el}}; \quad D_{el} = 1 + H_z \tag{39}$$





The exponent $D_{el}$ is called the "elliptical dimension" (although the curves of constant scale are generally only roughly elliptical). Similarly, in three dimensional space, if there are two horizontal directions that scale as $\lambda^{-1}$, and the vertical scales as $\lambda^{-Hz}$, then the elliptical dimension is $D_{el} = 2 + H_z$. Fig. 11 shows a schematic of various models of the atmosphere, with the both the classical 2D isotropic (totally stratified, flat) large scale model at one extreme and the 3D isotropic model at the other and the more realistic $D_{el} = 23/9D$ model discussed below. Interestingly, in the atmosphere – although highly variable - $l_s$ is typically small (meters to hundreds of meters) but $H_z < 1$ (close to the middle top set of curves in fig. 48). In contrast, in the solid earth, $l_s$ is very large (probably larger than the planet scale) but $H_z > 1$ (probably $\approx 2$, close to the bottom right curves, see [*Lovejoy and Schertzer*, 2007c] for a review). In the former case, the stratification becomes stronger at larger and larger scales whereas in the latter, it is stronger at smaller scales.

Equipped with a scale function, the general anisotropic generalization of the 1-D scaling law (eq. 12) may now be expressed by using the scale $\left\lVert \underline{\Delta r} \right\rVert$:

$$\Delta T \left( \underline{\Delta r} \right) \overset{d}{=} \phi_{\left\lVert \underline{\Delta r} \right\rVert} \left\lVert \underline{\Delta r} \right\rVert^{H} \tag{40}$$

This shows that the full scaling model or full characterization of scaling requires the specification of the notion of scale via the scale invariant generator $G$ and unit ball, (hence the scale function), the fluctuation exponent $H$, as well the statistics of $\phi_{\left\lVert \underline{\Delta r} \right\rVert}$ specified via $K(q)$, $c(\gamma)$ or – for universal multifractals, $C_1$, $\alpha$. In many practical cases – e.g. vertical stratification – the direction of the anisotropy is fairly obvious, but in horizontal sections, where there can easily be significant rotation of structures with scale, the empirical determination of $G$ and the scale function is a difficult, generally unsolved problem.





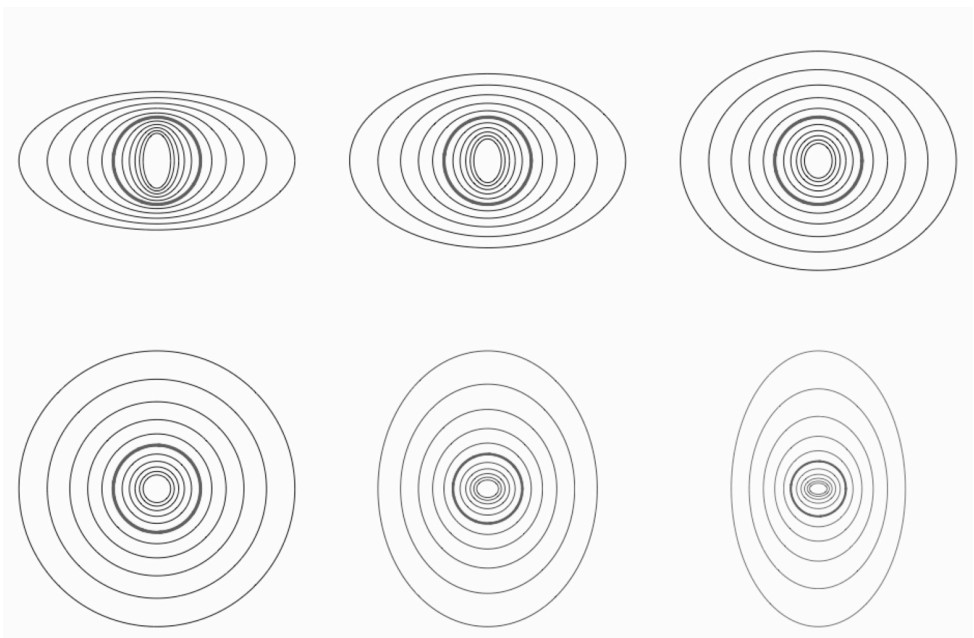

**Figure 48: A series of ellipses each separated by a factor of 1.26 in scale, red indicating the unit scale (here, a circle, thick lines).**
**Upper left to lower right, $H_z$ increasing from 2/5, 3/5, 4/5 (top), 1, 6/5, 7/5 (bottom, left to right). Note that when $H_z>1$, the stratification at large scales is in the vertical rather than the horizontal direction (this is required for modelling the earth's geological strata). Reproduced from [*Lovejoy*, 2019].**

### 4.1.4 The 23/9D model

Kolmogorov theory was mostly used to understand laboratory hydrodynamic turbulence which is mechanically driven and can be made approximately isotropic (unstratified) by the use of either passive or active grids. In this case, fluctuations in $\Delta v$ for points separated by $\underline{\Delta r}$ can be determined essentially via dimensional analysis using $\varepsilon$; the latter choice being justified since it is a scale by scale conserved turbulent flux. The atmosphere however is fundamentally driven by solar energy fluxes which create buoyancy inhomogeneities; in addition to energy fluxes, buoyancy is also fundamental. In order to understand
atmospheric dynamics we must therefore determine which additional dimensional quantities are introduced by gravity/ buoyancy. As discussed in [*Monin and Yaglom*, 1975], this is necessary for a more complete dimensional analysis.

In addition to the dynamical equations with quadratic invariant $\varepsilon$ – the only dimensional quantity pertinent in the inertial range in isotropic turbulence – we must consider the thermodynamic energy equation for the potential temperature $\theta$ (e.g. [*Lesieur*, 1987]). Analysis shows that the $v$ and $\theta$ fields are only coupled by the $\Delta f$ buoyancy forcing term:





$$f = g \log \theta \qquad (41)$$

Where $g$ is the acceleration of gravity. $f$ is therefore the fundamental physical and dimensional quantity rather than $\theta$.

The classical way of dealing with buoyancy is to use the Boussinesq approximation, i.e. to assume the existence of a scale separation and then define density (and hence buoyancy) perturbations about an otherwise perfectly stratified "background" flow. This leads to the classical isotropic buoyancy subrange turbulence discovered independently by [*Bolgiano*, 1959] and [*Obukhov*, 1959]. Unfortunately, it was postulated to be an *isotropic* range, yet it was never observed in the horizontal. Therefore, by the time it was later observed in the vertical, it had either been forgotten [*Endlich et al.*, 1969], or it's significance was not appreciated ([*Adelfang*, 1971]) and it was subsequently largely ignored.

Yet if there is wide range atmospheric scaling, then there is no scale separation and so that (as outlined in ch. 6 in [*Lovejoy and Schertzer*, 2013]), we can make a more physically based argument which is analogous to that used for deriving passive scalar variance cascades in passive scalar advection - the Corrsin-Obhukhov law, [*Corrsin*, 1951], [*Obukhov*, 1949], (itself analogous to the energy flux cascades that lead to the Kolmogorov law).

If we neglect dissipation and forcing, then $Df / Dt = 0$ where $D/Dt$ is the advective derivative) so that $f$ obeys a passive scalar advection equation and therefore the corresponding buoyancy force variance flux:

$$\varphi = \frac{\partial f^2}{\partial t} \qquad (42)$$

is conserved by the nonlinear terms. In this case, the only quantities available for dimensional analysis are $\varepsilon$ (units $m^2/s^3$) and $\varphi$ (units $m^2/s^5$). In this approach, there is no separation between a stratified "background" state and a possibly isotropic fluctuation field so that there is no rationale for assuming that the either the $\varphi$ or $\varepsilon$ cascades are associated with any isotropic regimes. Indeed, following [*Schertzer and Lovejoy*, 1983] and [*Schertzer and Lovejoy*, 1985b] it is more logical to assume that the two basic turbulent fluxes $\varepsilon$, $\varphi$ can co-exist and cascade over a single scale wide range regime with the former dominating in the horizontal, the latter in the vertical:

$$\Delta v(\Delta x) = \phi_h \Delta x^{H_h}; \quad \phi_h = \varepsilon^{1/3}; \quad H_h = 1/3$$
$$\Delta v(\Delta z) = \phi_v \Delta x^{H_v}; \quad \phi_v = \varphi^{1/5}; \quad H_v = 3/5 \qquad (43)$$

where $\Delta x$ is a horizontal and $\Delta z$ a vertical lag (for the moment we ignore the other horizontal coordinate $y$). Dimensionally, the fluxes $\varepsilon$, $\varphi$ define a unique length scale $l_s$:

$$l_s = \left( \frac{\phi_h}{\phi_v} \right)^{1/(H_v - H_h)} = \frac{\varepsilon^{5/4}}{\varphi^{3/4}} \qquad (44)$$



Fig. 47 shows that in the vertical the Bolgiano-Obhukov law holds quite well - especially near the surface, but at all altitudes, it is much better respected than the isotropic Kolmogorov law ($H_v$ = 1/3 or the alternative laws from quasi-linear gravity wave or quasi-geostrophic turbulence that give $H_v$ = 1, [*Lovejoy et al.*, 2007].

We can see that the two laws in eq. 43 are special cases of the more general anisotropic scaling law eq. 40 since for pure horizontal displacements ($\Delta z$ = 0) and pure vertical displacements, ($\Delta x$ = 0), eq. 40 yields:

$$\Delta f\left(\Delta x,0\right)\overset{d}{=}\phi\Delta x^{H}=\varepsilon^{1/3}\Delta x^{1/3}; \qquad \left\|\left(\Delta x,0\right)\right\|=\Delta x$$
$$\Delta f\left(0,\Delta z\right)\overset{d}{=}\phi l_{s}^{H_{z}-1}\Delta z^{H/H_{z}}=\varphi^{1/5}\Delta z^{3/5}; \qquad \left\|\left(0,\Delta z\right)\right\|=l_{s}\left|\frac{\Delta z}{l_{s}}\right|^{1/H_{z}}$$
(45)

i.e. if we identify $H = H_h$ =1/3 and $H_z = H_h/H_v$ =5/9, $\phi = \varepsilon^{1/3}$, we see that eq. 40 and 45 are equivalent. If in addition, we assume that the two horizontal directions are equivalent, we obtain $D_{el}$=1+1+$H_z$ = 23/9 = 2.555... hence the name the "23/9D model", see fig. 11 for a schematic.

**4.1.5 Scaling Space-time, Fourier space GSI**

In section 2.6, 2.7 we mentioned that for Lagrangian frame temporal velocity fluctuations we should use the size-lifetime relation that is implicit in the horizontal Kolmogorov law. If we assume horizontal isotropy then for velocity fluctuations, we have:

$$\Delta v\left(\Delta x\right)=\varepsilon^{1/3}\Delta x^{1/3}$$
$$\Delta v\left(\Delta y\right)=\varepsilon^{1/3}\Delta y^{1/3}$$
$$\Delta v\left(\Delta z\right)=\phi^{1/5}\Delta z^{3/5}$$
$$\Delta v\left(\Delta t\right)=\varepsilon^{1/2}\Delta t^{1/2}$$
(46)

Following the developments in the previous subsection (eqs. 40, 43), we can express the full space-time scaling (eqs. 46) in a single expression valid for any space-time vector displacement $\underline{\Delta R}=\left(\underline{\Delta r},\Delta t\right)=\left(\Delta x,\Delta y,\Delta z,\Delta t\right)$ by introducing a scalar function of space-time vectors called the "(space-time) scale function", denoted $\left[\kern-0.15em\left[\underline{\Delta R}\right]\kern-0.15em\right]$, which satisfies the fundamental (functional) scale equation:




$$\left[\!\left[\lambda^{-G_{st}}\underline{\Delta R}\right]\!\right] = \lambda^{-1}\left[\!\left[\underline{\Delta R}\right]\!\right]; \quad G_{st} = \begin{pmatrix} G_s & 0 \\ 0 & H_\tau \end{pmatrix}; \quad H_\tau = (1/3)/(1/2) = 2/3$$

(47)

where $G_s$ is the 3X3 matrix spatial generator:

$$G_s = \begin{pmatrix} 1 & 0 & 0 \\ 0 & 1 & 0 \\ 0 & 0 & H_z \end{pmatrix}$$

(48)

(with rows corresponding to $(x,y,z)$; and the 4x4 matrix $G_{st}$ is the extension to space-time. We have introduced the notation "

$\left[\!\left[\quad\right]\!\right]$," for the space-time scale function in order to distinguish it from the purely spatial scale function denoted "$\|\quad\|$".

Using the space-time scale function, we may now write the space-time generalization of the Kolmogorov law as:

$$\Delta v(\underline{\Delta R}) = \varepsilon_{\left[\!\left[\Delta R\right]\!\right]}^{1/3}\left[\!\left[\underline{\Delta R}\right]\!\right]^{1/3}$$

(49)

where the subscripts on the flux indicate the space-time scale over which ε is averaged. This anisotropic intermittent (multifractal) generalization of the Kolmogorov law is thus one of the key emergent laws of atmospheric dynamics and serves as a prototype for the emergent laws governing the other fields.

The result analogous to that of the previous subsection, the corresponding simple ("canonical") space-time scale function is:

$$\left[\!\left[\underline{\Delta R}\right]\!\right]_{can} = l_s\left(\left(\frac{\|\underline{\Delta r}\|}{l_s}\right)^2 + \left(\frac{|\Delta t|}{\tau_s}\right)^{2/H_\tau}\right)^{1/2}$$

1595                                                                                          (50)

Where $\tau_s = \phi^{-1/2}\varepsilon^{1/2}$ is the "sphero-time" analogous to the sphero-scale $l_s = \phi^{-3/4}\varepsilon^{5/4}$ (see also [*Marsan et al.*, 1996]). With scale function (eq. 50), the fluctuations (eq. 49) respect eqs. 46.

We now seek to express the generalized Kolmogorov law in a Eulerian frame. The first step is to consider the effects on the scale function of an overall advection. We then consider statistical averaging over turbulent advection velocities.

Advection can be taken into account using the Gallilean transformation $\underline{r} \rightarrow \underline{r} - \underline{v}t$, $t \rightarrow t$ which corresponds to the following matrix $A$:




$$A = \begin{pmatrix} 1 & 0 & 0 & u \\ 0 & 1 & 0 & v \\ 0 & 0 & 1 & w \\ 0 & 0 & 0 & 1 \end{pmatrix}$$

(51)

where the mean wind vector has components: $\underline{v} = (u,v,w)$, [*Schertzer et al.*, 1997b] and the columns and row correspond to

$x,y,z,t$. The new "advected" generator is $G_{st,advec} = A^{-1} G_{st} A$ and the scale function $[\![\underline{\Delta R}]\!]_{advec}$ which is symmetric with

respect to $G_{st,advec}$ is: $[\![\underline{\Delta R}]\!]_{advec} = [\![A^{-1} \underline{\Delta R}]\!]$. The canonical advected scale function is therefore:

$$[\![\underline{\Delta R}]\!]_{advec,can} = [\![A^{-1} \underline{\Delta R}]\!]_{can} = l_s \left( \left( \frac{\Delta x - u\Delta t}{l_s} \right)^2 + \left( \frac{\Delta y - v\Delta t}{l_s} \right)^2 + \left( \frac{\Delta z - w\Delta t}{l_s} \right)^{2/H_z} + \left( \frac{\Delta t}{\tau_s} \right)^{2/H_\tau} \right)^{1/2}$$

(52)

Note that since $D_{st,advec} = TrG_{st,advec} = Tr\left( A^{-1} G_{st} A \right) = TrG_{st} = D_{st}$, such constant advection does not affect the elliptical

dimension ("$Tr$" indicates "trace"; see however below for the "effective" $G_{eff}$, $D_{eff}$).

It will be useful to study the statistics in Fourier space; for this purpose we can use the result (e.g. ch. 6 of [*Lovejoy and*

*Schertzer*, 2013]) that the Fourier generator $\tilde{G} = G^T$ so that:

$$\tilde{G}_{st,advec} = A^T G_{st}^T \left( A^{-1} \right)^T$$

(53)

The corresponding canonical dimensional Fourier space scale function is therefore:

$$[\![\underline{K}]\!]_{advec,can} = [\![A^T \underline{K}]\!]_{can} = l_s^{-1} \left( (k_x l_s)^2 + (k_y l_s)^2 + (k_z l_s)^{2/H_z} + \left( \tau_s (\omega + \underline{k} \cdot \underline{v}) \right)^{2/H_t} \right)^{1/2}$$


(54)

In other words the real space Gallilean transformation $\underline{r} \to \underline{r} - \underline{v}t; \quad t \to t$ corresponds to the Fourier space

transformation $\underline{k} \to \underline{k}; \quad \omega \to \omega + \underline{k} \cdot \underline{v}$ (this is a well-known result, notably used in atmospheric waves).

The above results are for a deterministic advection velocity whereas in reality, the advection is turbulent. Even if we consider

a flow with zero imposed mean horizontal velocity (as argued by [*Tennekes*, 1975]) in a scaling turbulent regime with $\Delta v_l \approx$

$\varepsilon^{1/3} l^{1/3}$, the typical largest eddy; the "weather scale" $L_w$ will be the scale of the earth ($\approx L_e$) and it will have a mean velocity $V_w$

$\approx \Delta v_w \approx \varepsilon_w^{1/3} L_w^{1/3}$ and will survive for the corresponding eddy turn over time $\tau_{eddy} = \tau_w = L_w/V_w = \varepsilon_w^{-1/3} L_w^{2/3}$ estimated as $\approx 10$




days above. In other words, if there is no break in the scaling then we expect that smaller structures will be advected by the largest structures in the scaling regime.

The statistics of the intensity gradients of real fields are influenced by random turbulent velocity fields and involve powers of

such scale functions but with appropriate "average" velocities. In this case, considering only the horizontal and time, we introduce the nondimensional variables (denoted by a circumflex " $\hat{}$ "):

$$\widehat{\Delta x} = \frac{\Delta x}{L_w}; \quad \widehat{\Delta y} = \frac{\Delta y}{aL_w}; \quad \widehat{\Delta t} = \frac{\Delta t}{\tau_w}; \quad \hat{v}_x = \mu_x = \frac{\overline{v}_x}{V_w}; \quad \hat{v}_y = \mu_y = \frac{\overline{v}_y}{V_w}$$

(55)

the symbols $\mu_x$, $\mu_y$ are used for the components of the nondimensional velocity; they are less cumbersome than $\hat{v}_x$, $\hat{v}_y$ where:

$$V_w = \left(\overline{v_x^2} + a^2 \overline{v_y^2}\right)^{1/2}; \quad \tau_w = \frac{L_w}{V_w}$$


(56)

Note that here $V_w$ is a large-scale turbulent velocity whereas $\overline{v}_x$, $\overline{v}_y$ are given by the overall mean advection in the region of interest and $\mu_x < 1$, $\mu_y < 1$ (since $\overline{v^2} > \left(\overline{v}\right)^2$ ). The use of the averages (indicated by the overbars) is only totally justified if the second power of the scale function is averaged; presumably, it is some other power that is physically more relevant and there will thus be (presumably small) intermittency corrections (which we ignore). It is now convenient to define:

$$\underline{\mu} = \left(\mu_x, \mu_y\right) \quad \left|\underline{\mu}\right|^2 = \mu_x^2 + \mu_y^2$$


(57)

which satisfies $\left|\underline{\mu}\right| < 1$. In terms of the nondimensional quantities this yields an "effective" nondimensional scale function:

$$\left\lVert \widehat{\underline{\Delta R}} \right\rVert_{eff} = \left| C \widehat{\underline{\Delta R}} \right|$$

(58)

Where the matrix $C$ is given in eq. 11, its rows and columns correspond to $x, y, t$ (left to right, top to bottom, and the vertical bars indicate the usual isotropic vector norm). The effective scale function in eq. 58 is only "trivially anisotropic" since it is

scaling with respect to an "effective" $G$ matrix $G_{eff} = I =$ the identity, the matrix $C$ simply determines the trivial space-time anisotropy.





As discussed in [*Lovejoy and Schertzer*, 2013], the above real space scale function is needed to interpret "satellite winds" (deduced from times series of satellite cloud images), and in section 2.7 the Fourier equivalent of equation 58 (based on the inverse matrix $C^{-1}$ and Fourier scale function $\left\|\underline{K}\right\|_{eff} = \left| C^{-1} \underline{K} \right|$, see fig. 25 and eq. 11). It was extensively empirically tested in [*Pinel et al.*, 2014] where the full parameters of the $\underline{C}$ matrix were estimated.

## 4.2 Empirical testing of the 23/9D model

### 4.2.1 Testing the 23/9D model with aircraft wind data

The first experimental measurement of the joint ($\Delta x$, $\Delta z$) structure function of the horizontal velocity was made possible by the development of aircraft navigation systems with highly accurate "TAMDAR" GPS based vertical positioning system, ([*Pinel et al.*, 2012], fig. 49). High vertical accuracy is needed to distinguish aircraft flying on isobars and from those flying on isoheights. The problem with earlier aircraft velocity spectra - going back to the famous and still cited [*Nastrom and Gage*, 1983] analysis - is that the aircraft fly on isobars, and these were gently sloping. As pointed out in [*Lovejoy et al.*, 2009a], at some critical horizontal distance (that depended on the various fluxes ε, φ and the slope of the isobar), the vertical (Bolgiano-Obukhov) statistics ($\Delta z^{3/5}$) begin to dominate over the horizontal Kolmogorov statistics ($\Delta x^{1/3}$). In the spectral domain this implies a transition from $E(k) \approx k^{-5/3}$ to $k^{-11/5}$ (using β $=1+2H$, i.e. ignoring intermittency). The history of aircraft wind spectra – in particular the multidecadal (and continuing) attempts to shoe-horn the spurious horizontal Bolgiano-Obhkov spectra into a $k^{-3}$ regime (in accord with quasi-geostrophic turbulence theory) is thoroughly reviewed in ch. 2 of [*Lovejoy and Schertzer*, 2013], appendix B of [*Lovejoy*, 2022b].



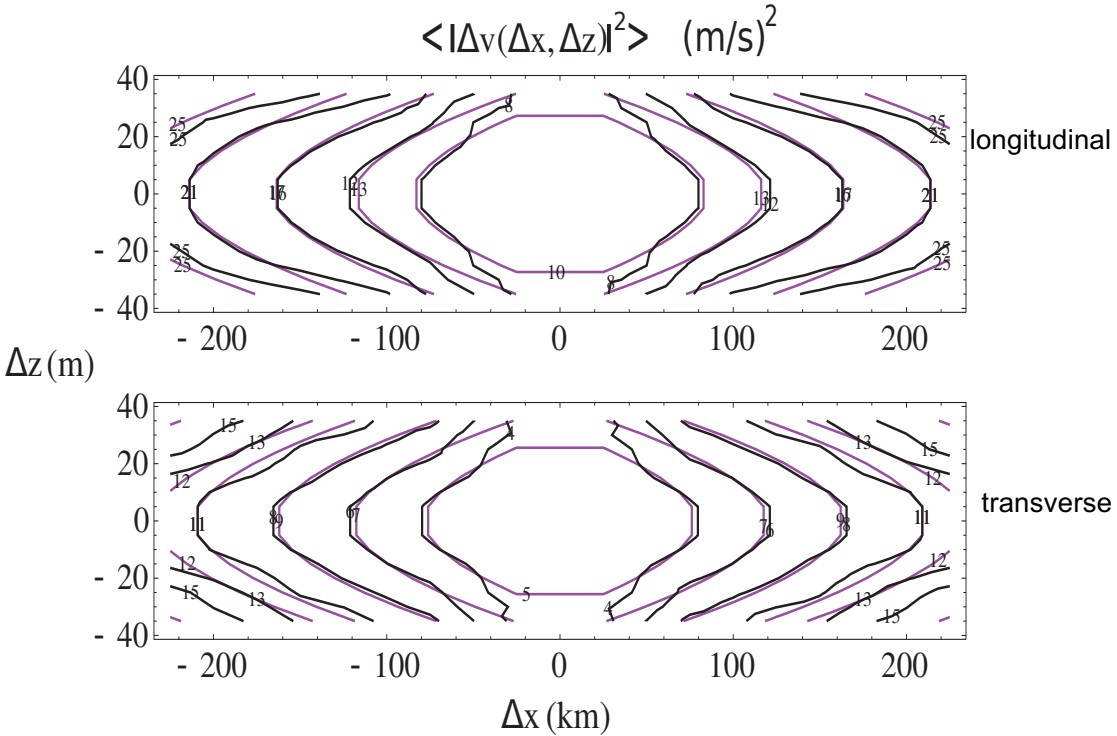

**Fig. 49: A contour plot of the mean squared transverse (top) and longitudinal (bottom) components of the wind as estimated from a year's (≈14500) TAMDAR flights, 484000 wind difference measurements (i.e. the second order structure function from difference fluctuations) data from flight legs between 5 and 5.5km. $H_z$ was estimated as 0.57±0.02. Black shows the empirical contours, purple the theoretical contours with $H_z$ = 5/9. Reproduced from [*Pinel et al.*, 2012].**

To illustrate what the 23/9D model implies for the atmosphere, we can make multifractal simulations of passive scalar clouds, these were already discussed in fig. 5 that showed that in general, scaling leads to morphologies, structures that change with scale even though there is no characteristic scale involved. Figure 5 compares a zoom into an isotropic (self-similar) multifractal cloud (left) and into a vertical section of a stratified cloud with 23/9D. Whereas zooming into the self-similar cloud yields similar looking cross sections at all scales, zooming into the 23/9D cloud at the right of fig. 5 displays continuously varying morphologies. We see that at the largest scale (top), the cloud is in fairly flat strata, however as we zoom in, we eventually obtain roundish structures (at the sphero-scale), and then at the very bottom, we see vertically oriented filaments forming indicating stratification in the vertical direction (compare this with the lidar data, fig. 45).





### 4.2.2 Testing the 23/9D model with cloud data

The anisotropic stratification and elliptical dimension of rain areas (as determined by radar) goes back to [*Lovejoy et al.*, 1987], and with much more vertical resolution, from CloudSat, a satellite borne radar, fig. 50 (see the sample CloudSat image, fig. 4). From the figure, we see the mean relation between horizontal and vertical extent of clouds is very close to the predictions of the 23/9D theory, with sphero-scale (averaged over 16 orbits) of about 100m. The figure also shows that there is fair amount of variability (as expected since the sphero-scale is a ratio of powers of highly variable turbulent fluxes, eq. 44). Fig. 51 shows

the implications for typical cross-sections. The stratification varies considerably as a function of the sphero-scale (and hence buoyancy and energy fluxes).

Finally we can compare the CloudSat estimates with those of other atmospheric fields (table 3). The estimates for $T$ (temperature), $\log\theta$ (log potential temperature), $h$ (humidity) are from comparing aircraft and drop sonde exponents. These are inherently less accurate than using vertical sections see the right hand three columns. Overall, we find excellent agreement

with 23/9 theory. Recall that the leading alternative theory - quasi-geostrophic turbulence - has $H_z = 1$ (for small scales, the isotropic 3D turbulence value) or $H_z = 1/3$ for (large scales, the isotropic 2D turbulence value), it is clear that these can be eliminated with high degrees of confidence – both are at least 10 standard deviations from the more accurate estimates in table 3.




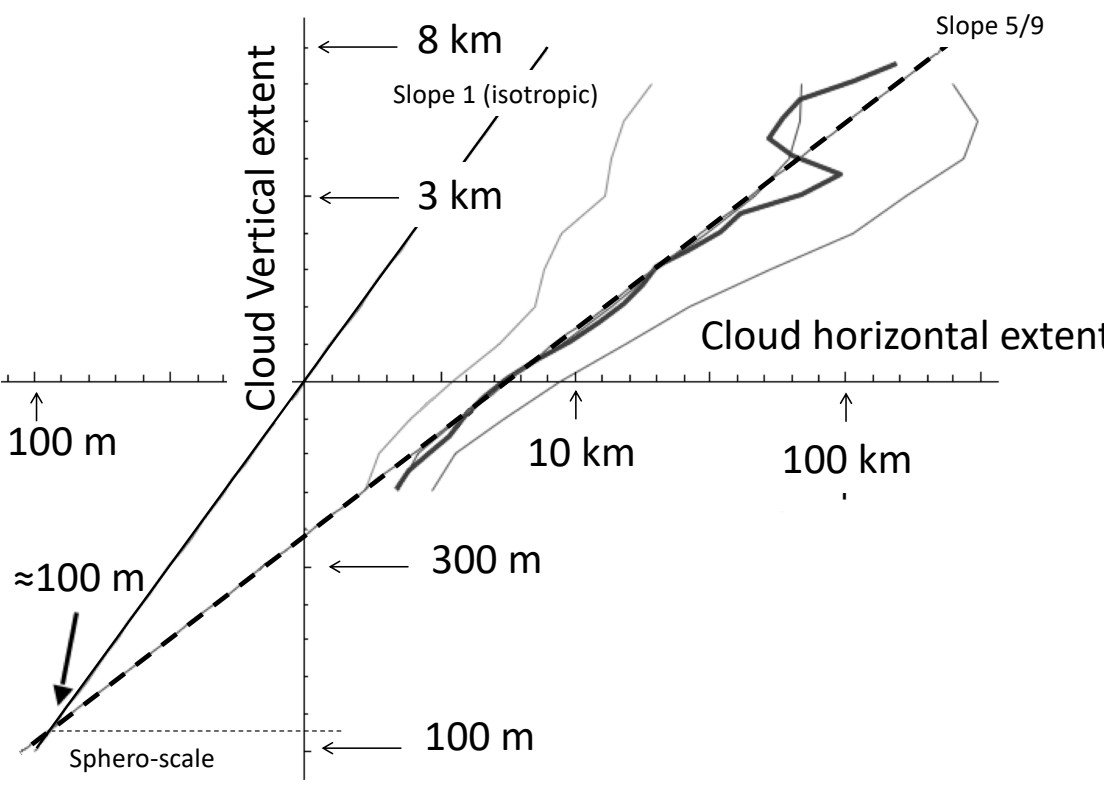

**Fig. 50: A space (horizontal) – space (vertical) diagram estimated from the absolute reflectivity fluctuations (first order structure functions) from 16 CloudSat orbits. Reproduced from [*Lovejoy et al.*, **2009b**].**




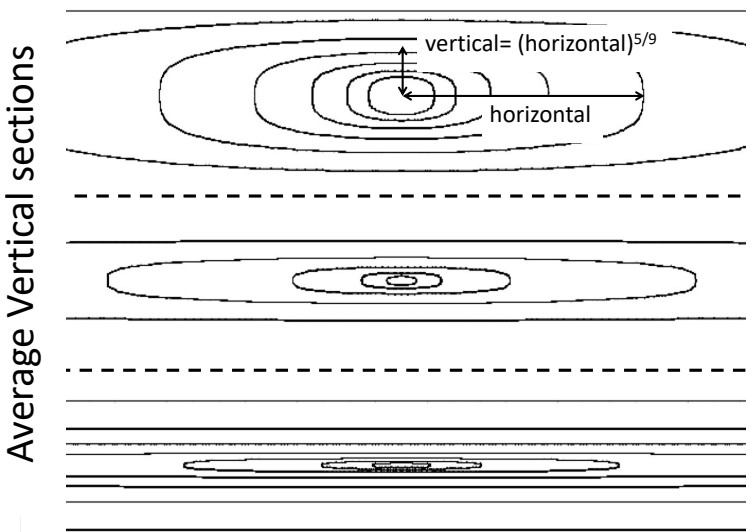

**Fig. 51: The theoretical shapes of average vertical cross-sections using the empirical parameters estimated from CloudSat - derived mean parameters: $H_z = 5/9$, with sphero-scales 1km (top), 100m (middle), 10m (bottom), roughly corresponding to the geometric mean and one-standard-deviation fluctuations. In each of the three, the distance from left to right horizontally is 100km, from top to bottom vertically is 20km. It uses the canonical scale function. The top figure in particular shows that structures 100km wide will be about 10km thick whenever the sphero-scale is somewhat larger than average [*Lovejoy et al.*, 2009a].**

|  | $T$ | $Log\,\theta$ | $h$ | $v$ | $B$ | $L$ |
|---|---|---|---|---|---|---|
| $H_z=H_h/H_v$ | 0.47±0.09 | 0.47±0.09 | 0.65±0.06 | 0.57±0.02 | 0.53±0.02 | 0.53±0.02 |

**Table 3: The above uses the estimate of the vertical $H_v$, the (horizontal) values for $H_h$ for $T$, log$\theta$, $h$ (humidity), these are from [*Lovejoy and Schertzer*, 2013]. For the horizontal velocity $v$, the aircraft data in fig. 48 were used. $B$ is the lidar reflectivity (spectral estimate in fig. 46). For clouds, the far right column ($L$) is an estimate using CloudSat cloud length and depth probability data from [*Guillaume et al.*, 2018] we find $H_z = 0.53±0.02$.**

### 4.2.3 The 23/9D model and Numerical Weather Models

What about numerical weather models? We mentioned that in the horizontal they show excellent scaling (and see fig. 8 for reanalysis spectra, and fig. 9, for the comparison of Mars and Earth spectra, fig. 38 for the cascade structures). Unfortunately, it is nontrivial to test the vertical scaling in models (they are typically based on pressure levels, and their statistics are different than those from true vertical displacements). Nevertheless, an indirect test is consider the number of vertical levels compared to horizontal levels. In the historical development of NWP's, the number of spatial degrees of freedom - product of horizontal degrees of freedom multiplied by the vertical number was limited by computer power. In any given model, the number of vertical levels compared to the number of horizontal pixels is a somewhat an ad hoc model choice. Yet if we consider




([*Lovejoy*, 2019]) the historical development of the models since 1956 (fig. 52), we see that the number of levels (the vertical range) as a function of the number of zonal degrees of freedom (horizontal range) has indeed followed the 5/9 power law so that:

(vertical range) ≈ (horizontal range)$^{Hz}$

(59)

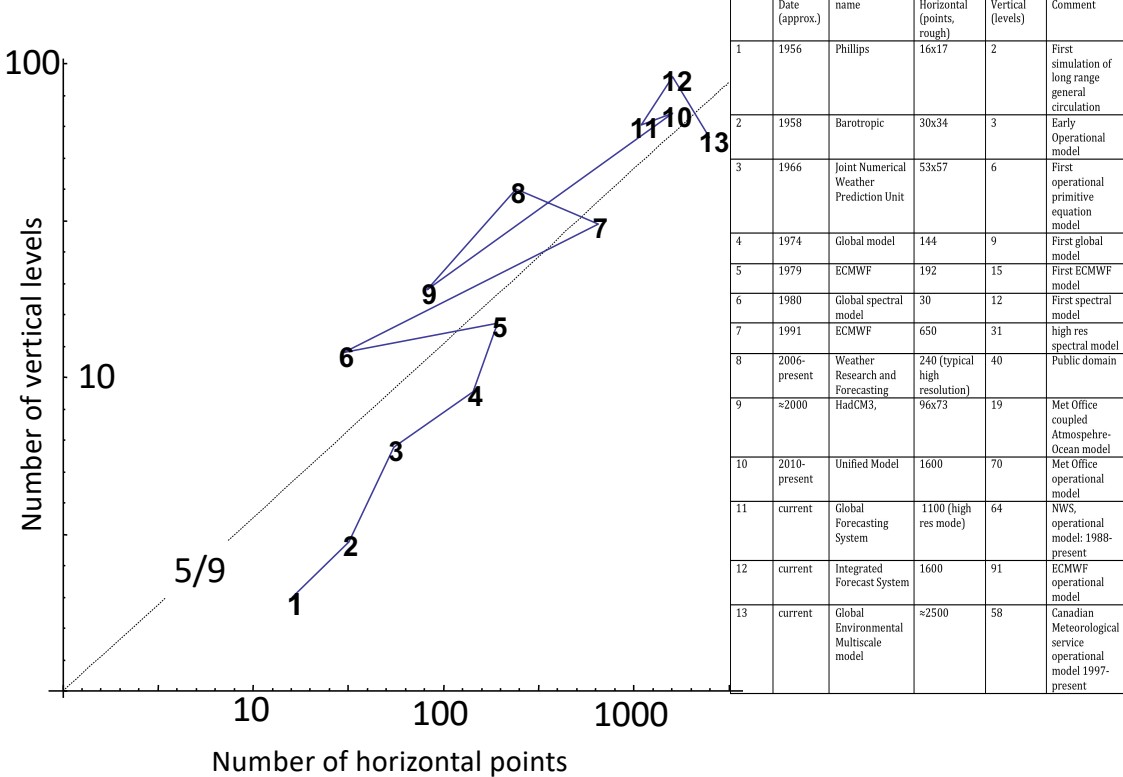

| | Date (approx.) | name | Horizontal (points, rough) | Vertical (levels) | Comment |
|---|---|---|---|---|---|
| 1 | 1956 | Phillips | 16x17 | 2 | First simulation of long range general circulation |
| 2 | 1958 | Barotropic | 30x34 | 3 | Early Operational model |
| 3 | 1966 | Joint Numerical Weather Prediction Unit | 53x57 | 6 | First operational primitive equation model |
| 4 | 1974 | Global model | 144 | 9 | First global model |
| 5 | 1979 | ECMWF | 192 | 15 | First ECMWF model |
| 6 | 1980 | Global spectral model | 30 | 12 | First spectral model |
| 7 | 1991 | ECMWF | 650 | 31 | high res spectral model |
| 8 | 2006-present | Weather Research and Forecasting | 240 (typical high resolution) | 40 | Public domain |
| 9 | ≈2000 | HadCM3, | 96x73 | 19 | Met Office coupled Atmosphere-Ocean model |
| 10 | 2010-present | Unified Model | 1600 | 70 | Met Office operational model |
| 11 | current | Global Forecasting System | 1100 (high res mode) | 64 | NWS, operational model: 1988-present |
| 12 | current | Integrated Forecast System | 1600 | 91 | ECMWF operational model |
| 13 | current | Global Environmental Multiscale model | ≈2500 | 58 | Canadian Meteorological service operational model 1997-present |

**Fig. 52: In the atmosphere it appears that the dynamics are dominated by Kolmogorov scaling in the horizontal ($H_h = 1/3$) and Bolgiano-Obhukhov scaling in the vertical (Hv = 3/5) so that $H_z = H_h/H_v = 5/9 = 0.555...$ Assuming that the horizontal directions have the same scaling, then typical structures of size LxL in the horizontal have vertical extents of $L^{Hz}$ hence their volumes are LDel with "elliptical dimension" $D_{el} = 2+H_z = 2.555...$; the "23/9D model" [*Schertzer and Lovejoy*, 1985a]. This model is very close to the empirical data and it contradicts the "standard model" that is based on isotropic symmetries and that attempts to combine a small scale isotropic 3D regime and a large scale isotropic (flat) 2D regime with a transition supposedly near the atmospheric scale height of 10km. The requisite transition has never been observed and claims of large scale 2D "geostrophic" turbulence have been shown to be spurious (reviewed in ch. 2 of [*Lovejoy and Schertzer*, 2013]). Reproduced from [*Lovejoy*, 2019].**





### 4.3 GSI in the horizontal: cloud morphologies, differential rotation, nonlinear GSI

#### 4.3.1 Differential rotation

Due to the larger north-south temperature gradients large atmospheric structures 10000km in the east-west direction are typically "squashed" to a size about $a \approx 1.6$ times smaller in the north-south direction (section 4.1.5). However, there is no
*systematic* change in this aspect ratio as we move to smaller scales, nor is there a plausible theory that might explain one. Although this statement is true of the data, it turns out that one of the limitations of GCM's is that they *do* have *horizontal* stratifications that are apparently spurious. If the east-west direction is taken as the reference, then GCM structures in the north - south direction follow: (North-South) = (East-West)$^{Hy}$ with $H_y$ = 0.80 for this, and a possible explanation, see: [*Lovejoy and Schertzer*, 2011].

With this possible exception, we conclude that unlike the vertical, there is little evidence for any *overall* stratification in the horizontal analogous to the vertical, but there is still plenty of evidence for the existence of different shapes at different sizes, and the fact that shapes commonly rotate by various amounts at different scales. We thus need to go beyond self-affinity and (at least) add some rotation. Mathematically, to add rotation to the blowup and squashing that we discussed earlier we only need to add off-diagonal elements to the generator $G$.

Fig. 53-56 shows a few examples of contours at different scales, each representing the shapes of the balls at systematically varying scales. We can see that we have freedom to vary the unit balls (here circles and rounded triangles) as well as the amounts of squashing and rotation. In fig. 53, with unit balls taken to be circles, we show the self-similar case the upper left, a stratified case upper right, a stratified case with a small amount of rotation (lower left), and another case with lots of rotation (lower right). Fig. 56 shows the same but with unit balls as rounded
triangles, and fig. 55 takes the lower right example and displays them over a factor of a billion in scale, and in fig. 56 we show an example with only a little rotation but over the same factor of a billion in scale. We can see that if these represent average morphologies of clouds at different scales, that even though there is a single unique rule or mechanism to go from one scale to another, that the average shapes change quite a bit with scale.


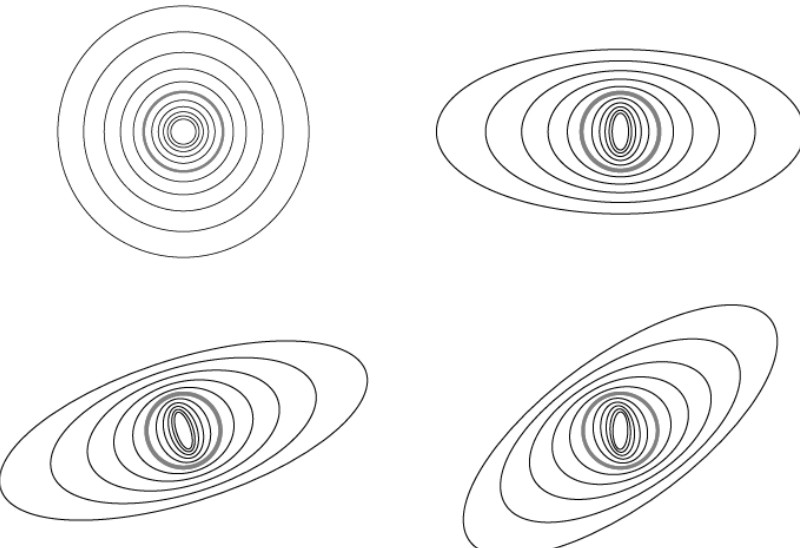

**Fig. 53: Blow-ups and reductions by factors of 1.26 starting at circles (red). The upper left shows the isotropic case, the upper right shows the self-affine (pure stratification case), the lower left example is stratified but along oblique directions, and the lower right example has structures that rotate continuously with scale while becoming increasingly stratified. The matrices used are:** $G = \begin{pmatrix} 1 & 0 \\ 0 & 1 \end{pmatrix}$ **,** $\begin{pmatrix} 1.35 & 0 \\ 0 & 0.65 \end{pmatrix}$ **,** $\begin{pmatrix} 1.35 & 0.25 \\ 0.25 & 0.65 \end{pmatrix}$ **,** $\begin{pmatrix} 1.35 & -0.45 \\ 0.85 & 0.65 \end{pmatrix}$ **(upper left to lower right). Reproduced from [Lovejoy, 2019].**





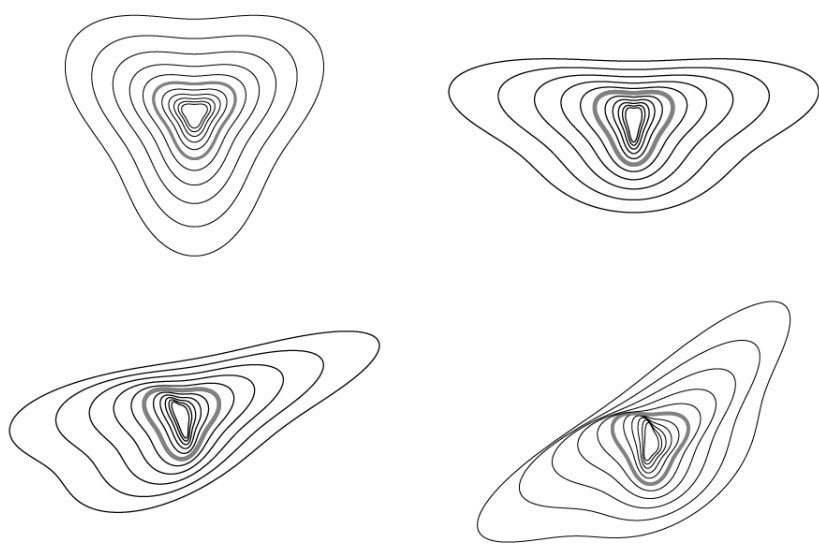

**Fig. 54:  The same as above except that now the unit ball is the rounded triangle.  Reproduced from [Lovejoy, 2019].**




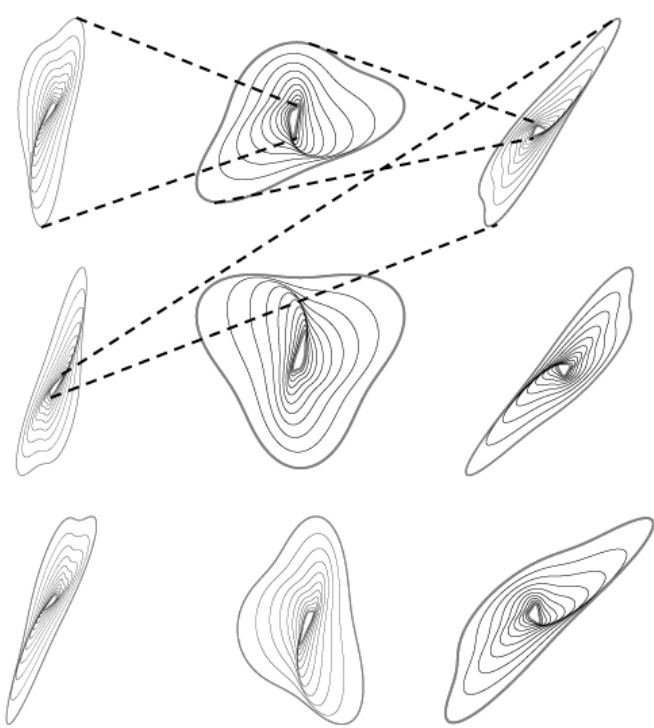

**Fig. 55: The same blow-up rule as in the lower right of fig. 53, but showing an overall blow-up by a factor of a billion. Starting with the inner thick grey ball in the upper left corner, we see a series of 10 blow-ups, each by a factor of 1.26 spanning a total of a factor of ten (the outer red ball). Then, that ball is shrunk (as indicated by the dashed lines) so as to conveniently show the next factor of ten blow-up (top middle). The overall range of scales in the sequence is thus 10⁹ = a billion. The scale-changing rule (matrix) used here is the same as the lower right in figs. 53, 54** $G = \begin{pmatrix} 1.35 & -0.45 \\ 0.85 & 0.65 \end{pmatrix}$. **Reproduced from [Lovejoy, 2019].**





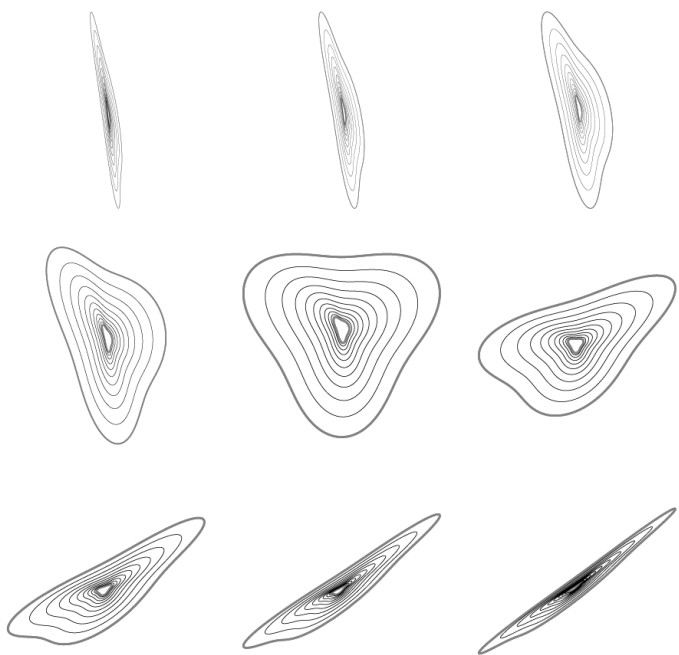

**Fig. 56: A different example of balls with squashing but with only a little rotation: the maximum rotation of structures in this example from very small to very large scales is 55°. The matrix used here was G = $\begin{pmatrix} 1.1 & 0.02 \\ 0.18 & 0.9 \end{pmatrix}$. Reproduced from [Lovejoy, 2019].**

### 4.3.2 Anisotropic multifractal clouds

We have explored ways in which quite disparate shapes can be generated using blowups, squashings and rotations. With the help of a unit ball, we generated families of balls any member of which would have been an equally good starting point. The unit ball has no particular importance, it does not have any special physical role to play. If we have a scaling model based on isotropic balls, then replacing them with these anisotropic balls will also be scaling when we use the anisotropic rule to change scales: any morphologies made using such a system of balls will be scale invariant. Mathematically anisotropic space-time

models (see [*Schertzer and Lovejoy*, 1987], [*Wilson et al.*, 1991], [*Lovejoy and Schertzer*, 2010a], [*Lovejoy and Schertzer*, 2010c]) are produced the same way as isotropic ones except that the usual vector norm is replaced by a space-time scale function and the usual dimension of space-time $D$ (=4) is replaced by $D_{el}$:

$$\left| \underline{R} \right| \rightarrow \left[\!\left[ \underline{R} \right]\!\right]; \quad D \rightarrow D_{el}$$

(60)





where $\underline{R} = (x, y, z, t)$.

We already showed a self-similar and stratified example where the balls were used to make a multifractal cloud simulation of a vertical section (fig. 5). Let's now take a quick look at a few examples of horizontal and three-dimensional multifractal cloud simulations.

The simulation of a cross-section of a stratified multifractal cloud in fig. 5 already shows that the effect of changing the balls can be quite subtle. Let's take a look at this by making multifractal cloud simulations with realistic (observed) multifractal
parameters (these determine the fluctuation statistics, not the anisotropy), and systematically varying the families of balls (fig. 57). In the figure, all the simulations have the same random "seed" so that the only differences are due to the changing definition of scale. First we can explore the effects of different degrees of stratification combined with different degrees of rotation. We consider two cases, in the first (fig. 57), there is roughly a circular unit ball within the simulated range, the second, fig. 58, all the balls are highly anisotropic. Each figure shows a pair: the cloud simulation (left) and the
family of balls that were used to produce it on the right.

From the third column in fig. 57 with no stratification, we can note that changing the amount of rotation (moving up and down the column) changes nothing; this is simply because the circles are rotated to circles, rotation is only interesting when combined with stratification. The simulations in fig. 58 might mimic small clouds (for example 1 km across) produced by complex cascade type dynamics that started rotating and stratifying at scales perhaps ten
thousand times larger. In both sets of simulations, the effect of stratification becomes more important up and down away from the centre line and the effects of rotation vary from the left to the right becoming more important as we move away from the third column.

Fig. 59 shows examples where rotation is strong and the scale changing rule is the same everywhere; only the unit ball is changed. By making the latter have some long narrow parts, we can obtain quite "wispy" looking clouds.

Fig. 60 shows another aspect of multifractal clouds. In section 3.5.5 we discussed the fact that in general, the cascades occasionally produce extreme events. If we make a sufficiently large number of realizations of the process, from time to time we will generate rare cloud structures that are almost surely absent on typical realizations. For example, a typical satellite picture of the tropical Atlantic Ocean would not have a hurricane, but from time to time hurricanes do appear there. The multifractality implies that this could happen quite naturally, without the need to invoke any special scalebound "hurricane
process". In the examples in fig. 60, we use a rotating set of balls (fig. 61). However, in order to simulate occasional, rare realizations, we have "helped" the process by artificially boosting the values in the vicinity of the central pixel. The two different rows are identical except for the sequence of random numbers used in their generation. For each row, moving from left to right, we boosted only the central region to simulate stronger and stronger vortices that are more and more improbable. As we do this, we see that the shapes of the basic set of balls begins to appear out of the chaos.





### 4.3.3 Radiative transfer in multifractal clouds

The cloud simulations above are for the density of cloud liquid water; they used false colours to display the more and less dense cloud regions. Real clouds are of course in three-dimensional space, and the eye sees the light that has been scattered by the drops. Therefore, if we make three-dimensional cloud simulations, instead of simply using false colours, we can obtain more realistic renditions by simulating the way light interacts with the clouds. The study of radiative transfer in multifractal clouds is in its infancy; see however: [Naud et al., 1997], [Schertzer et al., 1998], [*Lovejoy et al.*, 2009d], [Watson et al., 2009].

Fig. 61, 62 shows top and side views of a multifractal cloud with the usual false colours; fig. 64, 65 shows the same cloud rendered by simulating light travelling through the cloud, both top (64) and bottom (65) views. Finally in fig. 66, we show a simulation of thermal infrared radiation emitted by the cloud similar to what can observed from infrared weather satellites. We see that quite realistic morphologies are possible.

Up until now, we have only discussed space, but of course clouds and other atmospheric structures evolve in time. Since we have argued that the wind field is scaling – and the wind moves clouds around, it effectively couples space and time. We therefore have to consider scaling in space and in time: in space-time. The time domain opens up a whole new realm of possibilities for simulations and morphologies. Whereas the balls in space must be localized – since they represent typical spatial structures, "eddies" - in space-time they can be delocalized and form waves. In this case it turns out that it is easier to describe the system using the Fourier methods. Fig. 67 shows examples of what can be achieved with various parameters.





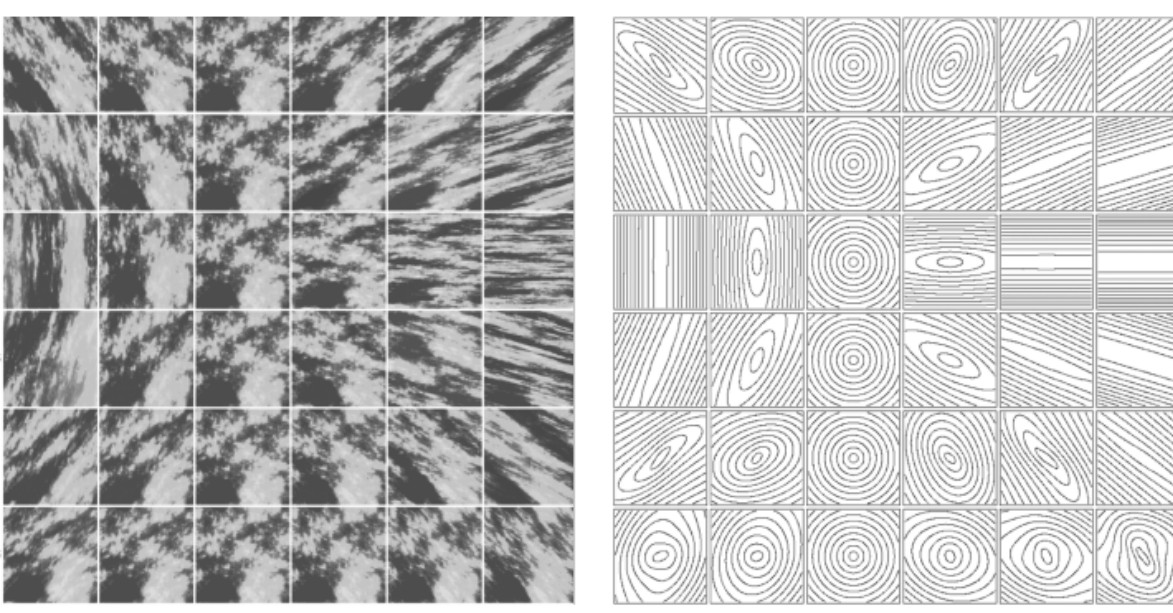

Fig. 57: **Left: Multifractal simulations with nearly isotropic unit scales with stratification becoming more important up and down away from the centre line and the rotation parameter (left to right) becoming more important as we move away from the third column.**

1835

**Right: The balls used in the simulations to the left.**

**This is an extract from the multifractal explorer website: http://www.physics.mcgill.ca/~gang/multifrac/index.htm. Reproduced from [Lovejoy and Schertzer, 2007b].**

1840





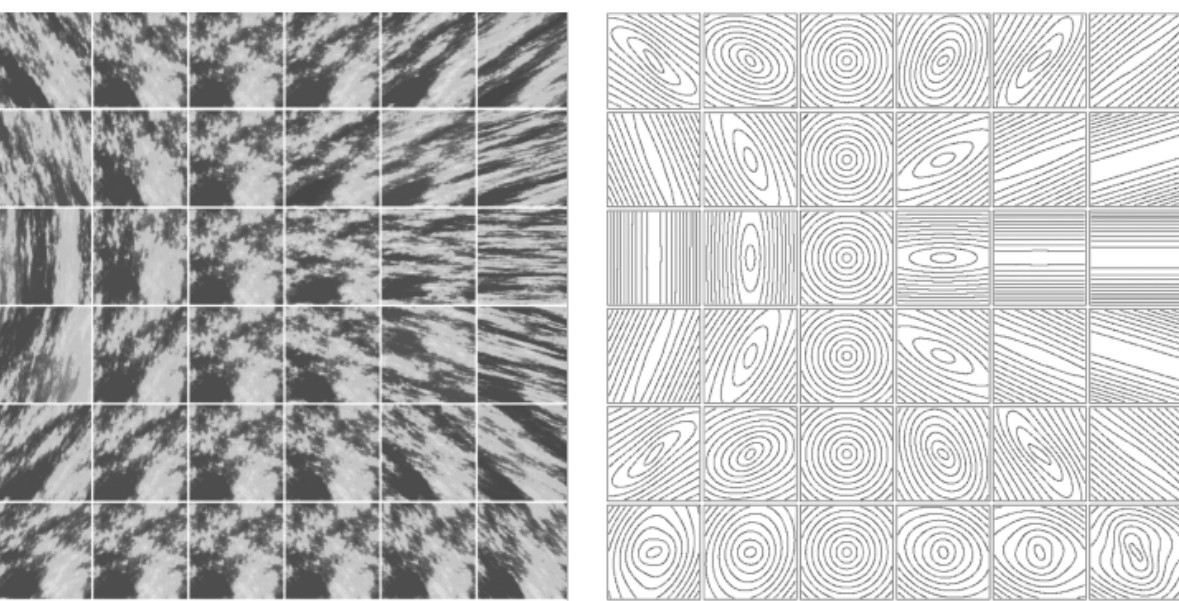

**Fig. 58:** The same as the above except that the initial ball is highly anisotropic in an attempt to simulate the effect of stretching due to a wide range of larger scales. Reproduced from [*Lovejoy and Schertzer*, 2007b].


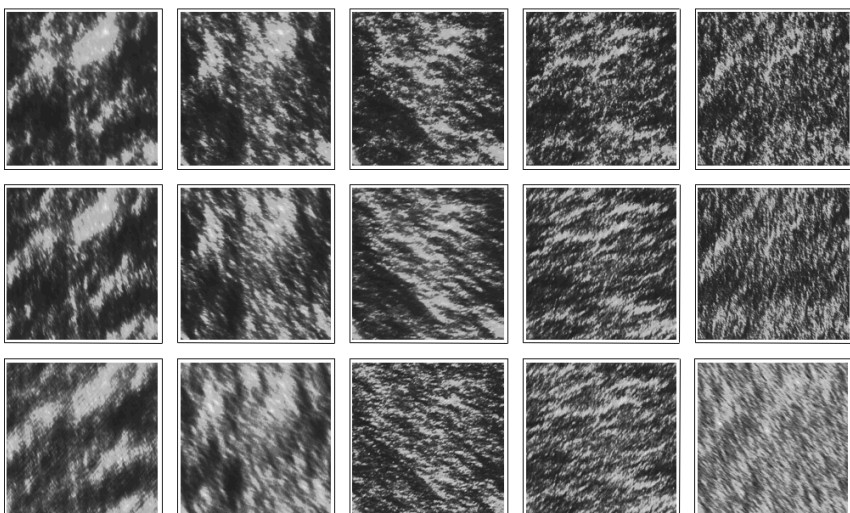

845 **Fig. 59: Simulations of cloud liquid water density with the scale changing rule the same throughout, only the unit balls are systematically modified so as to yield more and more "wispy" clouds. Reproduced from [Lovejoy et al 2009].**





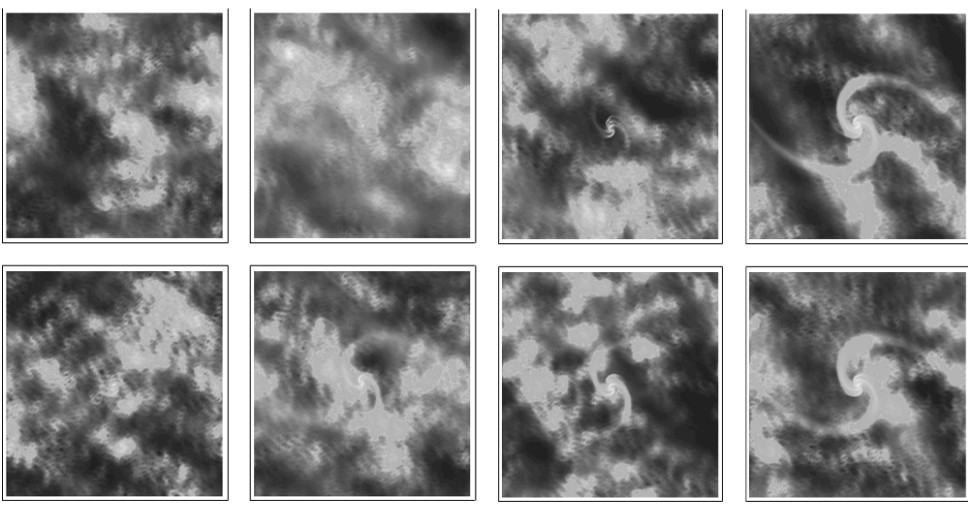

**Fig. 60: Each row has a different random seed, but is otherwise identical. Moving from left to right, shows a different realization of a random multifractal process with the central part boosted by factors increasing from left to right in order to simulate very rare events. The balls are shown in fig. 61. Reproduced from [Lovejoy and Schertzer 2013].**

1850

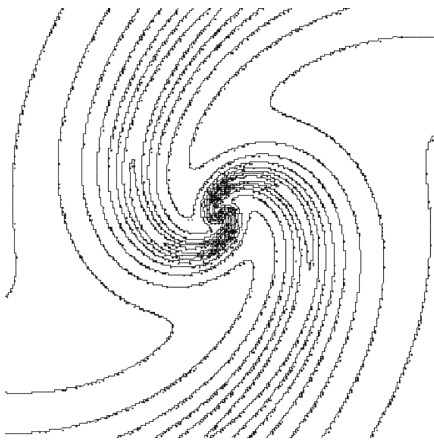

**Fig. 61: The balls used in the simulations above. Contours of the (rotation dominant) scale function used in the simulations 62. Reproduced from Lovejoy and Schertzer 2013.**



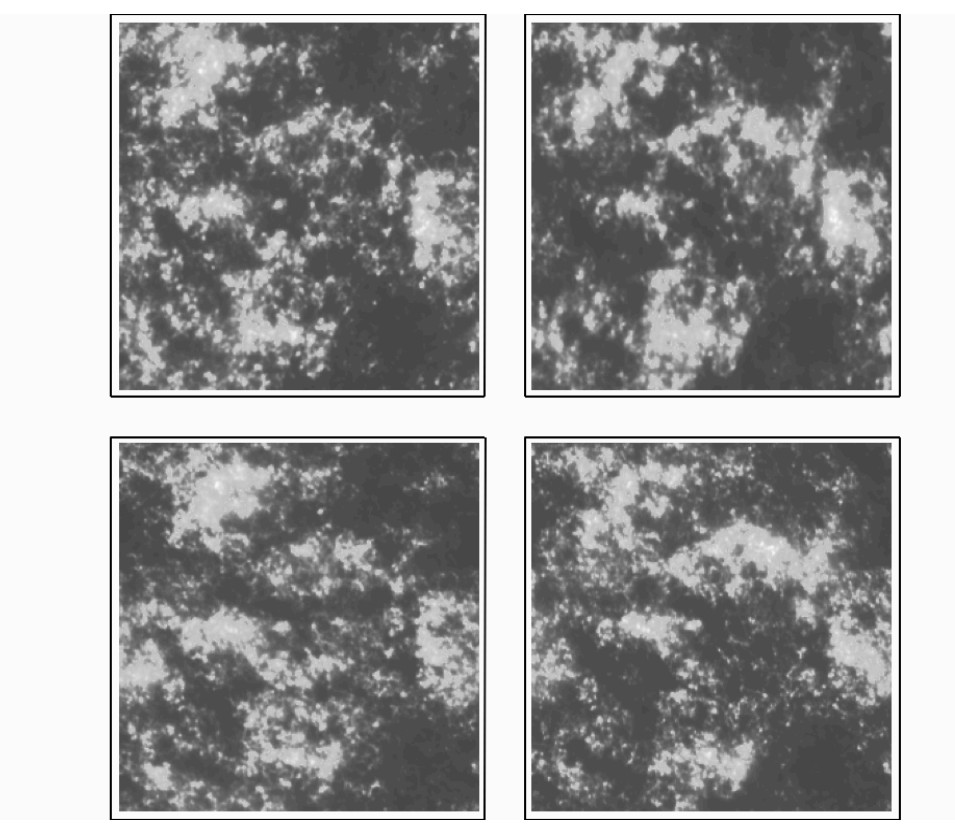

**Fig. 62: The top layer of cloud liquid water using a false colour rendition. Reproduced from [Lovejoy and Schertzer 2013].**



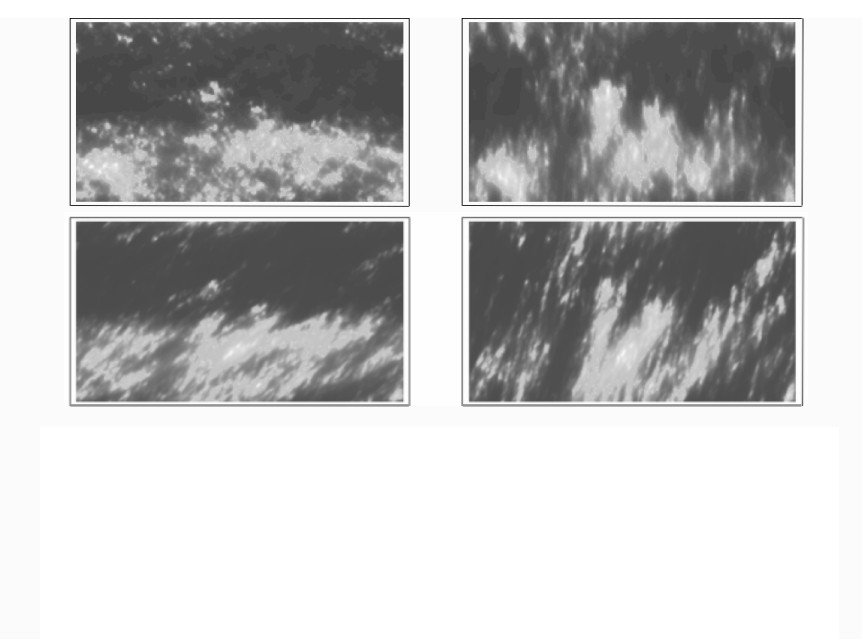

**Fig. 63: A side view of the previous. Reproduced from [Lovejoy and Schertzer 2013].**

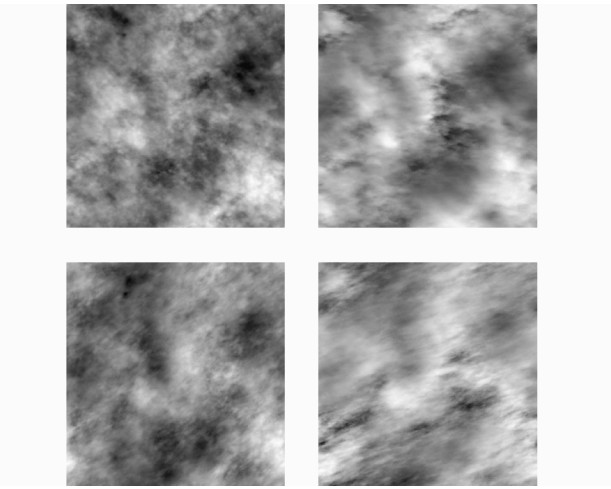

**Fig. 64: The top view with light scattering for the sun (incident at 45º to the right). Reproduced from [Lovejoy and Schertzer 2013].**


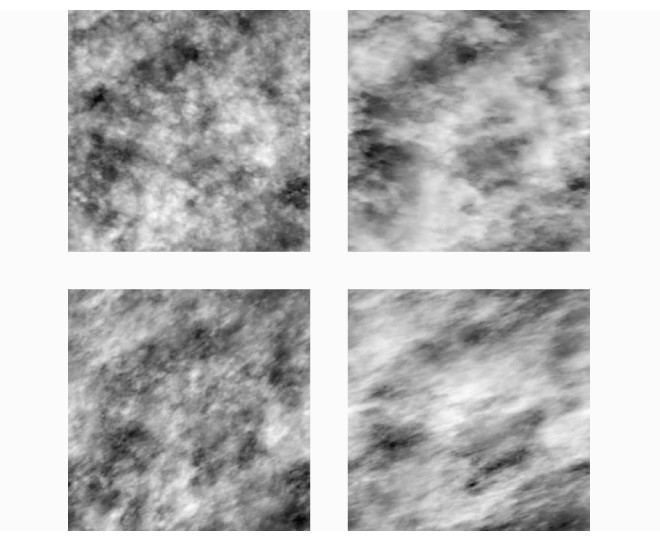

**Fig. 65: The same as fig. 64 except viewed from the bottom. Reproduced from [Lovejoy and Schertzer 2013].**

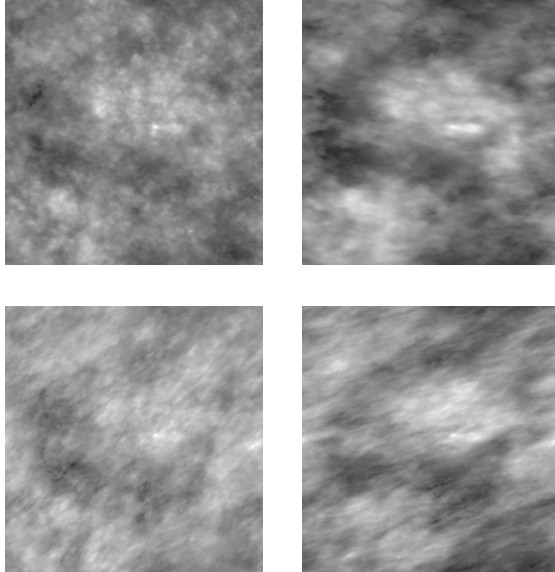

**Fig. 66: The same as the previous except for a false colour rendition of a thermal infra red field as might be viewed by an infra red satellite. Reproduced from [Lovejoy and Schertzer 2013].**


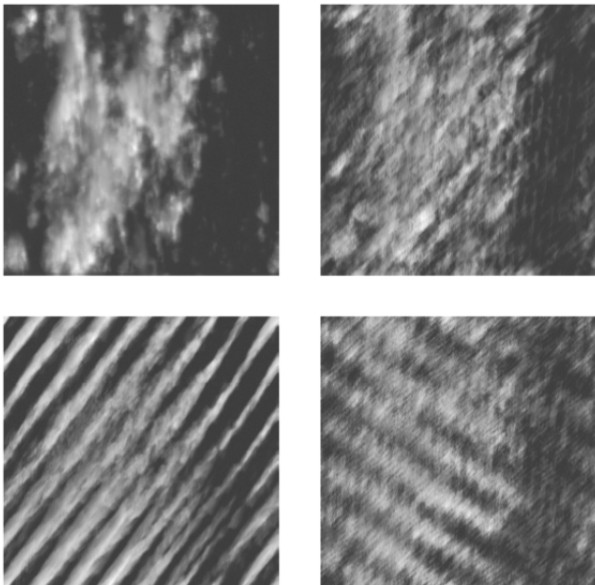

**Fig. 67: Examples of simulations in space-time, showing wave-like morphologies. The same basic shapes are shown but with wavelike character increasing clockwise from the upper left. Reproduced from reference [*Lovejoy et al.*, 2008b].**

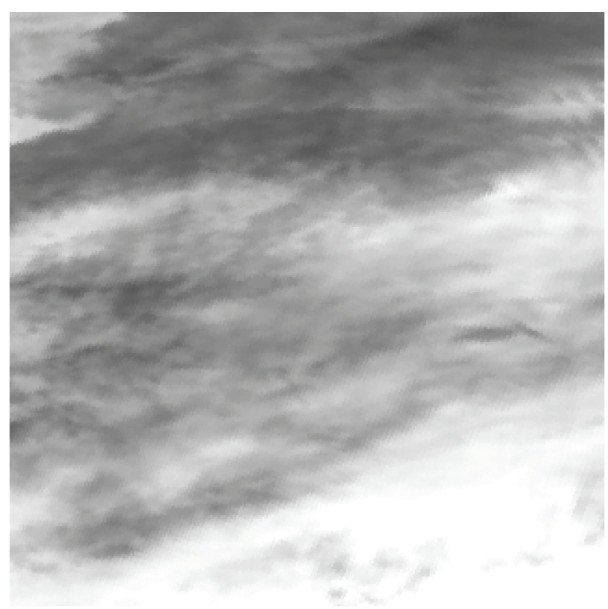

**Fig. 68: An infra red satellite image from a satellite at 1.1 *km* resolution, 512x512 pixels. Reproduced from [Lovejoy and Schertzer 2013].**




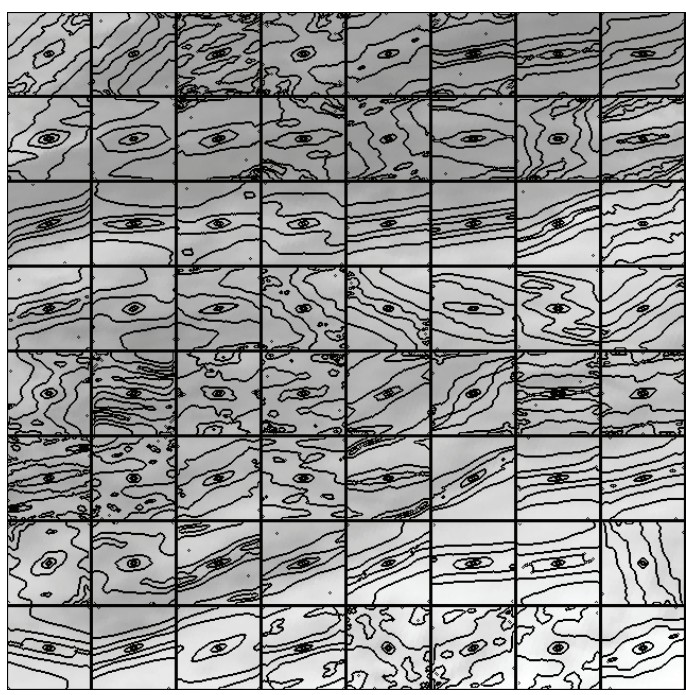

**Fig. 69: Estimates of the shapes of the balls in each 64x64 pixel box from the image in fig. 68. Reproduced from [Lovejoy and Schertzer 2013].**

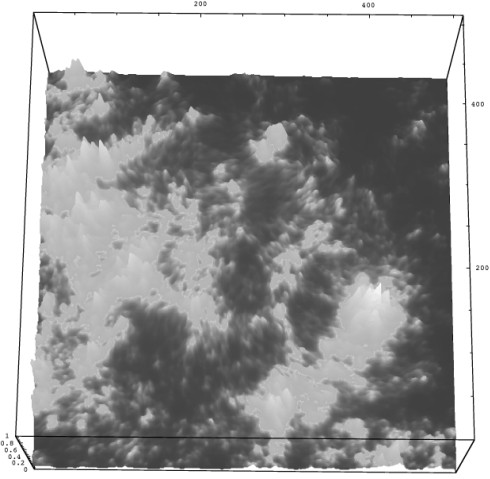

**Fig. 70: A multifractal simulation of a cloud with texture, morphology varying in both location and scale, simulated using nonlinear GSI; the anisotropy depends on both scale and position according to the balls shown in fig. 71. Reproduced from Lovejoy and Schertzer 2013.**





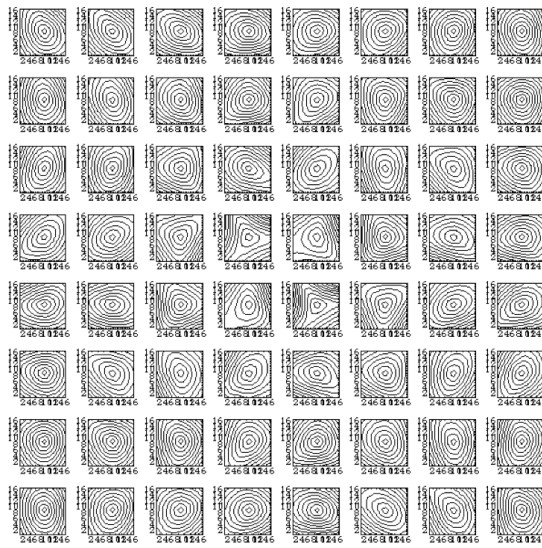

**Fig. 71: The set of balls displayed according to their relative positions used in the simulation shown in fig. 70. Reproduced from [Lovejoy and Schertzer 2013].**

### 4.3.3 Nonlinear GSI: Anisotropy that changes from place to place as well as scale to scale

Generalized Scale Invariance, is necessary since zooming into clouds would displays systematic changes of morphology with the magnification, so that in order to be realistic, we needed to generalize the idea of self-similar scaling. The first step was to account of the stratification; with a well-defined direction meant that in pure stratification there is no rotation with scale. To model the horizontal plane, we needed to add rotation and to a first approximation, we could think of the different cloud

morphologies as corresponding to different cloud types – cumulus, stratus, cirrus etc.

But there is still a problem. Up until now, we have discussed linear GSI where the generator is a matrix so that the scale changing operator $\lambda^{-G}$ is also a linear transformation. Now we need to generalize this to account for the fact that because cloud types and morphologies not only change with scale, they also change with spatial location (and in time). Fig. 66, shows the problem with a real satellite infrared cloud picture; it seems clear that the textures and morphologies vary from one part of the

image to another. Using a type of two-dimensional fluctuation analysis, we can try to estimate the corresponding "balls". When the image is broken up into an 8 X 8 array of sub images, (fig. 69, with a fair bit of statistical scatter) we can confirm that the balls are quite different from one place to another.

In the more general nonlinear GSI, the notion of scale depends not only on the scale, but also on the location. In nonlinear GSI we introduce the generator of the infinitesimal scale change $\underline{g}\left(\underline{r}\right)$ (consider just space and use the notation, $\underline{r} = (x_1, x_2,$

$x_3)$). Using $g(\underline{r})$ we can obtain the following equation for the scale function:


$$g_i \frac{\partial}{\partial x_i}\|\underline{r}\| = \|\underline{r}\| \tag{61}$$

(summation over repeated indices).

Locally (in a small enough neighbourhood of a point), linear GSI is defined by the tangent space, i.e. the elements of the linear generator are:

$$G_{ij} = \frac{\partial g_i}{\partial x_j} \tag{62}$$

In the special case of linear GSI this yields:

$$x_i G_{ij} \frac{\partial}{\partial x_j}\|\underline{r}\| = \|\underline{r}\| \tag{63}$$

Full details and more examples are given in ch. 7 of [*Lovejoy and Schertzer*, 2013].

Figs. 70, 71 show an example. The physics behind this is analogous to those in Einstein's theory of general relativity. In the latter, it is the distribution of mass and energy in the universe that determine the appropriate notion of distance, i.e. the metric. With GSI, it is the nonlinear turbulent dynamics that determine the appropriate notions of scale and size. The GSI notion of scale is generally *not* a metric, it is *not* a distance in the mathematical sense.

With nonlinear GSI a bewildering variety of phenomena can be described in a scaling framework. The framework turns out to be so general that it is hard to make further progress. It's like saying "the energy of the atmosphere is conserved". While this is undoubtedly true – and this enables us to reject models that fail to conserve it - this single energy symmetry is hardly adequate for modelling and forecasting the weather. One can imagine that if one must specify the anisotropy both as a function of scale and as a function of location, that many parameters are required. At a purely empirical level, these are difficult to estimate since the process has such strong variability and intermittency. In order to progress much further, we'll undoubtedly need new ideas. However, the generality of GSI does make the introduction of scalebound mechanisms unnecessary.

### 4.3.4 The scale bound approach and the phenomenological fallacy

We have given the reader a taste of the enormous diversity of cloud morphologies that are possible within the scaling framework. We discussed morphologies that were increasingly stratified at larger scales, that rotated with scale but only a bit or that rotated many times. There were filamentary structures, there were structures with waves and there were structures whose character changed with position. Although all of these morphologies changed with scale, they were all consequences of mechanisms that were scale invariant. The scalebound approach is therefore logically wrong and scientifically unjustified. When scalebound mechanisms and models based solely on phenomenological appearances are invoked; they commit a





corollary of the scalebound approach: the "phenomenological fallacy" [*Lovejoy and Schertzer*, 2007a]. More concisely: the phenomenological fallacy is the inference of mechanism from phenomenology (appearances).

## 5. Conclusions

Starting in the 1970's, deterministic chaos, scaling and fractals transformed our understanding of many nonlinear dynamical systems including atmospheric science: they were the main components of the "nonlinear revolution". While deterministic chaos is largely a deterministic small number of degrees of freedom paradigm, scaling, fractals – and later multifractals – is a stochastic, high number of degrees of freedom framework that is particularly appropriate to the atmosphere. Indeed, scaling

has been explicit and central in turbulence theories ever since the Richardson 4/3 law of turbulent diffusion in the 1920's.

For the last century, two strands of atmospheric science have developed largely in parallel. The first, "dynamical meteorology", is an essentially mechanistic, phenomenologically based approach: it was largely scalebound because the relevant processes were believed to occur of narrow ranges of scale so that to explain the wide range of atmospheric variability, a large number of such processes were required. The second, a statistical turbulence approach was on the contrary based on

the scaling idea -  that there is a simple statistical relation between structures, processes at potentially widely different scales. Yet, the classical turbulence notion of scaling was highly restrictive. For one, it assumed that processes were not far from Gaussian whereas real world turbulence is on the contrary highly intermittent. For another, it reduced scaling to its isotropic special case: "self-similarity" - effectively confounding the two quite different scale and direction symmetries.

This review focuses on the new developments in scaling that started in the early 1980's that overcame these

restrictions: multifractals to deal with scaling intermittency (section 3) and Generalized Scale Invariance (section 4) to deal with scaling stratification and more generally scaling anisotropy. GSI, clarifies the significance of scaling in geoscience since it shows that scaling is a rather general symmetry principle: it is thus the simplest relation between scales. Just as the classical symmetries (temporal, spatial invariance, directional invariance) are equivalent (Noether's theorem) to conservation laws (energy, momentum, angular momentum), the (nonclassical) scaling symmetry conserves the scaling exponents $G, K, c$.

Symmetries are fundamental since they embody the simplest possible assumption, model: under a system change, there is an invariant. In physics, initial assumptions about a system are that it respects symmetries. Symmetry breaking is only introduced on the basis of strong evidence or theoretical justification: in the case of scale symmetries, to introduce characteristic space or time scales.

There are now massive data analyses of all kinds confirming and quantifying atmospheric scaling over wide ranges in the

horizontal and vertical. Since this includes the wind field, it implies that the dynamics (i.e. in time) are also scaling. Sections 1, 2 discuss how over the range of milliseconds to at least hundreds of millions of years – temporal scaling objectively defines five dynamical ranges: weather, macroweather, climate, macroclimate and megaclimate. By considering the development of the scalebound framework from the 1970's (Mitchell) to the 2020's (Von der Leyden et al), that is shown to be further and





further divorced from empirical science. This is also true of the usual interpretation of space-time (Stommel) diagrams that are re-interpreted in a scaling framework (section 2.6). These scalebound frameworks have survived because practising atmospheric scientists increasingly rely instead on General Circulation Models that are based on the primitive dynamical equations. Fortunately, the outputs of these models inherit the scaling of the underlying equations, and are hence themselves scaling, they can therefore be quite realistic.

Similar comments apply to the still dominant isotropic theories of turbulence that - although based on scaling - illogically place priority on the directional symmetry (isotropy), ahead of the scaling one – and this in spite of the obvious and strong atmospheric stratification. In order for these theories to be compatible with the stratification – notably the 10km scale height - they attempt to marry models of small scale three dimensional isotropic turbulence with ("layerwise") two dimensional (quasi-geostrophic") turbulence at large scales. It turns out that the only empirical evidence supporting such an implicit "dimensional transition" is spurious: it comes from aircraft data following isobars rather than isoheights. Although this has been known for over a decade, thanks to the wide range scaling of the GCMs, it does not impact practising atmospheric scientists. The main casualty is that the continued invocation of isotropic turbulence theories blinds the community from promising new stochastic, scaling approaches that are especially needed for macroweather and climate modelling especially long range forecast and climate projections.

The review also emphasizes the impact of the analysis of massive and new sources of atmospheric data. This involves the development of new data analysis techniques, for example trace moments (section 3) that not only directly confirm the cascade nature of the fields, but also give direct estimates of the outer scales (that turn out to be close to planetary scales (horizontal) and the scale height (vertical). However, for scales beyond weather scales (macroweather), fluctuations tend to decrease rather than increase with scale and this requires new data analysis techniques. Haar fluctuations are arguably optimal - being both simple to implement and simple to interpret (section 2, appendix B).

**Appendix A**

This appendix summarizes the technical characteristics of the data presently in fig. 2 and the corresponding multifractal parameters that characterizes their scaling. The only update in fig. 2 is the (top) megaclimate series that was taken from [*Grossman and Joachimski*, 2022] rather than the [*Veizer et al.*, 1999] stack whose characteristics are given in table A1, A2.

| | No. | Regime | Description | Resolution, time | Resolution, space |
|---|---|---|---|---|---|
| Time | 1 | weather | Thermistor, 2 hours | 1/15 s | 1 mm |





| | | | | | |
|---|---|---|---|---|---|
| | 2 | weather | Lander, 3 yrs | hourly | 1m |
| | 3 | weather | Montreal, 17yrs | hourly | 1m |
| | 4 | macroweather | 20 CR: 0 - 40 N, every 2° longitude | monthly | 2°x2° |
| | 5 | climate | GRIP paleo temp. | 85 years | 1m |
| | 6 | climate | EPICA paleo temp. | Depth | 1m |
| | 7 | macroclimate | Zachos stack | 5kyrs | global |
| | 8 | megaclimate | Veizer stack | 553kyrs | global |
| Space | 9 | weather | Aircraft | 0.5s | 280m |
| | 10 | weather | ECMWF reanalysis | daily | 1° |
| | 11 | macroweather | ECMWF reanalysis | monthly | 1°x1° |
| | 12 | climate | 20CR reanalysis | 140 years | 2°x2° |

**Table A1: A summary of the data used in fig. 2, 4. study (for more details, see [Lovejoy, 2018]). In fig. 2, the (top) series was the updated series from [Grossman and Joachimski, 2022]. It replaces the similar but somewhat lower resolution Veizer stack (no. 8 in the table).**

1990

| | Data | $H$ | $C_1$ | $\alpha$ | $q_D$ | $\gamma_{max}$ (theory) | $\gamma_{max}$ (observed) |
|---|---|---|---|---|---|---|---|
| Time | 1 | 0.54±0.003 | 0.013±0.001 | 1.60±0.08 | 3.1 | 0.34 | 0.39 |
| | 2 | 0.36±0.02 | 0.011±0.002 | 1.46±0.05 | 3.4 | 0.31 | 0.38 |
| | 3 | 0.38±0.01 | 0.021±0.001 | 1.50±0.07 | 6.2 | 0.22 | 0.27 |
| | 4 | -0.24±0.01 | 0.052±0.003 | 1.56±0.02 | 7.2 | 0.32 | 0.22 |





|  |  |  |  |  |  |  |  |
|---|---|---|---|---|---|---|---|
|  | 5 | 0.20±0.02 | 0.047±0.006 | 1.40±0.12 | 5.1 | 0.30 | 0.30 |
|  | 6 | 0.41±0.01 | 0.01±0.01 | 1.46±0.15 | 5.0 | 0.24 | 0.28 |
|  | 7 | -0.30±0.03 | 0.083±0.014 | 1.49±0.13 | 3.3 | 0.44 | 0.31 |
|  | 8 | 0.33±0.03 | 0.107±0.016 | 1.52±0.31 | 1.7 | 0.65 | 0.52 |
| Space | 9 | 0.485±0.004 | 0.055±0.002 | 1.52±0.16 | 3.5 | 0.38 | 0.38 |
|  | 10 | 0.55±0.02 | 0.070±0.005 | 1.41±0.06 | 13.0 | 0.35 | 0.38 |
|  | 11 | 0.56±0. 018 | 0.154±0.006 | 1.55±0.03 | 8.4 | 0.56 | 0.43 |
|  | 12 | 0.47±0.02 | 0.182±0.011 | 1.64±0.11 | 5.2 | 0.62 | 0.51 |

**Table A2:** The scaling parameters $H$, $C_1$, $\alpha$ and probability exponents $q_D$. The far right columns give theoretical estimates of the maximum spike heights using the parameters $C_1$, $\alpha$, $q_D$ and the scale ratio $\lambda$ of the plots in fig. 2 (= 1000 except for data sets 10, 11, 12 where $\lambda$ = 360, 360, 180 respectively; the spike plot for data set #6 is not shown). The theory column uses these parameters with the multifractal theory described in the text to estimate the solution to the equation $c\left(\gamma_{\max}\right) = 1$. The "observed" column determines $\gamma_{\max}$ from the spike plot directly: $\gamma_{\max} = \log\left(\left|\Delta T\right| / \overline{\left|\Delta T\right|}\right)_{\max} / \log\lambda$ where $\left(\left|\Delta T\right| / \overline{\left|\Delta T\right|}\right)_{\max}$ is the maximum spike. For comparison, for $\lambda$ =1000, Gaussian probabilities of $10^{-3}$, $10^{-6}$, $10^{-9}$ yield respectively $\gamma_{\max}$ = 0.20, 0.26, 0.30. Error estimates for the right hand columns (extremes) were not given due to their sensitivity to the somewhat subjective choice of range over which the regressions were made.

## Appendix B: Estimation methods for wide range scaling processes

### B.1: Introduction

In 1994, a new $H<0$ technique was proposed by [Peng et al., 1994] that was initially applied to biological series; the Detrended Fluctuation Analysis (DFA) method. The key innovation was simply to first sum the series (effectively an integration of order one), that has the effect of adding one to the value of $H$. The consequence is that in most geophysical series and transects (as long as $H>-1$), the resulting summed series had $H>0$ allowing the more usual difference and difference-like fluctuations to be applied. Over an interval $\Delta t$, the DFA method estimates fluctuations in the summed series by using the standard deviation of the residuals of a polynomial fit over the interval length $\Delta t$ (i.e. a different regression for each fluctuation at each time interval). We return to this in more detail in section B4, but for the moment, note that the interpretation of the DFA "fluctuation function" is sufficiently opaque that typical plots do not bother to even use units for the fluctuation amplitudes.



Over the following nearly decades, there evolved several more or less independent strands of scaling analysis, each with their own mathematical formalism and interpretations. The wavelet community dealing with fluctuations directly; the DFA community wielding a method that could be conveniently implemented numerically; and the turbulence community, focused on intermittency. In the meantime, most geo-scientists continued to use spectral analysis, occasionally with Singular Spectral Analysis (SSA), or Multi Taper Method (MTM) or other refinements.

New clarity was achieved by the first "Haar" wavelet [Haar, 1910]. There were two reasons for this: the simplicity of its definition and calculation and the simplicity of its interpretation [*Lovejoy and Schertzer*, 2012b]. To determine the Haar fluctuation over a time interval $\Delta t$, one simply takes the average of the first half of the interval and subtracts the average of the second half (Fig. 17 bottom, see section B3 for more details). As for the interpretation, when $H$ is positive, then it is (nearly) the same as a difference, whereas whenever $H$ is negative, the fluctuation can be interpreted as an "anomaly" (in this context an anomaly is simply the average over a segment length $\Delta t$ of the series with its long term average removed, see section Section B3). In both cases, in addition to a useful quantification of the fluctuation amplitudes, we also recover the correct value of the exponent $H$. Although the Haar fluctuation is only useful for $H$ in the range -1 to 1, this turns out to cover most of the series that are encountered in geoscience (see e.g. fig. 18).

## B.2 Fluctuations revisited

The inadequacy of using differences as fluctuations forces us to use a different definition. The root of the problem is that "cancelling" series ($H<0$) are dominated by high frequencies whereas "wandering" series ($H>0$) are dominated by low frequencies (see figs. B3, B4 discussed in section B2). As we discuss below, differencing can be thought of as a filtering operation that emphasizes the low frequencies so much that in the $H>1$ case, the result depends on the very lowest frequencies present in the series (later). Conversely, the difference filter doesn't much affect the high frequencies, so that in the $H<0$ case, difference fluctuations are determined by the very highest frequencies. In either case, the difference-filtered results are spurious in the sense that they depend on various details of the empirical samples: the overall series length and the small scale resolution respectively. Mathematically, the link between the mean amplitude of the fluctuation and the lags has been broken. How can we remedy the situation? First consider the case $-1<H<0$. If we could obtain a new series whose exponent was raised by 1 then it's exponent would be $1+H$ which would be in the range $0 \leq 1+H \leq 1$, hence *it's* fluctuations could be analysed by difference fluctuations. But this turns out to be is easy to achieve. Return to the simple sinusoid, frequency $\omega$, period $\Delta t = 1/\omega$: $T(t) = A \sin \omega t$ where the amplitude $A$ of this elementary fluctuation is identified with $\Delta T$. Now, consider its derivative: $A\omega \cos\omega t$. Since the difference between sine and cosine is only a phase, taking derivatives yields oscillations/fluctuations with the same period $\Delta t$ but with an amplitude multiplied by the factor $\omega = 1/\Delta t$. Now consider the integral: $-A\omega^{-1} \cos\omega t$;





fluctuations of the series obtained by integration are simply multiplied by $\omega^{-1}$, or equivalently by $\Delta t$. Therefore, if the average fluctuations are scaling with $<\Delta T(\Delta t)> \approx \Delta t^H$, then we expect any reasonable definition of fluctuation to have the property that fluctuations of derivatives (for discrete series, differences) or fluctuation of integrals (sums) to also be scaling but with exponents respectively decreased or increased by 1. More generally, it turns out that a filter $\omega^{-H}$ corresponds to an $H^{\text{th}}$ order integral (when $H<0$, to a derivative).

With this in mind, consider a series with $-1 \leq H \leq 0$ and replace it by its "running sum" $s(t_i) = \sum_{j=1}^{i} T(t_j)$ so that its mean differences will have the scaling $<\Delta s(\Delta t)> \approx \Delta t^{1+H}$ with exponent $1+H$ in the useful range between zero and one. But now notice that $\Delta s(\Delta t) = s(t)-s(t-\Delta t)$ is simply the sum of the $T(t)$ over $\Delta t$ values, hence $\Delta s(\Delta t) = \Delta t \overline{T_{\Delta t}}$ where $\overline{T_{\Delta t}}$ is the temporal average over the interval length $\Delta t$. We conclude that when $-1<H<0$, the mean of a time average over an interval of length $\Delta t$ has the scaling $\left\langle \overline{T_{\Delta t}} \right\rangle \approx \Delta t^H$. Indeed, this provides a straightforward interpretation: when $-1<H<0$, $H$ quantifies the rate at which the means of temporal averages decrease as the length of the temporal averaging $\Delta t$ increases. The only technical detail here is that when $H<0$, $\lim_{\Delta t \to \infty} \Delta t^H = 0$, so that this interpretation can only be true if the long term (large $\Delta t$) temporal average of the series is indeed zero. To ensure this, it is sufficient to remove the overall mean $\overline{T}$ of the series before taking the running sum ($T' = T - \overline{T}$). Whenever $-1 \leq H \leq 0$, the resulting average $\overline{T'_{\Delta t}}$ is therefore a useful and easy to interpret definition of fluctuation, called the "anomaly" fluctuation. To distinguish it from other fluctuations we may denote it by $\left( \Delta T(\Delta t) \right)_{anom} = \overline{T'}_{\Delta t}$.

From the way it was introduced by a running sum transformation of the series, we see that the anomaly fluctuation will be dominated by low frequency details whenever $H>0$ and by high frequency details whenever $H<-1$ (see however section B3 for some caveats when $H<0$). The variation of the standard deviation of the anomaly fluctuation with scale is sometimes called "climactogram", [*Koutsoyiannis and Montanari*, 2007]. However, because it is an anomaly statistic, it is only useful when $H<0$, see [*Lovejoy et al.*, 2013].

It turns out that many geophysical phenomena have both $-1 \leq H \leq 0$ and $0 \leq H \leq 1$ regimes (fig. 18) so that it is useful to have a single fluctuation definition that covers the entire range $-1 \leq H \leq 1$. From the preceding, it might be guessed that such a definition may be obtained by combining both differencing and summing; the result is the Haar fluctuation. To obtain $\left( \Delta T(\Delta t) \right)_{Haar}$ it suffices to take the differences of averages (or equivalently, averages of differences, the order doesn't matter): the result is $\Delta T(\Delta t)_{Haar} = \overline{T_{t,t-\Delta t/2}} - \overline{T_{t-\Delta t/2,t-\Delta t}} = \left( s(t) - 2s(t - \Delta t/2) + s(t - \Delta t) \right)/\Delta t$ i.e. the difference of





the average over the first and second halves of the interval between $t$ and $t+\Delta t$ (notice that in terms of the running sum, $s(t)$, it is expressed in terms of second differences). $\left(\Delta T\left(\Delta t\right)\right)_{Haar}$ is a useful estimate of the $\Delta t$ scale fluctuation as long as -

$1 \leq H \leq 1$. Note that we don't need to remove the overall mean $\overline{T}$ since taking differences removes any additive constant.

But what does the Haar fluctuation mean, how do we interpret it? Consider first the Haar fluctuation for a series with $0 \leq H \leq 1$. We have seen that for such series, the anomaly fluctuation changes little: it saturates. Therefore taking the temporal averages of the first and second halves of the interval yields roughly the values at the centre of the intervals so that

$$\left\langle \left(\Delta T\left(\Delta t\right)\right)_{Haar} \right\rangle = C_{diff} \left\langle \left(\Delta T\left(\Delta t\right)\right)_{dif} \right\rangle$$ where $C_{diff}$ is a "calibration" constant of order 1. Conversely, consider the Haar

fluctuation for a series with $-1 \leq H \leq 0$. In this case it is the anomaly fluctuation of the consecutive differences over intervals of length $\Delta t/2$ but for $H<0$, the differences "saturate" (yielding roughly a constant independent of $\Delta t$), so that when $-1 \leq H \leq 0$,

$$\left\langle \left(\Delta T\left(\Delta t\right)\right)_{Haar} \right\rangle = C_{anom} \left\langle \left(\Delta T\left(\Delta t\right)\right)_{anom} \right\rangle.$$ The numerical factors $C_{diff}$, $C_{anom}$ that yield the closest agreements depends on the statistics of the series. However, numerical simulations, much data analysis and the theory discussed below, shows that (especially for $H \approx >0.1$, see fig. B5) using $C_{diff} = C_{anom} = C = 2$ gives quite good agreement so that if we define the "calibrated"

Haar fluctuation $\left(\Delta T\left(\Delta t\right)\right)_{Haar,cal}$ as $= 2\left(\overline{T_{t+\Delta t/2,\Delta t/2}} - \overline{T_{t,\Delta t/2}}\right)$ then we find $\left\langle \left(\Delta T\left(\Delta t\right)\right)_{Haar,cal} \right\rangle \approx \left\langle \left(\Delta T\left(\Delta t\right)\right)_{dif} \right\rangle$ for

$0 \leq H \leq 1$ and $\left\langle \left(\Delta T\left(\Delta t\right)\right)_{Haar,cal} \right\rangle \approx \left\langle \left(\Delta T\left(\Delta t\right)\right)_{anom} \right\rangle$ for $-1 \leq H \leq 0$. Therefore the interpretation of the "calibrated" Haar fluctuation is very close to differences ($0 \leq H \leq 1$) and anomalies ($-1 \leq H \leq 0$) and it has the advantage of being applicable to any series with regimes in the range $-1 \leq H \leq 1$; this covers almost all geophysical fields that have been analysed to date.

So what about other ranges of $H$, other definitions of fluctuation? From the above, the obvious method of extending the range

of $H$'s is to use derivatives or integrals (for series, differences and running sums) which respectively decrease or increase the exponents. Hence for example, a fluctuation that is useful over the range $0 \leq H \leq 2$ can be obtained simply by taking the second differences rather than the first and combining this with summing ($s(t)$ above) to yield the "Quadratic Haar" fluctuation valid over the range $-1 \leq H \leq 2$: $\Delta T\left(\Delta t\right)_{Haar2} = \left(s(t) - 3s\left(t - \Delta t / 3\right) + 3s\left(t - 2\Delta t / 3\right) - s\left(t - \Delta t\right)\right) / \Delta t$. While these higher order fluctuations are quite adequate for estimating exponents $H$, the drawback is that their interpretations are no longer simple;

fortunately, they are seldom needed. More examples are given in the next section.

### B.3 Fluctuations as convolutions, filters and the $H$ limits

To get a clearer idea of what's happening, let's briefly put all of this into the framework of wavelets, a very general method for defining fluctuations. The key quantity is the "mother wavelet" $\Psi(t)$ which can be practically any function as long as it





has an overall mean zero: this is the basic "admissibility condition". The fluctuation at a scale $\Delta t$ at a location $t$ is then simply the integral of the product of the rescaled, shifted mother wavelet:

$$\Delta T\left(\Delta t\right) = \frac{1}{\Delta t} \int T\left(t'\right) \Psi\left(\frac{t-t'}{\Delta t}\right) dt'$$

(64)

$\Delta T(\Delta t)$ is the $\Delta t$ scale fluctuation. The fluctuations discussed in the previous section are the following special cases:

*Difference fluctuations*

$$\left(\Delta T\left(\Delta t\right)\right)_{diff} = T\left(t + \Delta t / 2\right) - T\left(t - \Delta t / 2\right); \quad \Psi\left(t\right) = \delta\left(t - 1/2\right) - \delta\left(t + 1/2\right)$$

(65)

*Anomaly fluctuations*

$$\left(\Delta T\left(\Delta t\right)\right)_{anom} = \frac{1}{\Delta t} \int_{t}^{t+\Delta t} T'\left(t'\right) dt'; \quad T'\left(t\right) = T\left(t\right) - \overline{T\left(t\right)}$$

(66)

$$\Psi\left(t\right) = I_{\left[-1/2,1/2\right]} - \frac{I_{\left[-1/2,1/2\right]}\left(t\right)}{\tau}; \quad \tau >> 1 \qquad \text{where} \qquad I_{\left[a,b\right]}\left(t\right) = \begin{array}{ll} 1 & a \leq t \leq b \\ 0 & otherwise \end{array}$$

(67)

*Haar fluctuations*

$$\left(\Delta T\left(\Delta t\right)\right)_{Haar} = \frac{2}{\Delta t}\left[\int_{t}^{t+\Delta t/2} T\left(t'\right) dt' - \int_{t+\Delta t/2}^{t+\Delta t} T\left(t'\right) dt'\right]; \quad \Psi\left(t\right) = \begin{array}{ll} 2; & 0 \leq t < 1/2 \\ -2; & -1/2 \leq t < 0 \\ 0; & otherwise \end{array}$$

(68)

These fluctuations are related by:





$$\left(\Delta T\right)_{Haar} = \left(\Delta\left(\Delta T\right)_{anom}\right)_{diff} = \left(\Delta\left(\Delta T\right)_{dif}\right)_{anom} \tag{69}$$

These are shown in fig. B1 and some other common wavelets in fig. B2. Table B1 gives definitions and Fourier transforms.

In order to understand the convergence/divergence of different scaling processes it is helpful to consider the Fourier transforms (indicated with a tilde). The general relation between the Fourier transform of the fluctuation at lag $\Delta t$, $\widetilde{\Delta T}$ and the series $\widetilde{T}$ obtained from the fact that the fluctuation is a rescaled convolution:

$$\widetilde{\Delta T}_{\Delta t}\left(\omega\right) = \widetilde{T\left(\omega\right)}\widetilde{\Psi\left(\omega\Delta t\right)} \tag{70}$$

We see that the fluctuation is simply a filter with respect to the original series (its Fourier transform $\widetilde{T}\left(\omega\right)$ is multiplied by

$\widetilde{\Psi\left(\omega\Delta t\right)}$ ). Table B1, shows $\Psi(\omega)$ for various wavelets and figs. B4, B5 shows their modulli squared).

Taking the modulus squared and ensemble averaging ("<.>"), we obtain:

$$E_{\Delta T,\Delta t}\left(\omega\right) = E_T\left(\omega\right)\left|\widetilde{\Psi\left(\omega\Delta t\right)}\right|^2; \quad E_{\Delta T,\Delta t}\left(\omega\right) = \left\langle\left|\widetilde{\Delta T}_{\Delta t}\left(\omega\right)\right|^2\right\rangle; \quad E_T\left(\omega\right) = \left\langle\left|\widetilde{T}\left(\omega\right)\right|^2\right\rangle \tag{71}$$

Where $E_{\Delta T}\left(\omega\right)$ and $E_T\left(\omega\right)$ are the spectra of the fluctuation and the process respectively.

We may now consider the convergence of the fluctuation variance using Parsevals' theorem:

$$\left\langle\Delta T\left(\Delta t\right)^2\right\rangle = 2\int_0^\infty E_{\Delta T,\Delta t}\left(\omega\right)d\omega \tag{72}$$

(we have used the fact the spectrum is a symmetric function). Now consider scaling processes:

$$E_T\left(\omega\right) \approx \omega^{-\beta} \tag{73}$$

and consider the high and low frequency dependence of the wavelet:





$$\left|\widetilde{\Psi(\omega)}\right| \approx \begin{array}{ll} \omega^{H_{low}}; & \omega \to 0 \\ \omega^{H_{high}}; & \omega \to \infty \end{array} \tag{74}$$

Plugging these forms into the integral for the fluctuation variance, we find that the latter only converges when:

$$H_{low} > H' > H_{high}; \quad H' = \frac{\beta - 1}{2} \tag{75}$$

When $H'$ is outside of this range, the fluctuation variance diverges; in practice it is dominated by either the highest ($H'<H_{high}$) or the lowest ($H'>H_{low}$) frequencies present in the sample. We use the prime since this discussion is valid for second order moments - not only for Gaussian processes (where $H'=H$), but also for multifractals where $H'=H-K(2)/2$.

When $H_{low} > H' > H_{high}$, then the variance is finite and the fluctuation variance $\left\langle \Delta T \left( \Delta t \right)^2 \right\rangle$ is:

$$\left\langle \Delta T \left( \Delta t \right)^2 \right\rangle = 2\Delta t^{-1} \int_0^\infty E_T\left(\frac{\omega}{\Delta t}\right) \left|\Psi(\omega)\right|^2 d\omega \tag{76}$$

If $E_T$ is a pure power law (Eq. 63), then we obtain:

$$\left\langle \Delta T \left( \Delta t \right)^2 \right\rangle = 2\Delta t^{-1} \int_0^\infty \left(\frac{\omega}{\Delta t}\right)^{-\beta} \left|\Psi(\omega)\right|^2 d\omega = 2\Delta t^{\beta-1} \int_0^\infty \omega^{-\beta} \left|\Psi(\omega)\right|^2 d\omega \tag{77}$$

the integral is just a constant and since $\beta-1 = 2H'$, as expected we recover the scaling of the fluctuations ($\Delta t^{2H'}$). We also have $C(\Delta t)=C$ is a pure "calibration" constant (independent of $\Delta t$).

The difference between different fluctuations is the integral on the far right. As long as it converges, the difference between

using two different types of fluctuations is therefore the ratio $C$:

$$C = \left( \int_0^\infty \omega^{-\beta} \left|\Psi_{ref}(\omega)\right|^2 d\omega \Big/ \int_0^\infty \omega^{-\beta} \left|\Psi_{Haar}(\omega)\right|^2 d\omega \right)^{1/2} \tag{78}$$

where $\Psi_{ref}$ is the reference wavelet (here we consider differences or anomalies). Fig. B5 shows $C$ for the reference wavelet = the anomaly for $H'<0$ and the difference for $H'>0$; it can be seen that the canonical value $C=2$ is a compromise that is mostly accurate for $H'>0.1$ but is not so bad for negative $H'$. If needed, we could use the theory value from the figure, but in real

world applications, there will not be perfect scaling; there may be zones of both positive and negative $H'$ present so that this might not be advantageous. Finally, we defined $H'$ as the RMS fluctuation exponent, and have discussed the range over which

this second order moment converges for different wavelets characterized by $H_{low}$, $H_{high}$. In the quasi Gaussian case, we have $\langle(\Delta T(\Delta t))^q\rangle \approx \Delta t^{qH}$ so that $H' = H$ and the limits for convergence of the $q = 1$ and $q = 2$ moments are the same. However, more generally, in the multifractal case the limits will depend on the order of moment considered; with the range $H_{low} > H > H_{high}$

being valid for the first order moment.

In summary; when the wavelet falls off quickly enough at high and low frequencies, the fluctuation variance converges to the expected scaling form. Conversely whenever the inequality $H_{low} > H' > H_{high}$ is not satisfied, then the fluctuation variance depends spuriously on either high or low frequency details.

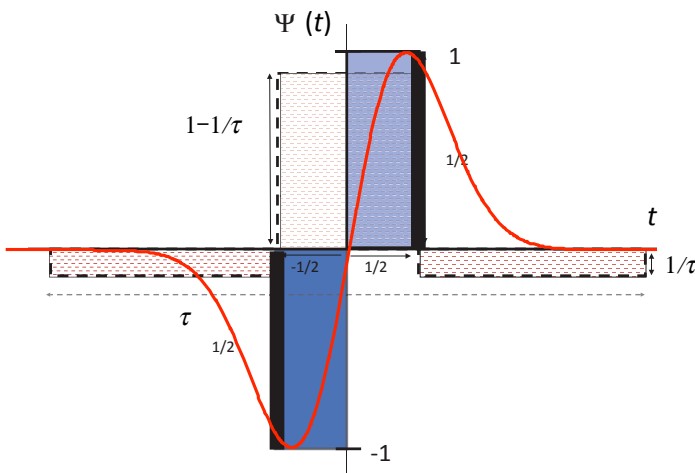

**Fig. B1. The simpler wavelets discussed in the text, see table A1 for mathematical definitions and properties. The black bars symbolizing Dirac delta functions (these are actually infinite in height!) indicate the difference fluctuation (poor man's wavelet), the stippled red indicates the anomaly fluctuation, the blue rectangles the Haar fluctuation (divided by 2) and the red line, the first derivative of the Gaussian.**




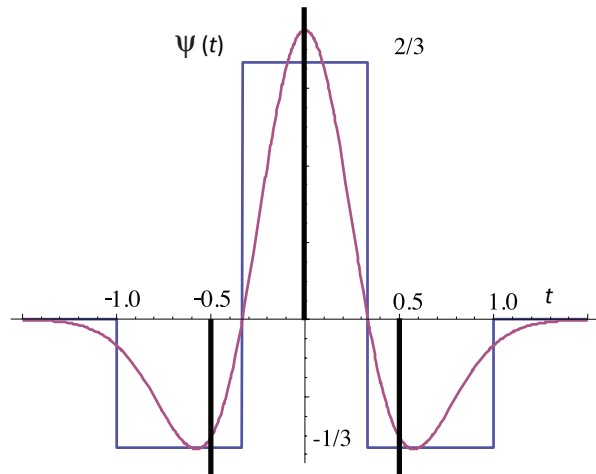


**Fig. B2.** The higher order wavelets discussed in the text: the black bars (representing Dirac delta functions) indicate the second difference fluctuation, the solid blue, the Quadratic Haar fluctuation and the red, the "Mexican Hat wavelet" or second derivative of Gaussian fluctuation.


| Name | Wavelet | Frequency domain | low ω | high ω | H' range |
|---|---|---|---|---|---|
| Poor man's (first difference) | $\delta(t-1/2)-\delta(t+1/2)$ | $2\sin(\omega/2)$ | $\approx \omega$ | $\approx 0$ | $0<H'<1$ |
| $2^{nd}$ difference | $\frac{1}{2}\big(\delta(t+1/2)+\delta(t-1/2)\big)-\delta(t)$ | $\sin^2(\omega/4)$ | $\approx \omega^2$ | $\approx 0$ | $0<H'<2$ |
| Anomaly | $I_{[-1/2,1/2]}(t)-\dfrac{I_{[-\tau/2,\tau/2]}(t)}{\tau};\quad \tau\gg 1$ | $\frac{2}{\omega}\Big(\sin\big(\frac{\omega}{2}\big)-\tau^{-1}\sin\big(\frac{\omega\tau}{2}\big)\Big)$ | $\approx \frac{2}{\omega}\sin\big(\frac{\omega}{2}\big)\to 1$ | $-1$ | $-1<H'<0$ |
| Haar | $\psi(t)=\begin{cases}2; & 0\le t<1/2\\ -2; & -1/2\le t<0\\ 0; & otherwise\end{cases}$ | $8i\omega^{-1}\sin^2\big(\frac{\omega}{4}\big)$ | $\approx \omega$ | $\approx \omega^{-1}$ | $-1<H'<1$ |
| Quadratic Haar | $\psi(t)=\begin{cases}-1/3 & 1/3<t<1\\ 2/3; & -1/3\le t\le 1/3\\ -1/3; & -1\le t<-1/3\\ 0; & otherwise\end{cases}$ | $\frac{2}{3\omega}\Big(3\sin\frac{\omega}{3}-\sin\omega\Big)$ | $\approx \omega^2$ | $\approx \omega^{-1}$ | $-1<H'<2$ |
| First derivative Gaussian | $\Psi(t)\propto \frac{d}{dt}e^{-t^2/2}$ | $\omega e^{-\omega^2/2}$ | $\approx \omega$ | $e^{-\omega^2/2}$ | $-\infty <H'<1$ |
| Mexican Hat | $\Psi(t)\propto \frac{d^2}{dt^2}e^{-t^2/2}$ | $\omega^2 e^{-\omega^2/2}$ | $\approx \omega^2$ | $e^{-\omega^2/2}$ | $-\infty <H'<2$ |




**Table B1   A comparison of various wavelets along with their frequency (Fourier) representation and low and high frequency behaviours.  At the right, the range of H' over which they are useful is indicated.  For the anomaly fluctuation, see the text.  The** 2175 **normalization for the Quadratic Haar is chosen to make it close to the Mexican Hat.**

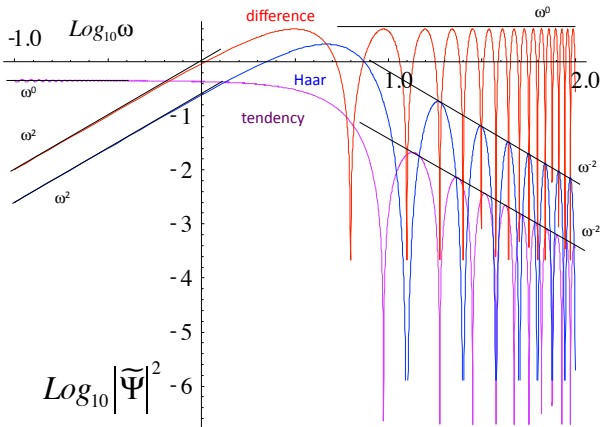

**Fig. B3:  The simple wavelets/fluctuations discussed in the text in the frequency domain.  The power spectrum of the wavelet filter** 2180 $\left|\widetilde{\Psi}\right|^2$ **is shown on a log-log plot ($\widetilde{\Psi}(\omega)$ is the Fourier transform of $\Psi(t)$).   The key asymptotic behaviour is shown by the reference lines.**

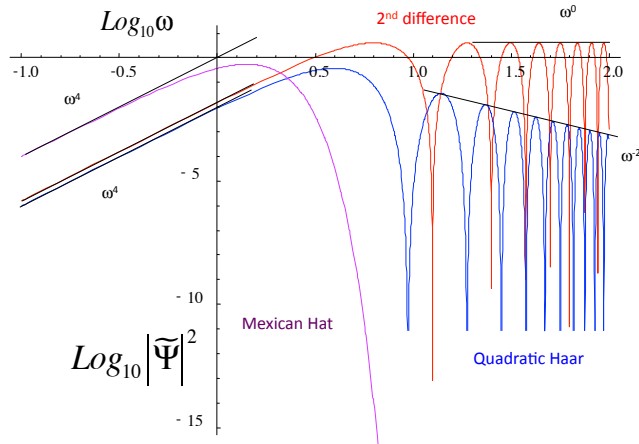




**Fig. B4: The power spectra filters for the higher order wavelets/fluctuations discussed in the text, along with reference lines indicating the asymptotic power law behaviours. Note that the Mexican Hat (second derivative of the Gaussian) decays exponentially at high frequencies, equivalent to an exponent $-\infty$.**

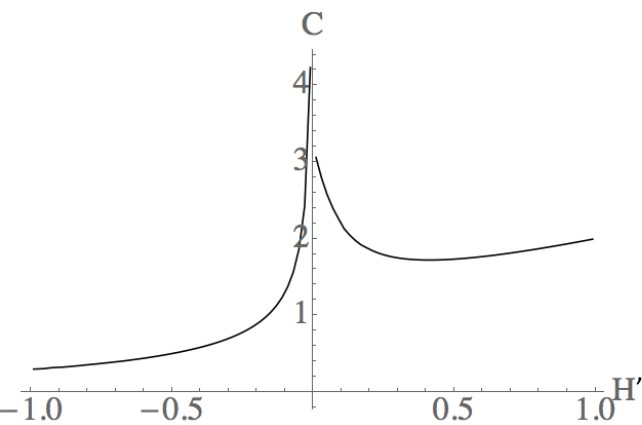

**Fig. B5: The theoretical calibration constant C for the RMS second moment (Eq. 78). Notice that for $H'\approx>0.1$ that it is close to the canonical value $C = 2$.**

### B.4 Stationarity/Nonstationarity

To illustrate various issues, we made a multifractal simulation with $H = -0.3$ (Fig. B6, $C_1 =0.1$, $\alpha=1.8$, section 3.2.2),

and then took its running sum, Fig. B7. Notice that as expected, while B6 with $H<0$ has cancelling fluctuations, Fig. B7 with $H = 1 -0.3 = 0.7$, is "wandering". Now compare the left and right hand sides of Fig. A7 – are they produced by the same stochastic process, or is the "drift" on the right hand side caused by an external agent that was suddenly switched on? When confronted with series such as this that appear to behave differently over different intervals, it is often tempting to invoke the action of a statistically nonstationary process. This is equivalent to the conviction that no conceivable (or at least plausible)

unique physical process following the same laws at all times could have produced the series.

It is worth discussing stationarity in more detail since it is a frequent source of confusion, indeed all manner of "wandering" signals lead to claims of nonstationarity. Adding to this is the fact that common stochastic processes - such as drunkard's walks (Brownian motion) - are strictly speaking nonstationary (see however below).

What's going on? The first thing to be clear about is that statistical stationarity is not a property of a series or even of finite

number of series, but rather of the stochastic process generating the series. *It is a property of an infinite ensemble of series.* It





simply states that the *statistical properties* are translationally invariant along the time axis, i.e. that they do *not* depend on *t*. Statistical stationarity is simply the hypothesis that the underlying physical processes that generate the series are the same at all instants in time. In reality, the "wandering" character of an empirical signal is simply an indication that the realization of the process has low frequency components with large amplitudes – that over the (finite) available range, that it tends to be dominated by them (this is a characteristic of all scaling processes with $H>0$).

However, once one assumes that the process comes from a certain theoretical framework – such as random walks - then the situation is quite different because this more specific hypothesis *can* be tested. But let's take a closer look. A theoretical Brownian motion process $x(t)$ (defined for all $t\geq0$) is an example of a process with stationary *increments*: the rule for the drunk to go either to the left or to the right (an incremental change of position) - is always the same $\Delta x(\Delta t)$ is indeed independent of $t$. The only reason that $x(t)$ is nonstationarity is that the drunkard's starting location is special - let's say $x(0)=0$ - so that the statistics of $x(t)$ depend on $t$. However on any finite domain it is a trivial matter to make the process perfectly stationary: one need only randomize the drunk's starting location $x(0)$. Note that this method doesn't work for pure mathematically defined Brownian motions that are defined on the infinite $x$ axis because it is impossible to define a uniform random starting position between $\pm\infty$ . However, real processes always have finite bounds and – more to the point – real scaling processes always have outer scales, so that in practice (i.e. over finite intervals), even classical random walks can be made stationary.

### B.5 Nonwavelet fluctuations: Detrended Fluctuation Analysis

We mentionned that the Detrended Fluctuation Analysis (DFA) technique was valid for $-1\leq H\leq n$ (with $n$ usually $=1$; linear DFA). Since it is valid for some negative $H$, it is an improvement over simply using differences and has been used in climate analyses (e.g. [*Kantelhardt et al.*, 2001], [*Koscielny-Bunde et al.*, 2006; *Monetti et al.*, 2003]). Unfortunately, the determination of the DFA fluctuations is not simple nor is its interpretation so that often, the units of the fluctuation function are not even given. Other difficulties with the DFA method have been discussed in [*Varotsos and Efstathiou*, 2017].

To understand the DFA, take the running sum $s(t)$ of the empirical time series (in DFA jargon, the "profile", see Fig. B6). Then break $s(t)$ into blocks of length $\Delta t$ and perform regressions with $n^{th}$ order polynomials (in Fig. B7, $\Delta t$ was taken as half the total interval length, and linear regressions were used, i.e. $n = 1$). For each interval length $\Delta t$, one then determines the standard deviation $F$ of the regression residues (see Fig. B8). Since $F$ is a fluctuation in the summed series, the DFA fluctuation in the original series, $(\Delta T)_{DFA}$ is given by:

$$\left(\Delta T\right)_{DFA} = \frac{F}{\Delta t} \tag{79}$$

In words, $(\Delta T)_{DFA}$ is the standard deviation of the residues of polynomial regressions on the running sums of the series divided by the interval length. Note that the usual DFA treatments do not return to the fluctuations in the original series, they analyze





$F$ (not $F/\Delta t$) which is the fluctuation of the running sum, not the original series. Due to its steeper slope (increased by 1), plots of $\log F$-$\log\Delta t$ look more linear than $\log(F/\Delta t)$ - $\log\Delta t$. For series with poor scaling - or with a transition between two scaling regimes – this has the effect of giving the illusion of good scaling and may have contributed to the popularity of the technique! However, in at least several instances in the literature, DFA of daily weather data (with a minimum resolution of 2 days), failed to detect the weather /macroweather transition - breaks in the scaling were spuriously hidden from view at the extreme small $\Delta t$ end of the analysis.

Finally the usual DFA approach defines the basic exponent $a$ not from the mean DFA fluctuation but rather from the RMS DFA fluctuation:

$$\left\langle F^2\left(\Delta t\right)\right\rangle^{1/2} \approx \Delta t^a \tag{80}$$

Comparing this to the previous definitions and using $\Delta T \approx \left(\Delta T\right)_{DFA}$ we see (Eq. 75) that:

$$a = 1 + H' = 1 + H - K(2)/2 \tag{81}$$

Interpretations of the DFA exponent $a$ typically (and usually only implicitly) depend on the quasi Gaussian assumption. For example, one sometimes discusses "persistence" and "antipersistence". These behaviours can be roughly thought of as types of "cancelling" or "wandering" but with respect to Gaussian white noises (i.e. with $H = -1/2$) rather than (as here) with respect to the mean fluctuations that are determined by the sign of $H$ which is equivalent to instead characterizing the scaling of the process with respect to the "conservative" (pure multiplicative, $H = 0$) multifractal process. A "persistent" Gaussian process is one in which the successive increments are positively correlated so that the variance of the process grows more quickly than for Brownian Motion noise, i.e. $a > 1/2$, while an "antipersistent" process on the contrary has successive increments that are negatively correlated so that it grows more slowly ($a < 1/2$, see eq. 81 with $K(2) = 0$ that holds for Gaussian processes). For Gaussian processes this distinction is associated with precise mathematical convergence/divergence issues. However, if the process is non Gaussian this criterion is not relevant and the classification itself is not very helpful whereas the sign of the fluctuation exponent $H$ remains fundamental (in particular, in the more general multifractal case). In terms of interpretation, the drawback of the $a > 1/2$, $a < 1/2$ classification is that the neutral (reference) case of persistence/antipersistence ($a = 1/2$) is white noise which itself is highly "cancelling" (it has $H = -1/2$), so that even for Gaussian processes, the persistence/antipersistence classification is not very intuitive.

In applications of the DFA method, much is said about the ability of the method to remove nonstationarities. Indeed, it is easy to see that an $n^{th}$ order DFA analysis removes an $n^{th}$ order polynomial in the summed series, i.e. an $n$-$1^{th}$ order polynomial in the original series. In this, it is no different from the "Mexican hat" or other wavelets and their higher order derivatives (or to the simple polynomial extensions of fluctuations such as the Quadratic Haar fluctuation discussed above). In any case, it removes such trends at all scales, not only at the largest so that it is misleading to describe it as removing nonstationarities. In addition, the stationarity assumption is still made with respect to the residuals from the polynomial regression – just as with


the wavelet based fluctuations. If one only wants to remove nonstationarities, it should be done as a "pretreatment" i.e. trends should only be removed over the entire series not over each interval, over each segment within the series. Finally, the most

2270 common and strong geophysical nonstationarities are due to the diurnal and annual cycles, and none of these techniques remove oscillations.

We conclude that the only difference between analyzing a data series with the DFA or with wavelet based fluctuation definitions is the extra and needless complexity of the DFA – the regression part – that makes its interpretation and mathematical basis unnecessarily obscure. Indeed, Fig. B9 numerically compares spectra, (Haar) wavelets and DFA exponent

estimates showing that Haar wavelets are at least as accurate as DFA but have the added advantage of simplicity of implementation and simplicity of interpretation.

If the underlying process is multifractal, then one naturally obtains huge fluctuations (in space, huge structures, "singularities") but these are totally outside the realm of quasi Gaussian processes so that when they are inappropriately interpreted in a quasi Gaussian frameworks, they will be are often mistakenly treated as nonstationarities (in space, mistakenly as inhomogeneities.

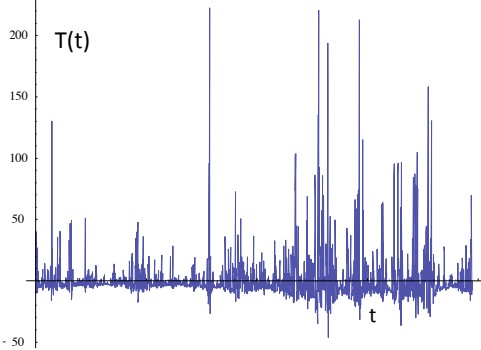

**Fig. B6: A simulation of a (multifractal) process with $H = -0.3$, $C_1=0.1$, $\alpha=1.8$ showing the tendency for fluctuations to cancel.**

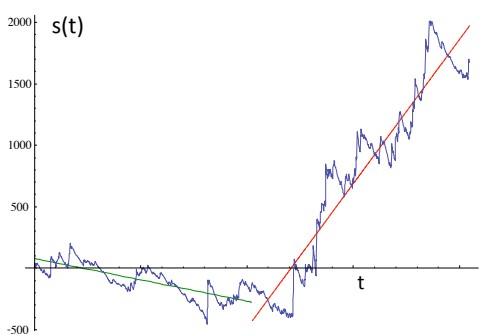

**Fig. B7: The running sum ($s(t)$) of the previous realization with $H = 1+(-0.3) = 0.7$; note the wandering character. Also shown are**
2285 **the two regression lines used to define the fluctuations of the Detrended Fluctuation Analysis technique, for fluctuations with lags**





**Δ*t* half of the length shown. For each regression, the fluctuation is estimated by the root mean square of the residues from the lines; see Fig. B8.**

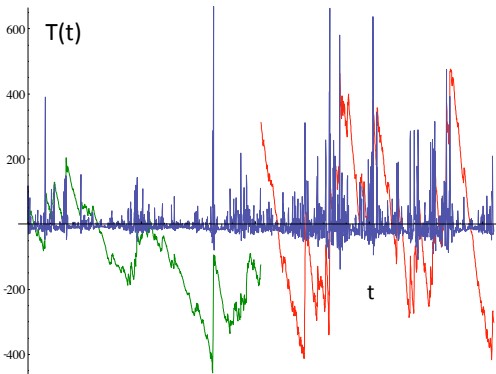

**Fig. B8: The H = -0.3 series of Fig. B6 with the residues of the two regression lines in Fig. B7 used to determine the DFA fluctuations for lags t half of the length shown (blown up by factors of three, green, red). For each regression, the fluctuation is estimated by the root mean square of the residues from the lines.**


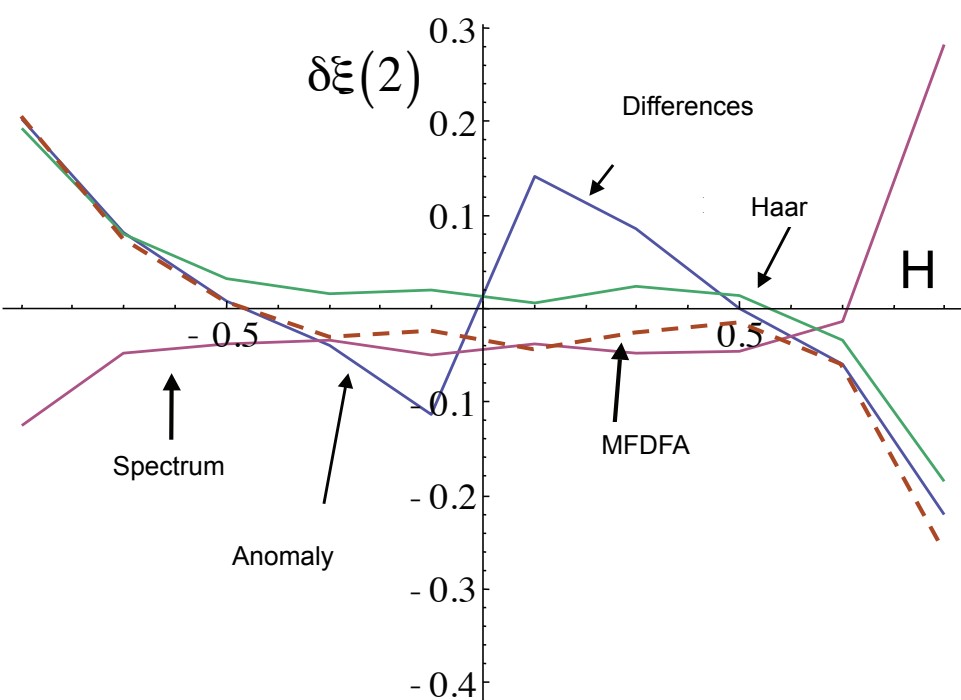

**Fig. B9: Comparison of the bias in estimates of the second order structure function ($\delta\xi(2)$), obtained from numerical multifractal simulations with parameters $\alpha = 1.8$, $C_1 = 0.1$ and with the fluctuation $H$ as indicated. Theoretically, the (unbiased) second order structure function exponent $\xi(2) = 2H$-$K(2)$ and with these parameters, $K(2) = 0.18$. For each value of H from -9/10 to +9/10 (at intervals of 1/10) 50 realizations of the process were analyzed, each of length $2^{14} = 16384$ points. Difference fluctuations were only applied to the $H > 0$ cases and anomaly fluctuations for $H < 0$. Spectra, Haar and Multifractal Detrended Fluctuation Analysis (MFDFA) were applied over the whole range. We see that the latter methods are quite accurate (to within about —0.05) over the range $\approx$ -0.7 < $H$ < $\approx$ 0.7. Over this range particular the Haar fluctuations have a bias of about +0.01 while the MFDFA have a bias of about -0.02. In comparison, the difference and anomaly fluctuations have stronger biases (of about ±0.1) near the limits of their ranges i.e. when $|H| \approx <0.2$. Adapted from [*Lovejoy and Schertzer*, 2012b].**

### Code Availability

Much software is available from the site:

http://www.physics.mcgill.ca/~gang/software/index.html

### Data availability

This review contains no original data analysis.





**Competing Interests**

The author is a member of the NPG editorial board.





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
