# Peer review of "Review Article: Scaling, dynamical regimes and stratification: How long does weather last? How big is a cloud?"

_Nonlinear Processes in Geophysics, 2023_

## Referee Comment (RC1)

**Review of Scaling, dynamical régimes and stratification: How long does weather last? How big is a cloud?    S. Lovejoy**

GENERAL

I regard this paper as a *tour de force*, well worthy of publication in *NPG*. However, there it would have an element of preaching to the choir. The message really needs to be put in front of the core atmospheric science and climate readership, whose reluctance to embrace new thinking is one of the targets of the paper. *JAS, MWR, QJRMS, Climatic Change, Revs Geophys, npj-Climate and Atmospheric Science* are all possibilities immediately coming to mind. Not all of course might accommodate 150 pages.

COMMENTARY

Line

49: the month is based on the (current) 29.7-day period of the moon's orbit around Earth. I agree the calendar as widely used wavers between 28 and 31.

83: the dissipation time has been argued to be on molecular scales, much shorter than millimetric or millisecond - scales which reflect the resolution of observational instruments. See https://doi.org/10.3390/meteorology1010003. Dissipation is radiation of IR photons to space. Does OLR scale? It should.

98: Include scales upward from the mean free path at STP and even more can be added. Maxwell-Boltzmann volumes of gas do not exist in the atmosphere - their continuous translational symmetry is broken by persistence of molecular velocity after collision.

128: Virtually all quantitative images of clouds are two dimensional, or one-dimensional slices. How can three-dimensional variability be addressed? Or should it be 23/9 D?

164: I guess that answers my question at line 128.

187: "doe"? Reproduced or Adapted?

195: row.

212: 'expert judgement' should be referenced - and viewed sceptically given the nonlinearity of the system being dealt with.

223: the average scale height is 7.4 km

228: cite Lovejoy et al, *GRL,* **34,** L15802 (2007)

238-9: 'This review" reads ambiguously to me. It is clearly not the current paper, but nor is it Lovejoy (2019).

243: developed.

262: suggest colon after covered.

286-8: this sentence needs punctuation.

312: but the variance doesn't converge! $(1.5<\alpha<2)$.

338: See Kadau et al, *Phil Trans Roy Soc A* **368,** 1547-1560 (2010). Also see above comments on 'millimetric' and dissipation.

359: wasn't it the turbulent Loch Lomond?

429 et seq: Heisenberg, von Weiszäcker, Onsager all got the same result as Kolmogorov's 1941 paper but did so immediately after WW2 and in ignorance of Kolmogorov's paper. Landau criticised Kolmogorov in 1944 for ignoring intermittency. As a matter of historical interest, Heisenberg did his doctorate for Sommerfeld and Wien at Munchen on the transition from laminar to turbulent flow. Sommerfeld wanted to pass him with a high grade, but Wien wanted to fail him. A compromise was reached, and Heisenberg got his doctorate with the lowest grade of pass. He then left for Born at Gottingen on the grounds that turbulence was too difficult - with well-known results.

525-538: Figure 13 is very telling. As is Figure 14. Personally, I think Ghil's recent approach is inexcusable. NOAA has even less excuse.

667-668: Eliminate one of the "to's"

788-792: Is it not Lagrangian sampling of Eulerian GCM-based analyses?

801: 'Galilean' - and elsewhere.

821-822: Grammar needs revision.

904: Lévy - and elsewhere.

918: no apostrophe in the possessive its.

1066-1076: Specify units of $L_{eff}$ in either Table 1 or its caption.

1069: Several typos here. 'intermittency', alpha not a, exponent.

1152: Reynolds' not Reynold's.

1174: typo - reflectivity factor

1298: 'estimates' not 'estates'?

1304: Reynolds'

1322:  is the probability.......?

1379: 'special'

1450: Hovde et al, *Int. J. Remote Sensing* **32,** 5891-5918 (2011) and https://doi.org/10.3390/atmos12111414 might add to this data and section.

1546: its not it's

1712: test is to consider.....

1723-1730: This argument belongs in the text rather than the figure caption. Dynamical meteorologists obeying 23/9 scaling, however inadvertently, is worth more prominence.

1768: $10^9$

1772: 55 great circle degrees?

1815 et eq: Outgoing IR radiation is critically affected by clouds. Have OLR fields been examined for scaling?

1855-6: it's black and white in what I downloaded. Also 1865-6.

1888: English needs amendment.

1944 et seq: A Maxwell-Boltzmann (equilibrated) gas has
continuous translational symmetry. It is broken at molecular and
photon scales, see https://doi.org/10.3390/meteorology1010003

1966: scale height is 7.4 km

---

## Author Comment (AC1)

**Review of Scaling, dynamical régimes and stratification: How long does weather last? How big is a cloud?    S. Lovejoy**

GENERAL

I regard this paper as a *tour de force*, well worthy of publication in *NPG*. However, there it would have an element of preaching to the choir. The message really needs to be put in front of the core atmospheric science and climate readership, whose reluctance to embrace new thinking is one of the targets of the paper. *JAS, MWR, QJRMS, Climatic Change, Revs Geophys, npj-Climate and Atmospheric Science* are all possibilities immediately coming to mind. Not all of course might accommodate 150 pages.

SL:  Thanks for the encouragement!  It may be that the NPG venue is not the most appropriate, but at least – since it is open access -  this may not make so much difference anymore.  And there is the need for another book.

COMMENTARY

Line

49: the month is based on the (current) 29.7-day period of the moon's orbit around Earth. I agree the calendar as widely used wavers between 28 and 31.

SL: Thanks, I clarified that!

83: the dissipation time has been argued to be on molecular scales, much shorter than millimetric or millisecond - scales which reflect the resolution of observational instruments. See https://doi.org/10.3390/meteorology1010003. Dissipation is radiation of IR photons to space. Does OLR scale? It should.

SL: Thanks for the reference, I added the information and the reference to the text.

98: Include scales upward from the mean free path at STP and even more can be added. Maxwell-Boltzmann volumes of gas do not exist in the atmosphere - their continuous translational symmetry is broken by persistence of molecular velocity after collision.

SL:  The figure caption does not make reference to the dissipation scale, only the range of scales visible in the image.

128: Virtually all quantitative images of clouds are two dimensional, or one-dimensional slices. How can three dimensional variability be addressed? Or should it be 23/9 D?

SL:  I try to address this in the sections that follow, especially section 4.

164: I guess that answers my question at line 128.

SL: Yes.

187: "doe"? Reproduced or Adapted?

SL: Adapted, thanks.

195:row.
SL: Thanks.

212:'expert judgement' should be referenced - and viewed sceptically given the nonlinearity of the system being dealt with.

SL:  Conventionally here, the nonlinearity is taken into account by the "climate feedback" parameter, the inverse of the climate sensitivity.  To some degree of approximation, the temperature response of the earth to a small perturbation is linear (anthropogenic forcing is of the order of 2.5W/m² compared to an average (absorbed) solar radiation of 240 W/m².

223: the average scale height is 7.4 km
SL:  OK.

228: cite Lovejoy et al, *GRL,* **34,** L15802 (2007)
SL:  OK, thanks!

238-9: 'This review" reads ambiguously to me. It is clearly not the current paper, but nor is it Lovejoy (2019).

SL: Thanks, I added "the present review"

243: developed.

SL: Thanks.

262: suggest colon after covered.
SL: Thanks.

286-8: this sentence needs punctuation.
SL: Thanks.

312: but the variance doesn't converge! ($1.5<\alpha<2$).

SL: The variance of the generator (the log) of the process doesn't converge, but the variance of the process itself will generally (but not necessarily) converge.

338: See Kadau et al, *Phil Trans Roy Soc A* **368**, 1547-1560 (2010). Also see above comments on 'millimetric' and dissipation.

SL: Thanks for the reference, I have included this and a few to Tuck's work.

359: wasn't it the turbulent Loch Lomond?

SL: Thanks we were both almost right, it was a pier (not a bridge) and it was Loch Long not Loch Lomond!

429 et seq: Heisenberg, von Weiszäcker, Onsager all got the same result as Kolmogorov's 1941 paper but did so immediately after WW2 and in ignorance of Kolmogorov's paper. Landau criticised Kolmogorov in 1944 for ignoring intermittency. As a matter of historical interest, Heisenberg did his doctorate for Sommerfeld and Wien at Munchen on the transition from laminar to turbulent flow. Sommerfeld wanted to pass him with a high grade, but Wien wanted to fail him. A compromise was reached, and Heisenberg got his doctorate with the lowest grade of pass. He then left for Born at Gottingen on the grounds that turbulence was too difficult - with well-known results.

SL: Thanks. Some of this is in my 2019 book, ch. 4.

525-538: Figure 13 is very telling. As is Figure 14. Personally,
I think Ghil's recent approach is inexcusable. NOAA has even
less excuse.

SL: Yes!

667-668: Eliminate one of the "to's"

SL: Thanks.

788-792: Is it not Lagrangian sampling of Eulerian GCM-based
analyses?

SL: I added the part in parentheses: "these space time diagrams are Lagrangian
(albeit deduced from Eulerian data and reanalyses)."

801: 'Galilean' - and elsewhere.

SL: Thanks.

821-822: Grammar needs revision.

SL: Thanks.

904: Lévy - and elsewhere.

SL: Thanks.

918: no apostrophe in the possessive its.

SL: Thanks.

1066-1076: Specify units of $L_{eff}$ in either Table 1 or its
caption.
SL: Thanks.

1069: Several typos here. 'intermittency', alpha not a,
exponent.
SL: Thanks.

1152: Reynolds' not Reynold's.

SL: Thanks.

1174: typo - reflectivity factor

SL: Thanks.

1298: 'estimates' not 'estates'?

SL: Thanks.

1304: Reynolds'

SL: Thanks.

1322: is the probability.......?

SL: Thanks.

1379: 'special'

SL: Thanks.

1450: Hovde et al, *Int. J. Remote Sensing* **32,** 5891-5918 (2011) and https://doi.org/10.3390/atmos12111414 might add to this data and section.

SL: Added in several places, thanks.

1546: its not it's

SL: Thanks.

1712: test is to consider.....

SL: Thanks.

1723-1730: This argument belongs in the text rather than the figure caption. Dynamical meteorologists obeying 23/9 scaling, however inadvertently, is worth more prominence.

SL: Yes, good idea, I modified the text and caption accordingly.

1768: $10^9$
SL: Thanks.

1772: 55 great circle degrees?

SL: This plot is in flat space, so usual angles!

1815 et eq: Outgoing IR radiation is critically affected by clouds. Have OLR fields been examined for scaling?

SL: Yes, from IR imagery, for example as analysed in fig. 8, 25, 26.

1855-6: it's black and white in what I downloaded. Also 1865-6.

SL: Thanks.

1888: English needs amendment.

SL: Thanks.

1944 et seq: A Maxwell-Boltzmann (equilibrated) gas has continuous translational symmetry. It is broken at molecular and photon scales, see https://doi.org/10.3390/meteorology1010003

SL: Thanks.

1966: scale height is 7.4 km

SL: Thanks.  I put "≈" in front of it: only the order of magnitude is important here.

---

## Author Comment (AC2)

Referee 2                                                                                                    1

Review of "Review Article: Scaling, dynamical regimes and stratification: How long does weather last? How big is a cloud?" by S.    3
Lovejoy                                                                                                       4

                                                                                                             5

This article reviews the developments of scaling approaches over the last few decades. Scaling approaches have led to novel    6
insights into atmospheric dynamics and recently provide the building blocks of novel prediction models and climate response    7
models. A review on this topic is needed and will be helpful in spreading the scaling approach to a wider range of scientists.    8

                                                                                                             9

While such a review is needed, I am not sure if the article in its present form will be able to reach a wider audience. My major    10
issue is with the length of the article. In my opinion the article is too long for a paper, which one could read in one sitting. I could    11
imagine it as a foundation of a book by adding more background material to make it easier to understand the topic.    12

 SL.  Thanks for the positive reaction!                                                                      13

This is an "old school" review, i.e. one that seeks to be fairly complete.  However, you are right that it could be used as    14
the foundation for a longer book.   I could add that it isn't easy to find the appropriate venue for this review -  NPG is    15
in fact designed for this type of subject matter.  However, the resulting publication will be open access, so that I hope    16
that it can still circulate widely.                                                                          17

NPG has no formal page limit for review articles but I encourage the author to shorten it with the reader in mind. The article is    18
well written and I find it hard to point to any obvious location which can be easily shortened. One possibility could be to have a    19
~20 page overview article and put the remainder into supplementary material.                                 20

SL: At this point a shorter review would simply be another paper!  I anticipate writing a shorter review of the climate    21
part of the paper in the next months.                                                                        22

Some more detailed comments:                                                                                 23

1) Line 23-24: This sentence reads awkward.                                                                   24

 SL: Thanks, fixed.                                                                                          25

2) Line 67: Why "lag"? An interval is not a lag. Am I missing something?                                      26

SL: "Lag" is sometimes used in autocorrelation functions for example.  I removed it since it didn't add clarity.    27

3) Line 96: cloud -> clouds                                                                                   28

SL: Thanks.                                                                                                    29

4) Line 125: range scaling -> range of scaling                                                                 30

SL: Thanks.                                                                                                     31

5) Line 134: levels quantify -> levels to quantify                                                             32

SL: Thanks.                                                                                                     33

6) Line 135: would expected -> would be expected                                                              34

SL: Thanks                                                                                                      35

7) Eq. 3: Hz -> H_z                                                                                             36

SL: Thanks, z is a subscript.                                                                                   37

8) The citation style is often incorrect. E.g. line 277: [Mandelbrot, 1981] termed -> Mandelbrot [1981] termed   38

and many other locations                                                                                        39

SL: Thanks                                                                                                       40

9) Eq. 9: An explanation for 2 in \zeta(2) is missing.                                                          41

SL: Added.                                                                                                      42

10) In many parts of the article the author relies mainly on his own studies and of his collaborators. It would be good to include a   43
more diverse set of studies which independently confirms the conclusions.                                       44

SL: I will add more references, it would be helpful if the referee could make some suggestions?                 45

11) Line 592: that is only true for \zeta(2)<-1                                                                 46

SL: I'm not sure what is suggested here.  The text seems to be correct as is?                                   47

12) Line 597: There is a huge class of wavelets. Which wavelet are you actually referring to?                   48

Later it becomes clear that Haar wavelets are used.                                                             49

SL: The paragraph is valid for all wavelets, - their relationship with fluctuations - it is a preparation for the   50
discussion of Haar wavelets and fluctuations that comes later.                                                  51

13) Line 604-605: Haar wavelets have some nice properties but in my experience their spectra are more noisy than DFA spectrum for example.

SL:  The DFA fluctuations are less noisy only because they are fluctuations of the running sum of the process, not of the process itself.  When DFA fluctuations of the process are used, they are just as variable as the Haar fluctuations.  In fact the smoothness – lack of noise – in the DFA fluctuations is actually a spurious hiding of the true noisiness.  This has been demonstrated by numerous numerics including in the cited references.  (I added material).

14) Line 821-822: This sentence is odd.

SL: Fixed.

15) Line 911: It might be good to use H only for the Hurst exponent and another symbol when a more general exponent is implied. That would potentially avoid any confusion.

SL: I added a paragraph on this included the suggestion on notation.

16) Lines 933-935: How GCMs become effectively stochastic on time scales longer than 10 days needs to be better explained.

 SL:  I added a sentence:

"Due to their sensitivity to initial conditions, there is an inverse cascade of errors [*Lorenz*, 1969], [*Schertzer and Lovejoy*, 2004] so that beyond the predictability limit,  small scale errors begin to dominate the global scales so that the GCMs  effectively become stochastic."

17) Line 949 and following: I am having a hard time understanding this part.

SL: I have added some extra equations to make this more explicit.

18) Section 5: This is a nice summary but a good review article should also point out knowledge gaps and future research directions for the community.

SL:  Yes, I will add material on this.

---

## Referee Report (RR1)

**2nd review of npg-2023-5**

This revised version reads very well, and I strongly recommend it for publication.

There is a handful of typographical corrections:-

line 121      'Reproduced'
line 298/9    ...... geodata have .......
line 508      asterisks?
line 728      hyphenation and spacing need adjustment
line 989      delete 'projection' and leave 'projections'